Potential for Zika virus transmission
by mosquitoes in temperate climates.
*Proc. R. Soc. B* **287**: 20200119.

health and disease and epidemiology

EIP, climate change, *Aedes*, $R_0$, Zika

**Author for correspondence:**
Marcus S. C. Blagrove
e-mail: marcus.blagrove@liverpool.ac.uk

Electronic supplementary material is available
online at https://doi.org/10.6084/m9.figshare.
c.5036414.

# Potential for Zika virus transmission by mosquitoes in temperate climates

Marcus S. C. Blagrove[1,2], Cyril Caminade[1,2], Peter J. Diggle[3],
Edward I. Patterson[4], Ken Sherlock[1], Gail E. Chapman[1], Jenny Hesson[1,6],
Soeren Metelmann[1,2], Philip J. McCall[5], Gareth Lycett[5], Jolyon Medlock[7],
Grant L. Hughes[4], Alessandra della Torre[8] and Matthew Baylis[1,2]

[1]Department of Epidemiology and Population Health, Institute of Infection and Global Health, University of
Liverpool, Liverpool Science Park—Innovation Centre 2, 131 Mount Pleasant, Liverpool L3 5TF, UK
[2]National Institute of Health Research Health Protection Research Unit in Emerging and Zoonotic Infections,
University of Liverpool, Liverpool, UK
[3]Lancaster Medical School, University of Lancaster, Lancaster, UK
[4]Departments of Vector Biology and Tropical Disease Biology, Centre for Neglected Tropical Diseases, and
[5]Vector Biology Department, Liverpool School of Tropical Medicine, Liverpool, UK
[6]Department of Medical Biochemistry and Microbiology, Zoonosis Science Center, Uppsala University,
Uppsalam, Sweden
[7]Medical Entomology and Zoonoses Ecology, Public Health England, HPA, Salisbury, UK
[8]Department of Public Health & Infectious Diseases, Sapienza University of Rome, Laboratory.
Affiliated to Instituto Pasteur Italia—Fondazione Cenci Bolognetti, Rome, Italy

MSCB, 0000-0002-7510-167X; SM, 0000-0002-2394-5301; GLH, 0000-0002-7567-7185

Mosquito-borne Zika virus (ZIKV) transmission has almost exclusively been
detected in the tropics despite the distributions of its primary vectors extend-
ing farther into temperate regions. Therefore, it is unknown whether ZIKV's
range has reached a temperature-dependent limit, or if it can spread into
temperate climates. Using field-collected mosquitoes for biological rel-
evance, we found that two common temperate mosquito species, *Aedes
albopictus* and *Ochlerotatus detritus*, were competent for ZIKV. We orally
exposed mosquitoes to ZIKV and held them at between 17 and 31°C, esti-
mated the time required for mosquitoes to become infectious, and applied
these data to a ZIKV spatial risk model. We identified a minimum tempera-
ture threshold for the transmission of ZIKV by mosquitoes between 17 and
19°C. Using these data, we generated standardized basic reproduction
number $R_0$-based risk maps and we derived estimates for the length of the
transmission season for recent and future climate conditions. Our standar-
dized $R_0$-based risk maps show potential risk of ZIKV transmission
beyond the current observed range in southern USA, southern China and
southern European countries. Transmission risk is simulated to increase
over southern and Eastern Europe, northern USA and temperate regions
of Asia (northern China, southern Japan) in future climate scenarios.

## 1. Introduction

The Zika virus (ZIKV) is a mosquito-borne flavivirus first identified in the
Ziika Forest of Uganda in 1947 [1]. Though infection is asymptomatic in 80%
of cases [2], a small proportion of patients develop clinical symptoms, including
the autoimmune condition Guillain–Barré syndrome [3]. There is also strong
evidence that ZIKV infection of expectant mothers can lead to congenital
Zika syndrome (including microcephaly) in the developing fetus [4]. The emer-
gence of these neurological syndromes associated with ZIKV in South America
led the World Health Organization to declare a Public Health Emergency of
International Concern from February to November 2016.

The primary vectors of ZIKV are generally considered to be *Aedes aegypti* and *Ae. albopictus* [5–7], both of which are widespread in tropical regions where ZIKV transmission has been reported. A total of 18 tropical species have been found to be infected in the field, with eight previously confirmed in the laboratory (reviewed in [8–10]). While both primary vectors, especially *Ae. albopictus*, also occur in some temperate regions, autochthonous transmission of virus has so far been largely restricted to the warmer parts of this species range [11].

Temperature has a profound effect on the ability of insect vectors to transmit pathogens, especially viruses. Decreasing environmental temperature increases the time taken for the vector to become infectious after taking an infected blood meal (termed the extrinsic incubation period (EIP)) [12]; for example, *Culex tarsalis* females infected with West Nile virus (WNV) can become infectious at 5 days post-infection at 30°C, but require 36 days at 14°C [13]. At these low temperatures, there may be insufficient time for the pathogen to replicate, invade the salivary gland and become infectious within the lifespan of the vector, imposing a critical threshold temperature for virus transmission [14,15]. It is essential to understand whether this current limitation in the range of ZIKV is a result of the climate being too cold for virus transmission by vectors, or if ZIKV transmission is theoretically possible in temperate climates but has not yet occurred (e.g. because has not yet become established). With large human populations living in temperate climates it is also important to estimate the extent of ZIKV transmission in such climates, both currently and using predicted future climate scenarios.

Currently, there are no peer-reviewed data that comprehensively show at what temperature ranges ZIKV can be transmitted by temperate mosquitoes such as *Ae. albopictus*. *Aedes albopictus* is already well established in temperate Europe, the USA and Asia. Furthermore, while low-temperature transmission has been studied for the tropical *Ae. aegypti* [16], no accurate minimum temperature for transmission by temperate-adapted mosquito species has been established. Such data are essential for the estimation of ZIKV transmission risk in temperate regions. Here, we test the competence of wild (or recently colonized) temperate mosquitoes *Ae. albopictus* and *Ochlerotatus detritus* and derive the EIP of ZIKV over a wide range of temperatures. We then use these experimental data from *Ae. albopictus* to model and estimate temperate areas at risk from the virus by combining data on reported *Ae. albopictus* presence [17] and current and future climate using an ensemble of calibrated general circulation models driven by the standard representative concentration pathways (RCPs) scenarios from the Intergovernmental Panel on Climate Change (IPCC).

## 2. Material and methods

### (a) Larval collection and rearing

Individuals of three species of UK mosquito with previously demonstrated or suspected flavivirus competence [18–20] were collected from the field. *Ochleratatus detritus* immatures (4th instar larvae and pupae) were collected from marshland by Little Neston, Cheshire, UK (GPS coordinates: 53°16′37″ N 3°04′06″ W). *Culex pipiens pipiens* eggs were collected from Liverpool, Merseyside, UK (53°25′32″ N 2°50′09″ W). *Culiseta annulata* immatures were collected from Ness Botanic Gardens,

Cheshire, UK (53°16′13″ N 3°02′55″ W). All UK mosquitoes were collected in April–May 2016, and reared in the water which they were found (with no additional diet supplements) at ambient UK conditions. They were allowed to emerge and mate in 30 × 30 × 30 cm BugDorm cages in Leahurst Campus, University of Liverpool, until approximately one week post-emergence, where they were transported to an insectary at 25°C, 12 : 12 light : dark and 70% RH. Recently, colonized (third generation) *Ae. albopictus* (Verano strain), originating from individuals collected from the Verano Cemetery in Rome (41°54′05″ N 12° 31′14″ E), were provided by Università Sapienza (Rome, Italy). Adults were allowed to emerge and mate in 30 × 30 × 30 cm cages (BugDorm, Taiwan). *Aedes albopictus* were reared in an insectary at 25°C, 12 : 12 light : dark and 70% RH. Due to the large number of conditions and samples, experiments were conducted in batches rather than all at once.

### (b) Virus/blood meal preparation

At 7 days post-emergence, female adults were removed, transferred and counted into 1 l cylindrical polypropylene DISPOSAFE containers (Microbiological Supply Company, UK), with a fine mesh covering the container opening and stored for 24 h without access to sugar or water. Blood meals containing ZIKV were provided for 3 h using a Hemotek feeding system and 'Hemotek feeding membrane' (Hemotek Ltd, UK) odorized by rubbing against human skin (greater than 0.4 log decrease in viral titre was observed over this period). The PE243 isolate of ZIKV from Brazil was used for all infection experiments, virus was cultured in Vero E6 cells, using Dulbecco's Modified Eagle Medium (Thermo Fisher Scientific) and 10% fetal bovine serum (Thermo Fisher Scientific), and prepared to a final titre of $1 \times 10^6$ PFU ml$^{-1}$ in 50% human blood. Blood was obtained from NHS transfusion service in Speke, Merseyside, UK. Blood was heparinized to prevent coagulation. Unfed adults were removed from the cage, and the fed mosquitoes were incubated at the appropriate temperature and 70% RH for the respective time period with a 12 : 12 light : dark cycle.

### (c) Incubation and saliva extraction

Mortality and competence of wild-obtained *Oc. detritus* and colony *Ae. albopictus* were tested at six different temperatures: 17°C, 19°C, 21°C, 24°C, 27°C and 31°C. Adult females were sacrificed at eight time points: 0, 5, 7, 10, 14, 17, 21 and 28 days post-infection. As numbers of mosquitoes available were much lower, *Cx. pipiens pipiens* and *Cs. annulata* were only incubated at 21°C and tested at 17 days post-infection. Saliva was extracted using the methods described in [18], mosquitoes were allowed to salivate for 15 min into 10 µl light mineral oil (NF/FCC) (Thermo Fisher Scientific), and expelled directly into TRIzol (Thermo Fisher Scientific); saliva secretion was observed under a dissection microscope from all individuals. Once saliva had been extracted, the entire body was also submerged in TRIzol (Thermo Fisher Scientific). Samples were stored at −80°C until RNA extraction.

### (d) Detection of virus

RNA was extracted according to TRIzol (Thermo Fisher Scientific) instructions, and cDNA was generated using SuperScript VILO (Thermo Fisher Scientific). Taqman (Thermo Fisher Scientific) qRT-PCR was used to detect the presence of ZIKV RNA in the samples. The primers and probes used were: sense AAR TACACATACCARAACAAAGTGGT, antisense TCCRCTCCCY CTYTGGTCTTG, probe 6Fam-CTYAGACCAGCTGAAR-Tamra; amplification regime, 5 min 95°C, 40 cycles of 15 s at 95°C and 60 s at 60°C [21]. The limit of detection of the analysis was measured by a known titre of live virus submitted to RNA

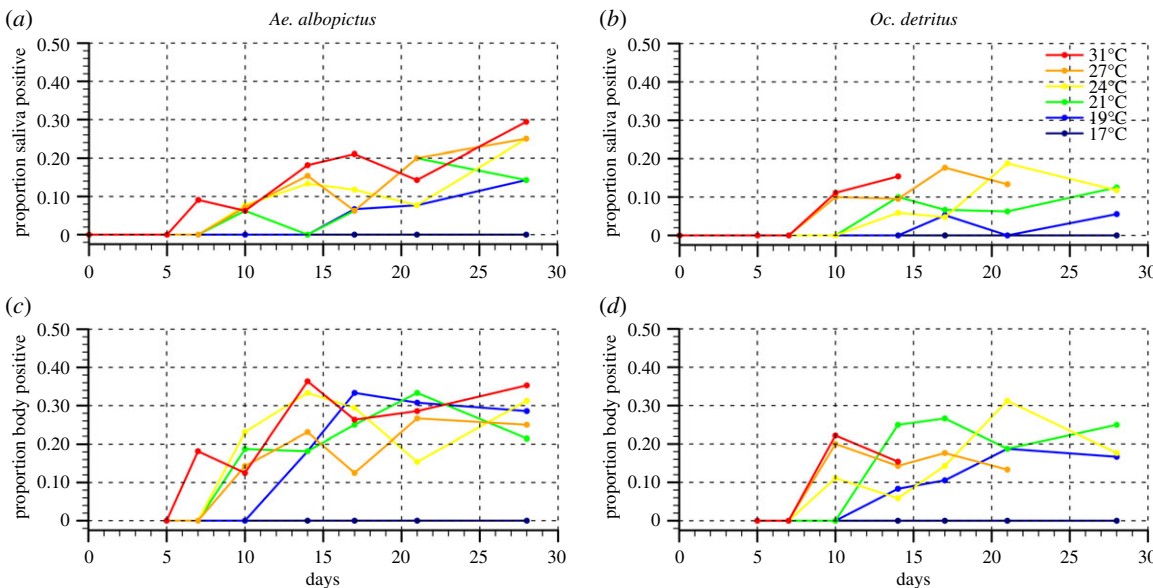

**Figure 1.** Infection of *Ae. albopictus* and *Oc. detritus* saliva and bodies. The proportion of total *Ae. albopictus* and *Oc. detritus* saliva (*a,b*) and bodies (*c,d*) positive for ZIKV at six temperatures (17°C, 19°C, 21°C, 24°C, 27°C and 31°C) and eight time points (0, 5, 7, 10, 14, 17, 21 and 28 days post-infection). Points have been horizontally jittered to reduce overlay. The number of mosquito individuals at each temperature is shown in electronic supplementary material, table S3. (Online version in colour.)

extraction, cDNA generation and qRT-PCR as per methods described here. Live virus titre was calculated from PFU ml$^{-1}$ of stock virus, briefly, Vero E6 cells were infected with serial dilutions of ZIKV and incubated under an overlay of DMEM with 2% FCS and 0.6% Avicel (FMC BioPolymer) at 37°C for 5–7 days. Cells were fixed with 4% formaldehyde then stained with Giemsa for visualization. The limit of detection is described as being $10^{0.6}$ PFU, the lowest value where all three triplicates were positive.

To demonstrate that ZIKV is infectious after incubation at 19°C, an additional 30 mosquitoes were fed with a ZIKV containing blood meal using the same methods described earlier. After the blood meal, the mosquitoes were incubated at 19°C for 21 days, before their saliva was extracted. Cytopathic effect (CPE) assays were used to test saliva samples collected for infectious virus. Salivary collections were pooled (into 10 groups of three) and added to Vero E6 cells maintained in DMEM with 2% FBS and 0.05 mg ml$^{-1}$ gentamycin at 37°C and 5% $CO_2$. Cells were monitored daily for CPE for 7 days.

### (e) Model construction and assumptions

First, we used a logistic regression model to estimate the EIP10, e.g. the EIP measured as the time until 10% of infected mosquitoes become infectious for a given temperature, based on our experimental data available at 17°C, 19°C, 24°C, 27°C and 31°C. See section 'Models of extrinsic incubation period (EIP10)' in the electronic supplementary material for further details.

Second, we include this newly obtained EIP10(T°) scheme for ZIKV and *Ae. albopictus* into a basic reproduction number $R_0$ model for ZIKV (one vector—one host model). Given recently published evidence, the model has been slightly modified.

$R_0(T)$ is given by

$$R_0(T) = \left( \frac{b\beta a(T)^2}{\mu(T)} \right) \left( \frac{\nu(T)}{\nu(T) + \mu(T)} \right) \left( \frac{m}{r} \right).$$

The biting (*a*), mortality (*μ*) and extrinsic incubation (*ν* = 1/ EIP10(T°)) rates depend dynamically on temperature. All other epidemiological parameters are fixed to a constant value (electronic supplementary material, table S1). A rainfall criterion is applied to mask desertic regions following [22]. The final $R_0(T)$ estimate is standardized to range between 0 and 1, to be

consistent with previously published models and make our results directly comparable with such studies [23]. We also calculate the potential length of the ZIKV transmission season (LTS in months) based on standardized $R_0(T)$. For further details about the standardized $R_0(T)$ model and LTS, see sections 'Derivation of $R_0(T)$ for ZIKV—*Ae. albopictus* model' and 'Derivation of the length of the ZIKV transmission season (LTS)—*Ae. albopictus* model' in the electronic supplementary material.

Finally, we integrate the $R_0(T)$ model with observed gridded climate data for the recent period and with an ensemble of climate change projections for the future to investigate regions at risk of ZIKV transmission by *Ae. albopictus*. See section 'Observed climate datasets and climate change scenarios input data' in the electronic supplementary material for further details.

## 3. Results

Zika RNA was detected in the saliva of both *Ae. albopictus* (Verano colony) and *Oc. detritus* at all temperatures from 19°C to 31°C, but not at 17°C (figure 1). The presence of ZIKV in the saliva of *Ae. albopictus* and *Oc. detritus* was recorded from or before day 17, with the earliest time point being day 7 at 31°C for *Ae. albopictus* (figure 1*a*) and day 10 at 27°C and 31°C for *Oc. detritus* (figure 1*b*). The titre of ZIKV in the saliva of *Ae. albopictus* is 3.8× higher than in the saliva of *Oc. detritus* ($p < 0.00001$) across all saliva-positive individuals (electronic supplementary material, tables S2). The bodies of all females were positive at day 0 (immediately after feeding) demonstrating that virus was ingested. On day 5, all bodies were negative for virus demonstrating the eclipse phase of virus infection with undetectably low titres. From day 7 onwards virus was detected in the bodies of both species (for numbers of mosquito bodies and saliva tested at each temperature and time point see electronic supplementary material, table S3). Mortality of *Ae. albopictus* and *Oc. detritus* was similar at all temperatures up to 24°C; above this, the mortality rate of the more temperate *Oc. detritus* was significantly higher (electronic supplementary material, figure S1).

**Table 1.** Estimated time to 10% infected ($EIP_{10}$) of *Ae. albopictus* and *Oc. detritus*. Estimated $EIP_{10}$ and 95% confidence intervals are based on a logistic regression model, Model 1 (for details about 'Model 1' its construction and alternatives considered please see section 'Models of extrinsic incubation period (EIP10)' in the electronic supplementary material).

| temperature (°C) | *Ae. albopictus* | | | *Oc. detritus* | | |
|---|---|---|---|---|---|---|
| | estimate (days) | lower 95% CI | upper 95% CI | estimate (days) | lower 95% CI | upper 95% CI |
| 19 | 22.6 | 18.3 | 29.1 | 25.6 | 20.8 | 33.7 |
| 21 | 20.4 | 17.0 | 25.7 | 23.4 | 19.4 | 30.4 |
| 24 | 17.2 | 14.6 | 21.0 | 20.2 | 16.8 | 25.9 |
| 27 | 14.0 | 11.1 | 17.5 | 17.0 | 13.4 | 22.3 |
| 31 | 9.7 | 4.7 | 14.4 | 12.7 | 7.5 | 18.6 |

The negative competence results for smaller numbers of tested *Cx. pipiens pipiens* and *Cs. annulata* can be found in the electronic supplementary material, table S4. Due to small numbers they are not discussed further in the main text.

In order to demonstrate that ZIKV in the saliva of the mosquitoes incubated at the lowest positive temperature (19°C) is viable, we tested the ability of virus extracted from saliva to infect VERO cells via a cytopathic effect (CPE) assay. Saliva from 30 $G_{11}$ *Ae. albopictus* (Verano colony) females was extracted 21 days after ZIKV challenge and were pooled into 10 lots of three samples. Four of these 10 pools were positive for viable virus, indicating at least 13% of saliva samples were positive (consistent with our results for *Ae. albopictus* at day 21 and 19°C; see figure 1a). This confirmed that our minimum temperature for demonstrable transmission does produce infectious virus and hence does present a transmission risk. Pictures of these cells can be found in the supplementary materials (see electronic supplementary material, table S5 and figure S2).

Using our experimental data, table 1 gives maximum-likelihood estimates and associated 95% confidence intervals for $EIP_{10}$, e.g. the time in days to reach 10% saliva-positive for *Ae. albopictus* and *Oc. detritus*.

Model 1, which represents the estimated proportion of mosquitoes susceptible to saliva infection, includes a linear effect of temperature and separate estimates of infection plateaux ($\delta$s) for the two species. Model 1 was identified as the preferred model (see electronic supplementary material, text, tables S1 and S6, and figure for further details about model design, selection and validation).

The employed logistic regression model (Model 1) identified a negative linear relationship between the duration of the EIP and temperature for both *Ae. albopictus* and *Oc. detritus* within the temperature ranges used in this study (electronic supplementary material, figure S3b). Additionally, the plateau value representing the proportion of mosquitoes susceptible to saliva infection ($\delta$s) for *Ae. albopictus* (0.27) was significantly greater than that of *Oc. detritus* (0.20) (electronic supplementary material, table S7 and figure S4), indicating that *Ae. albopictus* is more competent for ZIKV at all temperatures. The resulting standardized $R_0(T)$ function peaks around 29.2°C (electronic supplementary material, figure S3d). Our standardized $R_0(T)$ peak estimate for *Ae. albopictus* potential to transmit ZIKV (electronic supplementary material, figure S3d, solid red line) occurs at slightly higher temperature than highlighted previously for *Ae. aegypti* [24] and is very

similar to a published estimate [25] (electronic supplementary material, figure S3d, solid black line). Note that if we fitted an exponential scheme instead of a linear scheme for our EIP(T) estimates, this does not significantly impact the shape of the resulting $R_0(T)$ function as mortality (U-shape curve, electronic supplementary material, figure S3c) tends to determine the upper and lower suitability boundaries for standardized $R_0(T)$ (see electronic supplementary material, figure S5). Based on our laboratory experiments, the minimum temperature threshold for ZIKV transmission lies between 17°C and 19°C (figure 1a); this temperature range corresponds to standardized $R_0(T)$ ranging between 0.201 and 0.295; these values are highlighted in beige on figure 2 and electronic supplementary material, figure S3d. Above 19°C, we know that ZIKV transmission by *Ae. albopictus* occurs in the laboratory (figure 1a); these values are highlighted in orange, red and dark red on figure 2 and electronic supplementary material, figure S3d. In the following, we focus on regions where annual standardized $R_0(T)$ exceeds these critical thresholds, using the same colour code and we also discuss potential changes in the simulated length of the ZIKV transmission season (LTS thereafter) at global scale.

Annual standardized $R_0(T)$ maps reveal that ZIKV transmission risk is largest over the tropics but a moderate risk also extends into temperate regions of North America, Europe and Asia (figure 2a). Simulated standardized $R_0(T)$ values are largest over South America (figure 2c), central Africa and the coasts of Africa (figure 2b) and tropical Asia (figure 2f). Additionally, ZIKV could theoretically be transmitted all year long by *Ae. albopictus* in the tropics (South America, Africa and Asia; see electronic supplementary material, figure S6). While in temperate regions, the potential length of the transmission season (LTS) is shorter and varies between one and six months (electronic supplementary material, figure S6).

In Africa, highland regions (plateaux of Ethiopia, western Kenya, central Tanzania, plateaux of Angola and Madagascar) are simulated to be risk-free based on recent climatic conditions (figure 2b). ZIKV could be transmitted all year round in Central Africa (electronic supplementary material, figure S6b). The LTS decreases as a function of the latitude over Africa, with one to three months LTS simulated over the northern fringe of the Sahel and over temperate regions of southern Africa (electronic supplementary material, figure S6b). Standardized $R_0(T)$ barely exceeds the 0.2–0.3 threshold (which corresponds to potential ZIKV transmission

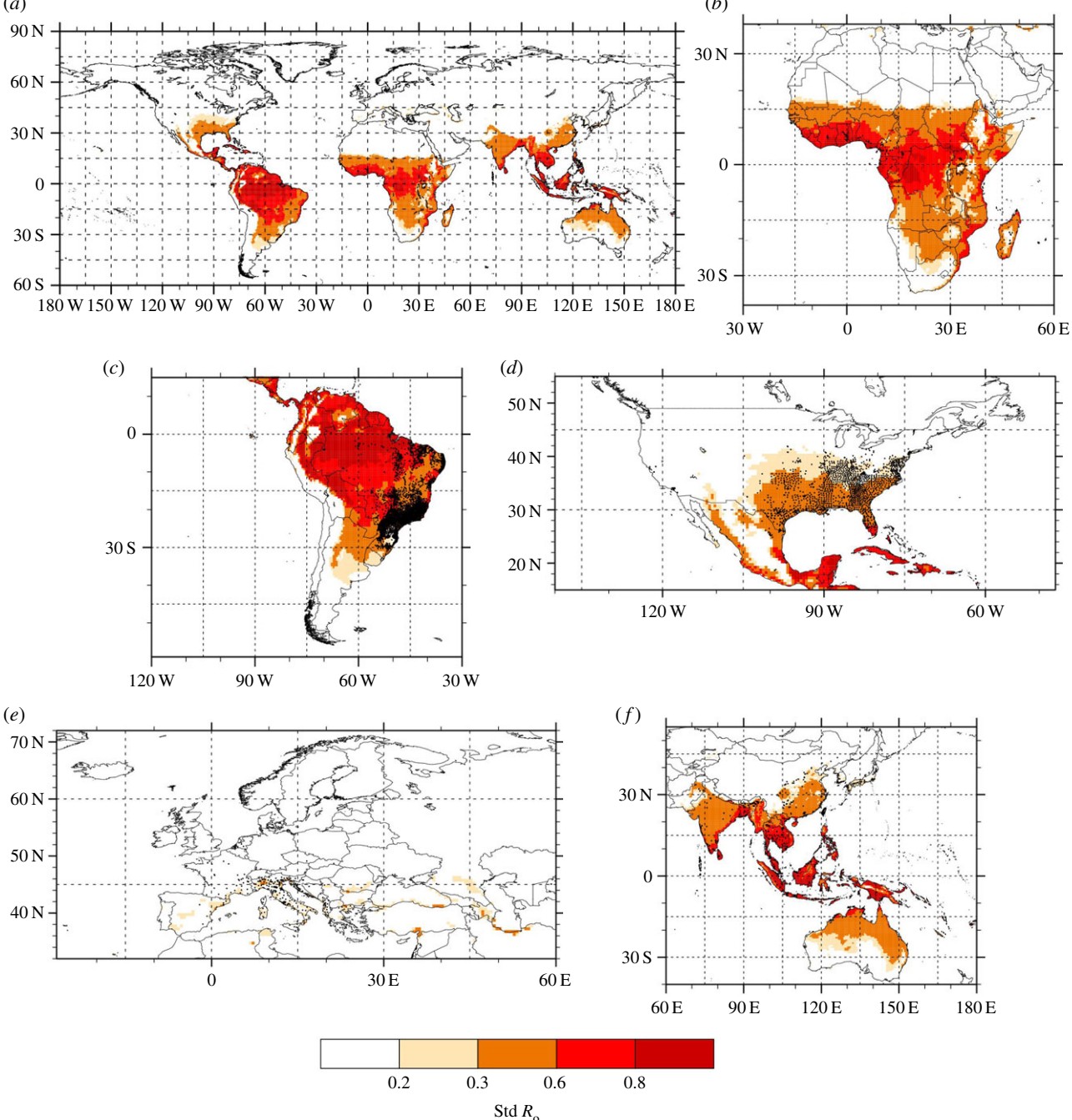

**Figure 2.** Annual mean standardized $R_0(T)$ estimates for *Ae. albopictus* potential to transmit ZIKV. This is carried out for the historical context (1980–2010 average) based on observed rainfall and temperature data for (*a*) the globe, (*b*) Africa, (*c*) South America, (*d*) North America, (*e*) Europe and (*f*) Asia. Black dots represent observed presence of *Ae. albopictus* based on [17]. Beige highlights standardized $R_0(T)$ values for which some ZIKV transmission by *Ae. albopictus* might occur in the laboratory (17–19℃); orange, red and dark red colours correspond to standardized $R_0(T)$ values for which ZIKV transmission by *Ae. albopictus* does occur in our infection experiments (above 19℃). (Online version in colour.)

in the laboratory, e.g. 17–19°C) on an annual basis over coastal Mediterranean regions, with potential hotspots over the eastern coasts of Spain, Sardinia, Sicily, northern Italy, southern France, the coasts of Croatia and Albania, Greece and, in Turkey, near Istanbul and at the southern coastal border with Syria (figure 2*e*). *Aedes albopictus* is already well established around the Mediterranean and the Adriatic [26]; however, the risk estimated by the annual standardized $R_0$ is relatively small in magnitude over southern Europe, suggesting a low probability of a large Zika outbreak.

In the USA, a large region where *Ae. albopictus* is already established could theoretically sustain ZIKV transmission during summer (figure 2*d*). The risk is largest over the southern states, in particular over the southern tip of Florida and Texas where autochthonous transmission of ZIKV was reported in 2016–2017 [27–29] and where ZIKV transmission could occur between seven to 12 months (electronic supplementary material, figure S6d). Across North America the largest standardized $R_0$ values are simulated over Cuba, the Dominican Republic, Puerto Rico and the Caribbean where active ZIKV circulation (very likely by another mosquito vector, *Ae. aegypti*) was reported in 2016–2017 and where year-round ZIKV transmission by *Ae. albopictus* is theoretically possible (electronic supplementary material, figure S6d).

Over temperate Asia, conditions over a large region covering southern China, southern Japan, and Taiwan could

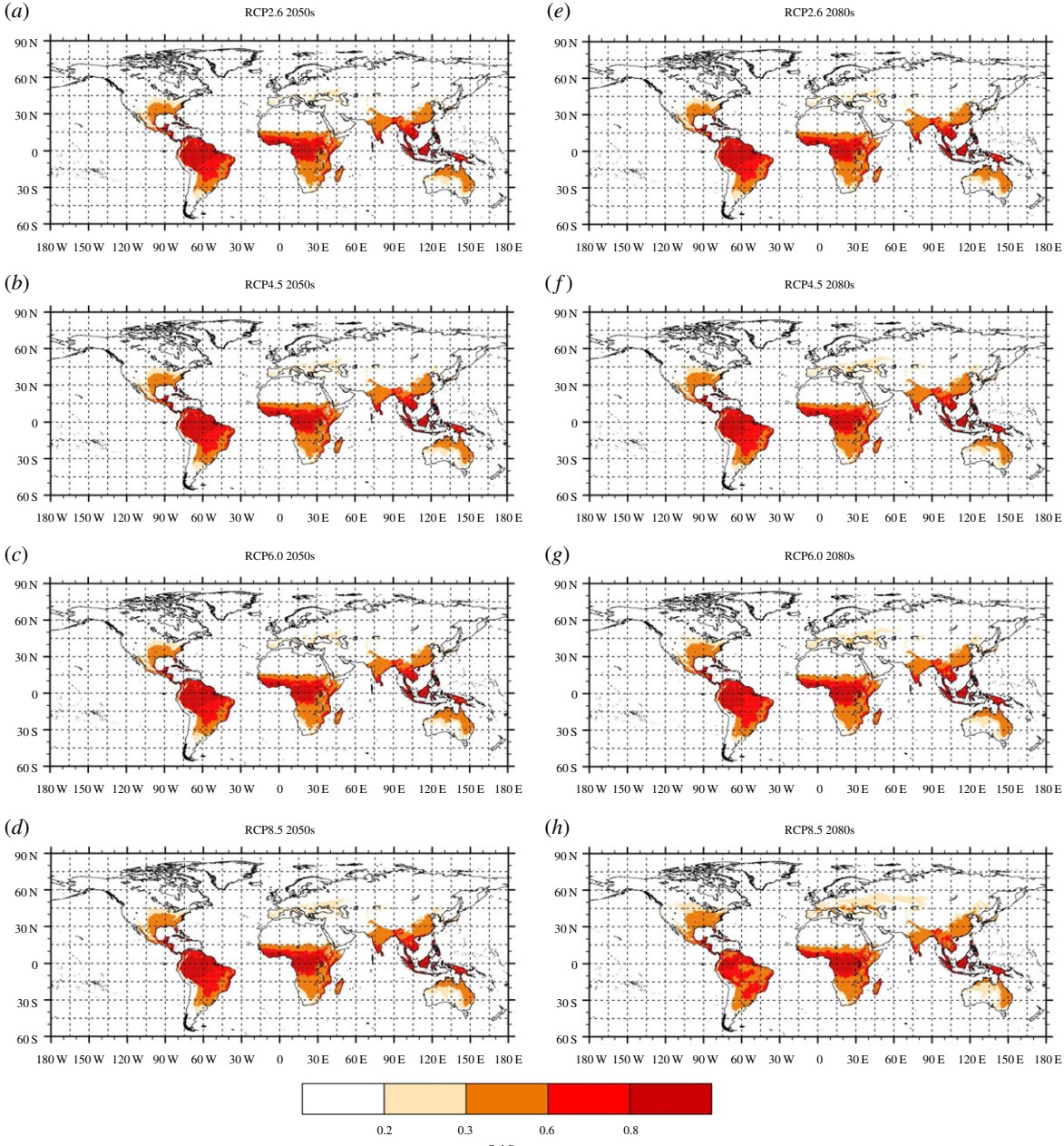

**Figure 3.** Annual mean standardized $R_0(T)$—future projections for *Ae. albopictus* potential to transmit ZIKV. This is carried out for the 2050s (2040–2059 average), left column (*a–d*) and the 2080s (2070–2089 average), right column (*e–h*), from the lowest (RCP2.6, top, *a,e*) to the highest (RCP8.5, bottom, *d,h*) emission scenario. Beige highlights standardized $R_0(T)$ values for which some ZIKV transmission by *Ae. albopictus* might occur in the laboratory (17–19°C); orange, red and dark red colours correspond to standardized $R_0(T)$ values for which ZIKV transmission by *Ae. albopictus* occurs in our infection experiments (above 19°C, figure 1*a*). (Online version in colour.)

sustain ZIKV transmission by *Ae. albopictus* (figure 2*f*). In particular, the largest standardized $R_0$ value is shown over the southern Guangdong, Guangxi and Hainan provinces, over Taiwan and the most southerly Japanese islands where observed autochthonous transmission of dengue was recently reported [30,31] and where simulated LTS exceeds six months (electronic supplementary material, figure S6*f*).

It is noteworthy that the vector, *Ae. albopictus,* covers a larger geographical domain than its potential ZIKV transmission zone in temperate Europe, northern America and Asia (by comparing vector presence in black dots and standardized $R_0$ in colour code on figure 2). If we remove the rainfall effect in our $R_0$ model (i.e. if we solely focus on temperature effects), the surface at risk increases significantly and to some extent unrealistically, with large values simulated in desert regions such as the Sahara (electronic supplementary

material, figure S7*b*), parts of the Middle East (electronic supplementary material, figure S7*a*), the Xinjiang Uygur Autonomous Region which comprises mountains and deserts (electronic supplementary material, figure S7*f*), and central Australia (electronic supplementary material, figure S7*f*). Risk also increases in Spain, western Turkey (electronic supplementary material, figure S7*e*), California and Arizona (electronic supplementary material, figure S7*d*).

Considering climate change scenarios (figure 3), the surface at risk and magnitude of standardized $R_0$ increase with the severity of the emission scenario (from RCP2.6 to RCP8.5) and with time (from the 2050s to the 2080s). The largest changes are simulated under the RCP8.5 emission scenario for the 2080s (figure 3*h*). Standardized $R_0$ substantially increases over the Mediterranean and Black sea regions (figure 3). If we consider the highest emission

scenario for the 2080s, a large increase in standardized $R_0$ is simulated over the south of France, the eastern coast of Spain, Italy, the Adriatic coasts, and parts of Hungary, Serbia, Croatia, Moldova, Romania and Bulgaria (figure 3h). The ZIKV transmission season could extend to six to seven months in future over a large area covering the south of France, Italy, the Balkans and regions surrounding the Black Sea (electronic supplementary material, figure S8). Interestingly, no ZIKV transmission could occur over the UK based on recent climate conditions (electronic supplementary material, figure S6e), but short seasonal transmission (one to three months) might occur over southern UK during the 2080s based on the highest emission scenario (electronic supplementary material, figure S8 h). ZIKV transmission could also extend farther north in the USA, such as into Nebraska, West Virginia and Pennsylvania (figure 3; electronic supplementary material, figure S9). In temperate Asia, the southeastern half of China and Yunnan province, Taiwan, southern Japan, and to a lesser extent the coasts of South Korea could be at risk of ZIKV transmission by temperate mosquitoes by the 2080s (figure 3; electronic supplementary material, figure S10). LTS could range between five and six months over temperate regions in Asia by the 2080s (electronic supplementary material, figure S10 h).

While *Ae. albopictus* is not believed to be the primary vector of ZIKV in Africa or South America, there is a clear increase in simulated standardized $R_0$ over high altitude regions (plateaux of Ethiopia, western Kenya, plateaux of Madagascar, see figure 3; electronic supplementary material, figure S11). A large part of South America is still simulated to be climatically suitable for the transmission of ZIKV in future (figure 3; electronic supplementary material, figure S12). However, there is a decrease in simulated $R_0(T)$ (figure 3) and a shortening of the ZIKV transmission season over Brazil in future for the 2080s (electronic supplementary material, figure S12 h), due to high temperatures.

## 4. Discussion

In this study, we assessed the competence of common species from both wild populations of *Oc. detritus* (UK), and newly colonized *Ae. albopictus* (Verano colony, Italy) and found that they are competent to transmit viable ZIKV at temperatures as low as 19°C. We also presented the most comprehensive analysis of the effect of temperature on ZIKV incubation to date, establishing that the minimum temperature for ZIKV transmission is between 17°C and 19°C in two different competent temperate vectors, and provide estimates of the time required for 10% to become infectious (as measured by detection of virus in saliva in infected females). Using these estimates, we modelled risk across temperate regions from *Ae. albopictus*, both in current climatic conditions and in future projected conditions.

We used the recently isolated PE243 strain of ZIKV from Brazil, and upper estimates of field-representative titres of ZIKV for mosquito infections [32]. This integration of laboratory studies on field-derived or recently field-derived mosquitoes with modelling makes this study highly applicable to the ongoing risk of ZIKV spread.

Our data show that *Ae. albopictus* is a more competent laboratory vector than *Oc. detritus*, in terms of the proportion of infectious individuals and the titre of the virus in the saliva,

being significantly higher in *Ae. albopictus*. However, there is a considerable proportion of infectious *Oc. detritus*, suggesting that there may be other temperate species which are also highly competent for ZIKV and present a transmission risk (for a detailed list of competent vectors for ZIKV, excluding *Oc. detritus*, see [10]).

Our finding of a minimum temperature for transmission of ZIKV in both *Ae. albopictus* and *Oc. detritus* of between 17°C and 19°C is supported by the findings of Heitmann *et al.* [33] showing ZIKV transmission in *Ae. albopictus* at 27°C but none at 18°C after 21 days. Interestingly, this minimum temperature is substantially higher than the minimum recorded temperature for transmission of the closely related WNV at 14°C [13], indicating that ZIKV may not present as much risk to temperate regions as WNV. ZIKV appears to develop at lower temperatures than DENV, for which most studies have shown no or excessively slow viral development at temperatures lower than 20°C [34], indicating that ZIKV presents a greater risk to temperate regions. Previous studies of ZIKV competence by *Ae. albopictus* have almost exclusively used only 'standard' tropical insectary temperatures of 27 ± 1°C or with a single additional lower temperature [33]. To our knowledge, no other study has examined ZIKV development in a temperate-adapted vector at multiple temperatures or investigated the effect of temperatures as low as 17°C, giving our work unique applicability to temperate regions. Indeed, it is widely believed that climate change may lead to the spread of many vectors and vector-borne diseases to more extreme latitudes and to higher altitudes [23], with strong evidence that in some cases this has already happened, driven by the changing climate [35,36]. Predicting extensions of vector-borne disease distributions into cooler, more temperate regions requires studies at these low temperatures, rather than extrapolation from higher temperature studies, so that the lower temperature threshold can be determined with an acceptable accuracy. Disease model parameterization often relies on extrapolation and our investigation of mosquito infections at low temperatures allows us to better estimate transmission potential, a critical factor to understanding the impact of climate change on vector-borne diseases [15].

Other studies have investigated the importance of wild-caught temperate vectors. Gendernalik *et al.* [37] demonstrated 80% competence in a common worldwide vector *Ae. vexans* (caught in the USA) incubated at 27°C. Studies on wild-caught *Ae. japonicus* highlight the importance of assessing the EIP at temperate-realistic temperatures, as Jansen *et al.*, 2018 [38] showed competence at 27°C but very little at 21°C. Combined, these studies show significant potential risk from temperate vectors, but highlight the need to assess transmission at lower temperatures to determine realistic risk in these regions.

Our $R_0$ risk maps indicate a significant current risk over large areas of southeast USA. These areas spread considerably further than the limit of where autochthonous transmission of ZIKV has occurred to date, which has been restricted almost exclusively to Florida and, to a lesser extent, Texas [27,28]. *Aedes aegypti* is thought to be the main ZIKV vector in these regions, however, our results suggest that *Ae. albopictus* could theoretically sustain ZIKV transmission in these states. Furthermore, our model's prediction of large risk of *Ae. albopictus* transmission of ZIKV over southern provinces of China, southern Japan, and Taiwan is supported by autochthonous

transmission of dengue virus. These data are reported on a yearly basis for the past 25 years over Guangdong, Guangxi and Hunan provinces, with the largest outbreak reported in 2014 [39], and are consistent with former published estimates [25]. This strongly supports concerns about the risk posed by travellers coming from ZIKV endemic countries to these southern provinces [40]. In contrast to the maps of USA and temperate Asia, our risk maps shows that despite *Ae. albopictus* presence in large areas of Europe [41], the recent climatic conditions do not result in large $R_0$ values. This suggests that the range of this important vector extends beyond the minimum temperature limit of ZIKV and consequently the presence of vector alone is not a good indicator of present day risk of sustained Zika transmission. However, seasonal transmission of ZIKV by *Ae. albopictus* is simulated over southern Europe. This finding is consistent with recently reported autochthonous cases of Zika in Hyères, near to Marseille in southeastern France [42]. Furthermore, our simulated hotspots match areas where recent minor autochthonous transmission of dengue or chikungunya virus by *Ae. albopictus* have been observed in Europe. Examples include: the 2007 chikungunya outbreak in Ravenna, Italy; suspected dengue fever in Cadiz, southern Spain in 2018; and the reported cases of dengue in southeastern France in 2010, 2013, 2014 and 2015 [43].

We show that ZIKV transmission risk could increase significantly in the future over southern and eastern Europe, extend farther north into the USA and into temperate regions of Asia; and decrease over South America and West Africa, if we consider the highest emission scenario for the 2080s. ZIKV transmission by *Ae. albopictus* could increase over high-altitude regions; this finding is similar to expected impacts of climate change on malaria [44]. Our results are also consistent with other studies that show likely trends in future [23,45]; however our modelling framework also allows us to explore the impact of rainfall changes on simulated $R_0$. *Aedes albopictus* can flourish in urban settings, using man-made containers to lay its eggs. Consequently, rainfall might not be a limiting factor in densely populated areas or on irrigated lands.

We acknowledge certain limitations in our experimental design, as follows.

(1) Our study focuses on the risk in temperate regions and hence does not account for transmission by the primary ZIKV vector *Ae. aegypti*, which is important in transmission in some of the warmer areas of regions we investigated. We don't consider this a limitation *per se*, but rather a focus on different regions.

(2) Our analysis is based solely on a temperate strain of *Ae. albopictus*; our finding that *Oc. detritus* is also a highly competent temperate vector, as well as other published data showing other competent vectors (reviewed in [10]), suggests that there may be a risk of transmission in temperate regions outside of the range of *Ae. albopictus*. For example, *Oc. detritus* larvae mature in brackish water, often in estuaries or expansive wetland areas [46,47] that are very different to the small volume containers, tyres or natural pools that are preferred by *Ae. albopictus* and *Ae. aegypti*. Locally, they can both achieve very high densities and are voracious human feeders; such species could present a major additional vector control challenge in the event of a ZIKV outbreak.

(3) Our experimental work focuses on fixed temperature incubation, even though diurnal temperature variability has been shown to be important for many of the parameters in the $R_0$ estimation [48,49]. The breadth of different temperature cycling regimens that would need to be tested make such work unfeasible in a study of this scale. Further studies isolating a single average temperature and testing multiple temperature regimens would be needed to assess this effect fully.

(4) Our work demonstrates that the minimum temperature for transmission of viable ZIKV is between 17°C and 19°C. Future experiments, including EIP and virus infectivity at these temperatures, are planned to narrow this range at low temperatures and therefore produce more accurate estimations of the absolute limit of risk.

Our estimates of the potential length of the ZIKV transmission season by *Ae. albopictus* match recently published findings. In our model, seasonal ZIKV transmission could occur during four to five months over southern Europe; this finding is consistent with another $R_0$ model estimate (about four months transmission shown in [45]). The pattern and magnitude of the simulated length of the ZIKV transmission season over northern and southern America is also consistent with [16,23], who also showed potential year-round ZIKV transmission over Central America, the Caribbean and the northern half of South America. Simulated LTS (about seven to eight months) over the southern provinces of China (Guangdong and Guangxi) also corresponds to observed seasonal transmission of dengue fever over these regions [50,51]. Our model only considers the impact of climate on the mosquito vector potential to transmit ZIKV. Such a model cannot account for other important factors including the number of infected travellers arriving, and the immunological history of the local population [52,53]. Herd immunity is an important factor in risk and is expected to be low in naive European, North American and temperate Asian populations. Conversely, antibody-dependent enhancement (ADE) of infection, most commonly associated with DENV, may enhance ZIKV viremia and therefore potentially increase the likelihood of mosquito acquisition of the virus [54]. Such ADE may have been a significant driving force in non-flavivirus-naive populations, but is likely to be reduced in temperate populations. The interplay between climatic and other external factors will likely have a large effect on the risk to a population.

Using field applicable experimental data, we present a comprehensive analysis of the relationship between ZIKV risk and temperature. Our results show that the risk of autochthonous ZIKV transmission extends beyond the regions in which it has been reported thus far, especially in Asia and the USA. They also show in Europe that vectors can be present in regions that are too cold for sustained ZIKV transmission. The comprehensive analysis of EIP and minimum temperature requirements presented here offers a route to enable informed risk management and outbreak preparedness in more specific temperate situations.

Data accessibility. All data and code used in analyses: 10.6084/m9.figshare.12465116

Authors' contributions. Wrote the paper: M.S.C.B., C.C. and M.B. Performed the work: M.S.C.B., C.C., E.I.P., K.S., G.E.C., J.H., S.M. and M.B. Performed analysis: M.S.C.B., C.C., E.I.P., P.J.D., S.M. and M.B. Experimental design: M.S.C.B., P.J.M., G.L., J.M. and M.B.

Provided materials: G.L.H. and A.d.T. Conceived work: M.S.C.B., C.C. and M.B.

Competing interests. The authors declare that there are no competing interests.

Funding. Experimental work was funded by an MRC Zika Rapid Response award to M.B. (MC_PC_15090). M.S.C.B., C.C., S.M. and M.B. acknowledge funding from the NIHR. The research was funded by the National Institute for Health Research Health Protection Research Unit (NIHR HPRU) in Emerging and Zoonotic Infections at the University of Liverpool in partnership with Public Health England (PHE) and Liverpool School of Tropical Medicine (LSTM). The views expressed are those of the author(s) and not necessarily those of the NHS, the NIHR, the Department of Health or Public Health England.

Acknowledgements. The authors thank Alain Kohl for providing the Zika virus, and Beniamino Caputo for establishing and maintaining *Aedes albopictus* albo-VERANO colony in Sapienza University. We also thank Tom Solomon and Steve Torr for their useful comments on the manuscript.

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
