## [Reviewer comments · Proceedings of the Royal Society B: Biological Sciences]

Review History

RSPB-2019-1973.R0 (Original submission)

Review form: Reviewer 1

Recommendation

Major revision is needed (please make suggestions in comments)

Scientific importance: Is the manuscript an original and important contribution to its field?

Good

General interest: Is the paper of sufficient general interest?

Good

Quality of the paper: Is the overall quality of the paper suitable?

Good

Is the length of the paper justified?

Yes

Should the paper be seen by a specialist statistical reviewer?

Yes

Do you have any concerns about statistical analyses in this paper? If so, please specify them explicitly in your report.

No

It is a condition of publication that authors make their supporting data, code and materials available - either as supplementary material or hosted in an external repository. Please rate, if applicable, the supporting data on the following criteria.

Is it accessible?

Yes

Is it clear?

Yes

Is it adequate?

Yes

Do you have any ethical concerns with this paper?

No

Comments to the Author

Summary: Blackgrove et al. determined the vector competence of 4 species of mosquitoes from Europe for ZIKV from Brazil at one virus dose. Two species [*Cx. pipiens pipiens* and *Culiseta anulata*] did not become infected. Two *Aedes* were susceptible and the duration of the extrinsic incubation period [EIP = time from exposure to transmission] was measured at 6 temperatures and 7 time points. The primary vector of ZIKV globally [*Ae. aegypti*] was not included in this paper. No transmission was detected at 17C supporting their calculated minimum EIP temperature threshold. Based on these data and a series of modelling assumptions, R_0 was modeled for different temperatures and used to map the distribution of risk of ZIKV transmission globally.

Thoughts: Specific comments have been added directly to the attached files and are summarized below.

1. Proportion transmitting. This estimate was based on the number of females that imbibed a blood meal and survived to be tested rather than the number infected. I wonder if it is reasonable to estimate the time to transmission based, in part, on uninfected females? To accommodate these data, the EIP calculations were based on 10% with salivary secretions positive at each time point, even though <10 females were used at some time points [Suppl Table 5]? Data presented in Fig. 1 indicate that <40% of the females tested were ever positive for body infection.

Examining the raw data, the authors may have biased their results by using fewer females at earlier time points and greater numbers at later time points when transmission was anticipated.

2. R_0 assumptions [Suppl Table 3].

a(T) assumed it takes twice as long for *albopictus* to digest a blood meal and refeed as *aegypti*? Does this assume gonotrophic concordance?

b was rescaled between 0 and 1 rather than use the values estimated?

Beta was also rescaled rather than use vector infection rates. Here a value of 1 would exceed the values in their experiments?

m and to some extent b failed to consider the probability of human blood feeding which can be quite low when other hosts are plentiful? In some areas, for example, *albopictus* is considered a major vector of dog heartworm.

3. R_0 estimates. These remained <1 for most areas which would indicate the virus should not

persist at the annual temperatures used? Would it be of interest to use min and max temperatures for selected geographical areas? Historically DENV and YFV caused major summer outbreaks at relatively northern latitudes, but did not persist.

Review form: Reviewer 2

Recommendation

Major revision is needed (please make suggestions in comments)

Scientific importance: Is the manuscript an original and important contribution to its field?

Good

General interest: Is the paper of sufficient general interest?

Good

Quality of the paper: Is the overall quality of the paper suitable?

Marginal

Is the length of the paper justified?

Yes

Should the paper be seen by a specialist statistical reviewer?

Yes

Do you have any concerns about statistical analyses in this paper? If so, please specify them explicitly in your report.

Yes

It is a condition of publication that authors make their supporting data, code and materials available - either as supplementary material or hosted in an external repository. Please rate, if applicable, the supporting data on the following criteria.

Is it accessible?

Yes

Is it clear?

N/A

Is it adequate?

N/A

Do you have any ethical concerns with this paper?

No

Comments to the Author

The subject of the manuscript, examining the possible transmission of Zika virus in temperate regions is an important issue. The most important finding in the study, in my opinion, is the finding of *Ochlerotatus detritus* being able to transmit Zika virus. As the authors state, this mosquito breeds in natural water sources which makes the vector control far more harder than with mosquitoes (e.g. *Aedes albopictus*) that breed in artificial containers. In addition, notification that ZIKV replicates in mosquitoes in temperate temperatures is important. Yet, there are some issues that need to be addressed in my opinion. The authors state that the minimum transmission temperature for ZIKV is 17-19°C, though the scale of 17-19°C is quite wide when we

are talking about the climate and this should be discussed or even additional experiments. There also are other critical issues that should be addressed in my opinion. First, there is no data of the copy numbers for ZIKV in the mosquito saliva. Is there difference in the viral loads in detritus and albopictus saliva? This data would support the conclusion of *Ae. albopictus* being more competent vector than *Oc. detritus* and I think authors should provide this data and discuss the differences. What is the infectious dose in the saliva? I also would recommend discussion of seasonal transmission. There are large areas, where the continuous transmission is very unlikely in future decades but when suitable vector is present, the temperatures might allow seasonal transmission, which is not considered at all in the manuscript. I suggest downgrading the conclusion of assessing the vector competence of four mosquito species as two of them were applicable in such a small number that comprehensive experiments could not be conducted. In the experiments it was found that in lower temperatures it took longer for the virus to be detected from the saliva. When the timeline from blood meal to positive saliva exceeds 20 days, it would be interesting to discuss the significance in practice, as what is the natural lifespan for these mosquito species in the nature. The spelling of the manuscript needs quite a lot of revision. Both the main text and supplementary material include two different fonts. The terms with capitals, (Southern, Northern, Central etc) should be corrected throughout the text.

Specific comments

Abstract

Line 47 change japan to Japan

Introduction

Lines 52-53 "Though infection is asymptomatic in 80% of cases (1), a small proportion of patients develop the autoimmune condition Guillain-Barré syndrome (2)". This should be rephrased as now it reads as ZIKV infection is either asymptomatic or it manifests as GBS.

Line 56 World Health Organisation, please change to World Health Organization

Line 67 Please clarify what you mean by "pathogen to develop"

Lines 71-73 "With large human populations in temperate climates it is also important to estimate how ZIKV transmission occur in such climates today, and with future climate change." Do authors mean that already occurs or that could occur? Please clarify

Line 73 "future climate change" is probably not the best term as climate change itself occurs already. I suggest to change it to in future climate for example.

Line 84 RCPx, what does the x stand for? Should it be s?

Results

Lines 88-89 "Zika virus was detected in the saliva of both *Ae. albopictus* (VERANO colony) and *Oc. detritus* at all temperatures from 19°C to 31°C, but not at 17°C" Please add copy numbers, this would be nice to see in a table. But it would clarify the effect of temperature on the transmission.

Line 152 change Tropics to tropics

Line 154 change central Africa to Central Africa

Line 157 should three dots after Malaysia be removed?

Line 163 "parts of Turkey" please specify which parts

Lines 168-171 "Examples include Ravenna in the Emilia-Romagna region in Italy where a chikungunya outbreak occurred in 2007; nearby Cadiz in southern Spain and on the eastern Catalan coasts with a few reported and suspected cases of dengue in 2018; south-eastern France where cases of dengue were reported in 2010, 2013, 2014 and 2015 (20)" Please modify this sentence.

Line 184 Correct to temperature only- effect

Line 187 change central Australia to Central Australia

Line 215 "However, there is a decrease in simulated $R_0(T)$ over Brazil in future for the 2080s (Fig. S6h)." some discussion on the reasons behind this would be good add.

Discussion

Line 256 Please clarify what does “high laboratory competence” mean

Line 261 please refer to the study not only to a reference number in the text.

Line 285 “in contrast to USA and temperate Asia” I would suggest referring to “in contrast to the maps of USA and Asia”

Lines 292-293 “Ae. albopictus is not a major issue in Africa yet”, please retype this. This refers to the presence of Ae. albopictus ‘

Line 293 should altitude regions be high altitude regions?

Line 309 “Locally they...” Does this refer to both Ae. albopictus and aegypti or other species?

Please clarify

Line 316 “Our model shows only the risk as a result of climate and vector species” please retype this. For example: as a result of the effect of climate and vector species

Lines 323-324 “The interplay between these, and other, factors will likely have a large effect on the risk to a population.” Please retype.

In general, the validity of the models used should be discussed more thoroughly as now there is very little about it.

In results the authors state that: “If we remove the rainfall effect in our R0 model (temperature only effect), the surface at risk increases significantly and to some extent unrealistically, with large values simulated in desert regions such as the Sahara (Fig. S3b), parts of the Middle East (Fig. S3a), the “ yet there is no discussion about the rainfall model

The limitations of the study should include discussion of the Cx. pipiens pipiens and Cs. annulata experiments as they were not comprehensive compared to other two species. One option would be to exclude data on these as obviously their vector competence must be studied further.

Materials and methods

Line 359 Please specify where the human blood for feeding the mosquitoes was achieved

Is RNA extraction with Trizol the best choice for saliva? In my experience I might suspect losing some RNA from the weak positive samples with this method.

Figures and tables

In all the map figures that are zoomed (so not including the world map), in supplementary material and the main text, the coordinates would be helpful especially in the boundaries of zoomed maps, in order to recognize easier where is e.g. equator (grids to the boundaries of these maps), so that it would be possible and easier to estimate more accurate (latitude and longitude) risk area.

Figure 2 There is no referring to figure 2c in the text

Figure 3 Dots are very small, should be bigger to be able to see them

Figure 4, parts a-h are not explained in the caption. Please add this.

Figure 4 line 233 “the most extreme” quite strong expression, prefer to high or higher emission scenario etc

Figure legends should consistently be placed under the figure and table legends above the table. There is variation in this now. Please correct.

Supplementary material

Lines 76-78 “we employed the Global Precipitation Climatology Centre (GPCC) global rainfall data available at similar spatial and time resolution for the same time period (7).” Exactly same or just similar resolution? Please specify.

Lines 78-79 “We calculated annual average for the 1980-2010 period, as advised by the World Meteorological Office guidelines” Please add a reference for this.

Lines 83-84 “for the ensemble mean of 5 GCM (hadgem2-es, ipsl-cm5a-lr, miroc-esm-chem, gfdl-esm2m, noresm1-m)” Could you explain why did you choose to make ensemble of these 5 GCMs specifically?

Supplementary figure 6 legend “The beige colour depicts standardized R0(T) values for which some ZIKV transmission by Ae. albopictus might occur in the laboratory [17-19°C]”. Please clarify this.

Supplementary table 1. Retype model names: 1. Equation, for example
Supplementary table 5 the caption of the table is above the table as it should be but the legend is placed under the table. Please correct.

Review form: Reviewer 3

Recommendation

Major revision is needed (please make suggestions in comments)

Scientific importance: Is the manuscript an original and important contribution to its field?

Acceptable

General interest: Is the paper of sufficient general interest?

Good

Quality of the paper: Is the overall quality of the paper suitable?

Marginal

Is the length of the paper justified?

Yes

Should the paper be seen by a specialist statistical reviewer?

Yes

Do you have any concerns about statistical analyses in this paper? If so, please specify them explicitly in your report.

No

It is a condition of publication that authors make their supporting data, code and materials available - either as supplementary material or hosted in an external repository. Please rate, if applicable, the supporting data on the following criteria.

Is it accessible?

Yes

Is it clear?

No

Is it adequate?

No

Do you have any ethical concerns with this paper?

No

Comments to the Author

The authors are describing an interesting study of ZIKV transmission, suggesting firstly that *Oc. detritus* may vector ZIKV in temperate regions at temperature 19C- 24C and that *Ae albopictus* could also transmit ZIKV at lower temperatures than previously reported (19, 21, 27C).

The result of *Oc. detritus* transmitting ZIKV is important, and its implications in the regions where this mosquito is present are significant. The authors state this in their discussion and point out the difficulty of vector control to mosquitoes that are breeding in natural water sources.

The main concerns in this study arise from interpretation of the obtained results, and considerations of what these results mean. The findings are interesting, but should be considered preliminary. Further evidence and more detailed information of the transmission temperature (and other) thresholds would be needed before applying the values to modelling and risk evaluations.

The experimental evidence shows that no transmission (virus in the saliva) occurs at temperature of 17C in either of the species tested (*Ae. albopictus* or *Oc. detritus*). Therefore, the statement that the authors have identified a minimum temperature threshold between 17 and 19 C (abstract line 42) is an overestimate. The authors are speculating (without evidence) that the transmission would occur also below 19C although they do not have any data to support this assumption. Further testing should be done to identify the exact lowest limit, whether that is between 17.1-18.9C in the tested mosquito species.

Overall the manuscript structure and clarity should be improved, and the relevant contents should be shown in the manuscript (and not in the supplementary file). There are too many different map images provided, it seems that the authors cannot themselves decide what is relevant and what is not. The authors should provide evidence for their laboratory findings to have implications also to real life. The authors do not show the data of the virus amount in the saliva of *Oc. detritus*, or compare that to those obtained from *Ae. albopictus*. The results should be discussed also in the light of the results obtained with *Ae. aegypti*, that showed the low temperatures not supporting virus transmission to saliva (Tesla B et al., 2018). It would also be important to demonstrate that in the case of *Ae. albopictus* and *Oc. detritus*, at low temperatures, the virus PCR positive saliva would actually be infectious (in cell culture or mosquito infection). And as a control, it would have been good to add *Ae. aegypti* to the tests to rule out e.g. viral strain properties or other possibly affecting factors that are different from the previously published work.

The manuscript should be edited to highlight the actual findings and not the modelling and estimates that are at present highly speculative. If such maps are shown, they should be clearly explained and discussed in the context of other, previously published risk maps. The authors state in the abstract "Our R0-based risk maps show significant risk of ZIKV transmission beyond the current observed range in southern USA, southern China and southern European countries." It may be so that the maps show risk beyond currently observed areas, but the maps look pretty modest in comparison to other predicted risk maps of *Ae. albopictus* transmission, for example: Ryan SJ et al, PLOS N trop Dis 2019. It would be important for the temperate regions to take the seasonality (e.g. summer months transmission) into account and also show these in the maps. The authors are only talking about sustained transmission (in the discussion lines 332-333). The authors should report only results from *Oc. detritus* and *Ae. Albopictus*, as there is comparable data for these two. There is only very limited testing done (not all temperatures/time points) for *Cx. Pipiens* and *Cs. Annulata*. So the statement that four mosquito species were studied (abstract lines 37, 38) is misleading.

One of the concerns of this paper is the small numbers of mosquitoes used in some of the experiments. The numbers of mosquitoes in experiments are only found in the supplementary file. The n-numbers should be presented in the actual manuscript in some form.

The authors should cite the relevant previous work on the topic, such as the work done using *Aedes japonicus* (Jansen et al., 2018) and *Aedes vexans* (Gendernalik et al., 2017). To what extent are the study methodologies comparable?

The authors are discussing their findings in context of *Ae. albopictus*, but very little is said about *Oc. detritus*. For example, the authors should give some overview to the readers about *Oc. detritus* distribution.

Discussion lines 333-335: "The comprehensive analysis of EIP and minimum temperature requirements presented here offer a route to enable informed risk-management and outbreak preparedness in more specific temperate situations." What do the authors mean by this? How is the data presented here applicable to risk management or outbreak preparedness?

Line 385: "As is standard practice with qRT-PCR experiments, a 'cut-off' of 40 Ct was applied to

minimise false positives towards the limit of detection." What do the authors mean by this? The used method is a probe- based assay? I would understand this if it was a SYBR green assay.

Decision letter (RSPB-2019-1973.R0)

23-Oct-2019

Dear Dr Blagrove:

I am writing to inform you that your manuscript RSPB-2019-1973 entitled "Potential for Zika virus transmission by mosquitoes in temperate climates" has, in its current form, been rejected for publication in Proceedings B.

This action has been taken on the advice of the Associate Editor and the referees, who have recommended that substantial revisions are necessary. Ideally these would include additional experiments, as there is clearly an impression that these results are preliminary. With this in mind we would be willing to consider a resubmission, provided the comments of the referees are fully addressed. However please note that this is not a provisional acceptance.

Please also note that one of the referees commented about the data archiving: 'The authors do not report/show one important piece of data concerning the work, which is the viral load in the specimens/samples. It would be relevant to compare the viral loads of ZIKV in the saliva samples of *Oc. detritus* and *Ae. albopictus* when discussing their vector potential or competence.' This needs to be addressed with a revision.

Yours sincerely,
Professor Loeske Kruuk
Editor
mailto: proceedingsb@royalsociety.org

Associate Editor

Board Member: 1

Comments to Author:

Thank you for submitting your manuscript "Potential for Zika virus transmission by mosquitoes in temperate climates" To Proceedings B. I have now received three reviews and evaluated the manuscript myself. While we all find the topic interesting and the manuscript generally written, several important issues have been raised. In particular, important areas seem to be oversimplified and more details and nuance are required. For example, reviewer 1 highlights several assumptions implicit to the R_0 calculations that should be teased apart or at least discussed. Similarly, reviewers 2 and 3 raised several more detailed nuances relevant to ZIKV transmission that need to be addressed, such as copy numbers, and how the study conclusions could be improved by additional experiments.

Reviewer(s)' Comments to Author:

Referee: 1

Comments to the Author(s)

Summary: Blackgrove et al. determined the vector competence of 4 species of mosquitoes from Europe for ZIKV from Brazil at one virus dose. Two species [*Cx. pipiens pipiens* and *Culiseta anulata*] did not become infected. Two *Aedes* were susceptible and the duration of the extrinsic incubation period [EIP = time from exposure to transmission] was measured at 6 temperatures and 7 time points. The primary vector of ZIKV globally [*Ae. aegypti*] was not included in this paper. No transmission was detected at 17C supporting their calculated minimum EIP temperature threshold. Based on these data and a series of modelling assumptions, R_0 was modeled for different temperatures and used to map the distribution of risk of ZIKV transmission globally.

Thoughts: Specific comments have been added directly to the attached files and are summarized below.

1. Proportion transmitting. This estimate was based on the number of females that imbibed a blood meal and survived to be tested rather than the number infected. I wonder if it is reasonable to estimate the time to transmission based, in part, on uninfected females? To accommodate these data, the EIP calculations were based on 10% with salivary secretions positive at each time point, even though <10 females were used at some time points [Suppl Table 5]? Data presented in Fig. 1 indicate that <40% of the females tested were ever positive for body infection. Examining the raw data, the authors may have biased their results by using fewer females at earlier time points and greater numbers at later time points when transmission was anticipated.
2. R_0 assumptions [Suppl Table 3].
 - a(T) assumed it takes twice as long for *albopictus* to digest a blood meal and refeed as *aegypti*? Does this assume gonotrophic concordance?
 - b was rescaled between 0 and 1 rather than use the values estimated?
 - Beta was also rescaled rather than use vector infection rates. Here a value of 1 would exceed the values in their experiments?
 - m and to some extent b failed to consider the probability of human blood feeding which can be quite low when other hosts are plentiful? In some areas, for example, *albopictus* is considered a major vector of dog heartworm.
3. R_0 estimates. These remained <1 for most areas which would indicate the virus should not persist at the annual temperatures used? Would it be of interest to use min and max temperatures for selected geographical areas? Historically DENV and YFV caused major summer outbreaks at relatively northern latitudes, but did not persist.

Referee: 2

Comments to the Author(s)

The subject of the manuscript, examining the possible transmission of Zika virus in temperate regions is an important issue. The most important finding in the study, in my opinion, is the finding of *Ochlerotatus detritus* being able to transmit Zika virus. As the authors state, this mosquito breeds in natural water sources which makes the vector control far more harder than with mosquitoes (e.g. *Aedes albopictus*) that breed in artificial containers. In addition, notification that ZIKV replicates in mosquitoes in temperate temperatures is important. Yet, there are some issues that need to be addressed in my opinion. The authors state that the minimum transmission temperature for ZIKV is 17-19°C, though the scale of 17-19°C is quite wide when we are talking about the climate and this should be discussed or even additional experiments. There also are other critical issues that should be addressed in my opinion. First, there is no data of the copy numbers for ZIKV in the mosquito saliva. Is there difference in the viral loads in *detritus* and *albopictus* saliva? This data would support the conclusion of *Ae. albopictus* being more competent vector than *Oc. detritus* and I think authors should provide this data and discuss the differences. What is the infectious dose in the saliva? I also would recommend discussion of seasonal transmission. There are large areas, where the continuous transmission is very unlikely in future decades but when suitable vector is present, the temperatures might allow seasonal transmission, which is not considered at all in the manuscript. I suggest downgrading the conclusion of assessing the vector competence of four mosquito species as two of them were applicable in such a small number that comprehensive experiments could not be conducted. In the experiments it was found that in lower temperatures it took longer for the virus to be detected from the saliva. When the timeline from blood meal to positive saliva exceeds 20 days, it would be interesting to discuss the significance in practice, as what is the natural lifespan for these mosquito species in the nature. The spelling of the manuscript needs quite a lot of revision. Both the main text and supplementary material include two different fonts. The terms with capitals, (Southern, Northern, Central etc) should be corrected throughout the text.

Specific comments

Abstract

Line 47 change japan to Japan

Introduction

Lines 52-53 "Though infection is asymptomatic in 80% of cases (1), a small proportion of patients develop the autoimmune condition Guillain-Barré syndrome (2)". This should be rephrased as now it reads as ZIKV infection is either asymptomatic or it manifests as GBS.

Line 56 World Health Organisation, please change to World Health Organization

Line 67 Please clarify what you mean by "pathogen to develop"

Lines 71-73 "With large human populations in temperate climates it is also important to estimate how ZIKV transmission occur in such climates today, and with future climate change." Do authors mean that already occurs or that could occur? Please clarify

Line 73 "future climate change" is probably not the best term as climate change itself occurs already. I suggest to change it to in future climate for example.

Line 84 RCPx, what does the x stand for? Should it be s?

Results

Lines 88-89 "Zika virus was detected in the saliva of both *Ae. albopictus* (VERANO colony) and *Oc. detritus* at all temperatures from 19oC to 31oC, but not at 17oC" Please add copy numbers, this would be nice to see in a table. But it would clarify the effect of temperature on the transmission.

Line 152 change Tropics to tropics

Line 154 change central Africa to Central Africa

Line 157 should three dots after Malaysia be removed?

Line 163 "parts of Turkey" please specify which parts

Lines 168-171 “Examples include Ravenna in the Emilia-Romagna region in Italy where a chikungunya outbreak occurred in 2007; nearby Cadiz in southern Spain and on the eastern Catalan coasts with a few reported and suspected cases of dengue in 2018; south-eastern France where cases of dengue were reported in 2010, 2013, 2014 and 2015 (20)” Please modify this sentence.

Line 184 Correct to temperature only- effect

Line 187 change central Australia to Central Australia

Line 215 “However, there is a decrease in simulated $R_0(T)$ over Brazil in future for the 2080s (Fig. S6h).” some discussion on the reasons behind this would be good add.

Discussion

Line 256 Please clarify what does “high laboratory competence” mean

Line 261 please refer to the study not only to a reference number in the text.

Line 285 “in contrast to USA and temperate Asia” I would suggest referring to “in contrast to the maps of USA and Asia”

Lines 292-293 “*Ae. albopictus* is not a major issue in Africa yet”, please retype this. This refers to the presence of *Ae. albopictus* ‘

Line 293 should altitude regions be high altitude regions?

Line 309 “Locally they...” Does this refer to both *Ae. albopictus* and *aegypti* or other species?

Please clarify

Line 316 “Our model shows only the risk as a result of climate and vector species” please retype this. For example: as a result of the effect of climate and vector species

Lines 323-324 “The interplay between these, and other, factors will likely have a large effect on the risk to a population.” Please retype.

In general, the validity of the models used should be discussed more thoroughly as now there is very little about it.

In results the authors state that: “If we remove the rainfall effect in our R_0 model (temperature only effect), the surface at risk increases significantly and to some extent unrealistically, with large values simulated in desert regions such as the Sahara (Fig. S3b), parts of the Middle East (Fig. S3a), the “ yet there is no discussion about the rainfall model

The limitations of the study should include discussion of the *Cx. pipiens pipiens* and *Cs. annulata* experiments as they were not comprehensive compared to other two species. One option would be to exclude data on these as obviously their vector competence must be studied further.

Materials and methods

Line 359 Please specify where the human blood for feeding the mosquitoes was achieved

Is RNA extraction with Trizol the best choice for saliva? In my experience I might suspect losing some RNA from the weak positive samples with this method.

Figures and tables

In all the map figures that are zoomed (so not including the world map), in supplementary material and the main text, the coordinates would be helpful especially in the boundaries of zoomed maps, in order to recognize easier where is e.g. equator (grids to the boundaries of these maps) , so that it would possible and easier to estimate more accurate (latitude and longitude) risk area.

Figure 2 There is no referring to figure 2c in the text

Figure 3 Dots are very small, should be bigger to be able to see them

Figure 4, parts a-h are not explained in the caption. Please add this.

Figure 4 line 233 “the most extreme” quite strong expression, prefer to high or higher emission scenario etc

Figure legends should consistently be placed under the figure and table legends above the table. There is variation in this now. Please correct.

Supplementary material

Lines 76-78 “we employed the Global Precipitation Climatology Centre (GPCC) global rainfall data available at similar spatial and time resolution for the same time period (7).” Exactly same or just similar resolution? Please specify.

Lines 78-79 “We calculated annual average for the 1980-2010 period, as advised by the World Meteorological Office guidelines” Please add a reference for this.

Lines 83-84 “for the ensemble mean of 5 GCM (hadgem2-es, ipsl-cm5a-lr, miroc-esm-chem, gfdl-esm2m, noresm1-m)” Could you explain why did you choose to make ensemble of these 5 GCMs specifically?

Supplementary figure 6 legend “The beige colour depicts standardized R0(T) values for which some ZIKV transmission by *Ae. albopictus* might occur in the laboratory [17-19°C]”. Please clarify this.

Supplementary table 1. Retype model names: 1. Equation, for example

Supplementary table 5 the caption of the table is above the table as it should be but the legend is placed under the table. Please correct.

Referee: 3

Comments to the Author(s)

The authors are describing an interesting study of ZIKV transmission, suggesting firstly that *Oc. detritus* may vector ZIKV in temperate regions at temperature 19C- 24C and that *Ae albopictus* could also transmit ZIKV at lower temperatures than previously reported (19, 21, 27C).

The result of *Oc. detritus* transmitting ZIKV is important, and its implications in the regions where this mosquito is present are significant. The authors state this in their discussion and point out the difficulty of vector control to mosquitoes that are breeding in natural water sources.

The main concerns in this study arise from interpretation of the obtained results, and considerations of what these results mean. The findings are interesting, but should be considered preliminary. Further evidence and more detailed information of the transmission temperature (and other) thresholds would be needed before applying the values to modelling and risk evaluations.

The experimental evidence shows that no transmission (virus in the saliva) occurs at temperature of 17C in either of the species tested (*Ae. albopictus* or *Oc. detritus*). Therefore, the statement that the authors have identified a minimum temperature threshold between 17 and 19 C (abstract line 42) is an overestimate. The authors are speculating (without evidence) that the transmission would occur also below 19C although they do not have any data to support this assumption. Further testing should be done to identify the exact lowest limit, whether that is between 17.1-18.9C in the tested mosquito species.

Overall the manuscript structure and clarity should be improved, and the relevant contents should be shown in the manuscript (and not in the supplementary file). There are too many different map images provided, it seems that the authors cannot themselves decide what is relevant and what is not. The authors should provide evidence for their laboratory findings to have implications also to real life. The authors do not show the data of the virus amount in the saliva of *Oc. detritus*, or compare that to those obtained from *Ae albopictus*. The results should be discussed also in the light of the results obtained with *Ae aegypti*, that showed the low temperatures not supporting virus transmission to saliva (Tesla B et al., 2018). It would also be important to demonstrate that in the case of *Ae albopictus* and *Oc. detritus*, at low temperatures, the virus PCR positive saliva would actually be infectious (in cell culture or mosquito infection). And as a control, it would have been good to add *Ae aegypti* to the tests to rule out e.g. viral strain properties or other possibly affecting factors that are different from the previously published work.

The manuscript should be edited to highlight the actual findings and not the modelling and estimates that are at present highly speculative. If such maps are shown, they should be clearly explained and discussed in the context of other, previously published risk maps. The authors state in the abstract “Our R0-based risk maps show significant risk of ZIKV transmission beyond the current observed range in southern USA, southern China and southern European countries.” It may be so that the maps show risk beyond currently observed areas, but the maps look pretty modest in comparison to other predicted risk maps of *Ae albopictus* transmission, for example: Ryan SJ et al, PLOS N trop Dis 2019. It would be important for the temperate regions to take the seasonality (e.g. summer months transmission) into account and also show these in the maps. The authors are only talking about sustained transmission (in the discussion lines 332-333). The authors should report only results from *Oc. detritus* and *Ae. Albopictus*, as there is comparable data for these two. There is only very limited testing done (not all temperatures/time points) for *Cx. Pipiens* and *Cs. Annulata*. So the statement that four mosquito species were studied (abstract lines 37, 38) is misleading.

One of the concerns of this paper is the small numbers of mosquitoes used in some of the experiments. The numbers of mosquitoes in experiments are only found in the supplementary file. The n-numbers should be presented in the actual manuscript in some form.

The authors should cite the relevant previous work on the topic, such as the work done using *Aedes japonicus* (Jansen et al., 2018) and *Aedes vexans* (Gendernalik et al., 2017). To what extent are the study methodologies comparable?

The authors are discussing their findings in context of *Ae albopictus*, but very little is said about *Oc. detritus*. For example, the authors should give some overview to the readers about *Oc. detritus* distribution.

Discussion lines 333-335: “The comprehensive analysis of EIP and minimum temperature requirements presented here offer a route to enable informed risk-management and outbreak preparedness in more specific temperate situations.” What do the authors mean by this? How is the data presented here applicable to risk management or outbreak preparedness?

Line 385: “As is standard practice with qRT-PCR experiments, a ‘cut-off’ of 40 Ct was applied to minimise false positives towards the limit of detection.” What do the authors mean by this? The used method is a probe- based assay? I would understand this if it was a SYBR green assay.

Author's Response to Decision Letter for (RSPB-2019-1973.R0)

See Appendix A.

RSPB-2020-0119.R0

Review form: Reviewer 1

Recommendation

Major revision is needed (please make suggestions in comments)

Scientific importance: Is the manuscript an original and important contribution to its field?

Good

General interest: Is the paper of sufficient general interest?

Good

Quality of the paper: Is the overall quality of the paper suitable?

Acceptable

Is the length of the paper justified?

No

Should the paper be seen by a specialist statistical reviewer?

Yes

Do you have any concerns about statistical analyses in this paper? If so, please specify them explicitly in your report.

No

It is a condition of publication that authors make their supporting data, code and materials available - either as supplementary material or hosted in an external repository. Please rate, if applicable, the supporting data on the following criteria.

Is it accessible?

Yes

Is it clear?

No

Is it adequate?

Yes

Do you have any ethical concerns with this paper?

No

Comments to the Author

In this paper the authors extrapolate solid laboratory data into a global model for ZIKV risk based on temperature driven parameters. As *Ae. albopictus* seems limited to areas with relatively high rainfall/humidity, it was not clear how the authors accommodated these data into their models. Specific comments have been entered directly on the attached files using tracked changes [a little cumbersome after converting the .pdf to .docx].

Summary thoughts follow.

1. Writing/English. The paper could use some additional editing to improve the sentence structure and general flow of ideas. I have tried to help [see attached file], but I feel a good 'read' would be useful.
2. Model. My greatest problem was using *Aedes albopictus* and *Ae. detritus* to model ZIKV transmission risk based on R_0 estimated for temperature areas [some of which don't have *detritus*] or limited populations of *albopictus*, and not including the primary vector, *Ae. aegypti*. As the authors acknowledge, large ZIKV outbreaks have occurred mainly where there are populations of *aegypti*. Although *albopictus* is an efficient laboratory vector of many arboviruses, it does not seem to be a primary vector of any arbovirus, probably because of its general host selection patterns. Similar comments relate to other species mentioned such as *Culex* spp., *Ae. japonicus*, *Ae. vexans*, etc. These species may become naturally infected but will not sustain transmission human-to-human transmission. As the authors point out, *albopictus* is broadly distributed in the USA, but secondary ZIKV and DENV transmission from travelers have mostly occurred where there are suitable populations of *aegypti* and not within the more northern temperate areas with established *albopictus* despite the repeated introductions of viruses by travelers.
3. Parameters. I didn't understand some of the model parameters:

$b=0.5$: Does this mean that half the albopictus bites are on humans result in transmission? If this is for infectious females, I think this could be low. If based all females, then it is too high.

$m = 28.2$: Does this mean that throughout the world there are, on average, 28 albopictus per human host? I agree that a constant value will not alter the shape of the temperature dependent curves, but rather should alter the magnitude of the R_0 estimates. What would R_0 be if this value was 2.8? This was even more problematic in a one host model where all females feed on humans -- not the normal feeding patterns observed for albopictus where many bites are diverted to other hosts.

Review form: Reviewer 3

Recommendation

Accept with minor revision (please list in comments)

Scientific importance: Is the manuscript an original and important contribution to its field?

Good

General interest: Is the paper of sufficient general interest?

Good

Quality of the paper: Is the overall quality of the paper suitable?

Acceptable

Is the length of the paper justified?

Yes

Should the paper be seen by a specialist statistical reviewer?

Yes

Do you have any concerns about statistical analyses in this paper? If so, please specify them explicitly in your report.

No

It is a condition of publication that authors make their supporting data, code and materials available - either as supplementary material or hosted in an external repository. Please rate, if applicable, the supporting data on the following criteria.

Is it accessible?

Yes

Is it clear?

N/A

Is it adequate?

N/A

Do you have any ethical concerns with this paper?

No

Comments to the Author

I am glad to see that the authors have tested the saliva samples for viable virus, however this part requires some clarification in the text.

Decision letter (RSPB-2020-0119.R0)

26-Feb-2020

Dear Dr Blagrove,

Thank you for the revised version of this manuscript, which has now been peer reviewed and the reviews have been assessed by an Associate Editor. The reviewers' comments (not including confidential comments to the Editor) and the comments from the Associate Editor are included at the end of this email for your reference. As you will see, the reviewers and the Editors have raised some concerns with your manuscript and we would like to invite you to revise your manuscript to address them. In addition to the reviewers' comments, please see the comment from the Associate Editor concerning the need to make clear the interest of the paper to a broad Proc B readership.

Research ethics:

Use of animals and field studies:

It is a condition of publication that you make available the data and research materials supporting the results in the article. Datasets should be deposited in an appropriate publicly available repository and details of the associated accession number, link or DOI to the datasets

must be included in the Data Accessibility section of the article (<https://royalsociety.org/journals/ethics-policies/data-sharing-mining/>). Reference(s) to dataset(s) should also be included in the reference list of the article with DOIs (where available).

Please submit a copy of your revised paper within three weeks. If we do not hear from you within this time your manuscript will be rejected. If you are unable to meet this deadline please let us know as soon as possible, as we may be able to grant a short extension.

Best wishes,
Professor Loeske Kruuk
mailto: proceedingsb@royalsociety.org

Associate Editor Board Member

Comments to Author:

I appreciate the efforts that the authors have made to address the initial reviewer concerns. However, several issues remain that have been raised by the reviewers. For example, reviewer 1 particularly draws attention to the importance of including *Ae. aegyti* in ZIKV models, and the need for clarity on the parameter estimates presented. Reviewer 2 points out additional information required for understanding and contextualizing the results from saliva samples. These aside, I am ultimately left questioning how much of an impact the results here would have in progressing our understanding of ZIKV risk, and if they would interest the broad readership of Proc B.

Reviewer(s)' Comments to Author:

Referee: 1

Comments to the Author(s).

In this paper the authors extrapolate solid laboratory data into a global model for ZIKV risk based on temperature driven parameters. As *Ae. albopictus* seems limited to areas with relatively high rainfall/humidity, it was not clear how the authors accommodated these data into their models. Specific comments have been entered directly on the attached files using tracked changes [a little cumbersome after converting the .pdf to .docx].

Summary thoughts follow.

1. Writing/English. The paper could use some additional editing to improve the sentence structure and general flow of ideas. I have tried to help [see attached file], but I feel a good 'read' would be useful.
2. Model. My greatest problem was using *Aedes albopictus* and *Ae. detritus* to model ZIKV transmission risk based on R_0 estimated for temperature areas [some of which don't have *detritus*] or limited populations of *albopictus*, and not including the primary vector, *Ae. aegypti*. As the authors acknowledge, large ZIKV outbreaks have occurred mainly where there are populations of *aegypti*. Although *albopictus* is an efficient laboratory vector of many arboviruses, it does not seem to be a primary vector of any arbovirus, probably because of its general host selection patterns. Similar comments relate to other species mentioned such as *Culex* spp., *Ae. japonicus*, *Ae. vexans*, etc. These species may become naturally infected but will not sustain transmission human-to-human transmission. As the authors point out, *albopictus* is broadly distributed in the USA, but secondary ZIKV and DENV transmission from travelers have mostly occurred where there are suitable populations of *aegypti* and not within the more northern temperate areas with established *albopictus* despite the repeated introductions of viruses by travelers.
3. Parameters. I didn't understand some of the model parameters:
 $b=0.5$: Does this mean that half the *albopictus* bites on humans result in transmission? If this is for infectious females, I think this could be low. If based all females, then it is too high.
 $m = 28.2$: Does this mean that throughout the world there are, on average, 28 *albopictus* per human host? I agree that a constant value will not alter the shape of the temperature dependent curves, but rather should alter the magnitude of the R_0 estimates. What would R_0 be if this value was 2.8? This was even more problematic in a one host model where all females feed on humans -- not the normal feeding patterns observed for *albopictus* where many bites are diverted to other hosts.

Referee: 3

Comments to the Author(s).

I am glad to see that the authors have tested the saliva samples for viable virus, however this part requires some clarification in the text.

Author's Response to Decision Letter for (RSPB-2020-0119.R0)

See Appendix B.

RSPB-2020-0119.R1 (Revision)

Review form: Reviewer 4

Recommendation

Major revision is needed (please make suggestions in comments)

Scientific importance: Is the manuscript an original and important contribution to its field?

Good

General interest: Is the paper of sufficient general interest?

Acceptable

Quality of the paper: Is the overall quality of the paper suitable?

Marginal

Is the length of the paper justified?

Yes

Should the paper be seen by a specialist statistical reviewer?

No

Do you have any concerns about statistical analyses in this paper? If so, please specify them explicitly in your report.

No

It is a condition of publication that authors make their supporting data, code and materials available - either as supplementary material or hosted in an external repository. Please rate, if applicable, the supporting data on the following criteria.

Is it accessible?

Yes

Is it clear?

No

Is it adequate?

No

Do you have any ethical concerns with this paper?

No

Comments to the Author

General comments

The manuscript by Blagrove et al. with the title Potential for Zika virus transmission by mosquitoes in temperate climates is split into two important research topics: first, vector competence studies of four temperate mosquito species, and second modelling, using outcomes of the experimental study for *Aedes albopictus* to evaluate the impact on disease risk (R0) on a global scale. To combine these two in one publication is very valuable. When reading the title this was not immediately clear, so we propose to make this clear in the title.

Overall the manuscript is well written, however, the manuscript needs to be thoroughly revised since it is hard to follow at times. Sections are often in the wrong place, e.g. many supplementary method sections should actually be placed in the main manuscript and there are discussion items in the results section. The discussion is very lengthy. We will elaborate more on these issues

below.

Although the work is an interesting read for a specific niche, we think it is currently not always understandable for the broader audience of Proc Roy Soc B that may not be familiar with the subject area. We will further elaborate on our main issues below.

Major comments

- 1) From reading the manuscript it is not immediately clear which experimental data points went into the model. There is no bridge presented in the M&M or Results section that explains this. Modelling is not my personal expertise, but for this journal it has to be understandable for a broader public. This needs thorough revision.
- 2) I feel that in both Introduction and Discussion the authors should put the story in more perspective. They only mention general information on temperature and effects on pathogens or ZIKV more specifically (paragraph starting with line 67). I would like to see more information on the effect of temperature on other arboviruses such as WNV, a flavivirus just like ZIKV. In the discussion the authors do mention an example of temperature on malaria transmission (line 317-310), however arbovirus examples would be better matching and relevant to the discussion.
- 3) Line 113-120. and 225-227. I advise to remove all data on the vector competence of *Cx. pipiens pipiens* and *Cs. annulata*. Not enough tests were performed to make a solid statement. The information only makes the manuscript more confusing.
- 4) There are several items in the Supplementary materials that should be put in the main manuscript:
 - a. Models of extrinsic incubation period (EIP10)
 - b. Detection of infectious virus
 - c. Derivation of $R_0(T)$ for ZIKV – *Ae. albopictus* model

Reading these parts are essential for understanding the manuscript. Some Figures and Tables can stay in the supplementary material such as Table S3, Figure S1, Figure S2, Table S4 and Figure S3 etc. These need to be placed in the order they appear in the main text though, which is not the case now and results in great difficulty reading and reviewing the results section of the manuscript.

- 5) The link between risk of ZIKV and vector presence is unclear. The presence of *Aedes albopictus* is plotted with black dots in Fig. 2. This needs more explanation. For example, risk of ZIKV is high in Suriname, however, *Ae. albopictus* is not present there, so it's not a realistic risk?

Minor comments

Keywords

- 1) Line 31. Too many keywords mentioned, only six allowed.

Abstract

- 2) Line 42. Specify the wide range of incubation temperatures.
- 3) Line 51. As result of what do you expect changes in the future? Temperature?

Introduction

- 4) Line 54. The Zika virus is no longer a 'rapidly emerging' disease.
- 5) Line 55 and Line 66. Include a reference at the end of these sentence to support the factual statements made here.
- 6) Line 87. This is about reported vector presence of *Ae. albopictus* specifically, adjust. Also provide the source.

Material and methods

- 7) Line 352-364. More information should be given on ambient conditions, larval diet, and year and season of sampling of the mosquitoes.
- 8) Line 320. What is an 'odorized' feeding membrane? What did it consist of?
- 9) Line 371. Give more details, which virus passage has been used, on which cells was the virus passaged, which medium was used? How much blood was used relative to virus? This should increase repeatability.
- 10) Line 378-385. More clarity is needed. For example how long were the mosquitoes salivated? The reference Blagrove et al. 2018 (48) does not give enough information. What type of mineral oil is used? How much? As it is now, there is not enough information to repeat the experiment.
- 11) Line 385. Carcass is mentioned, in the supplementary materials this is called body. Choose one of the two.

- 12) Line 390. 'the kit'.
 13) Line 395. Need more clarity. How do you assess PFU's? Which cells are used, which medium (agar)? And how long do you wait for PFU's to show?
 14) Line 396. 'Limit' without capital letter.
 15) Line 398-400. See Major comment 4.

Results

- 16) Line 92. How many mosquitoes were tested at each T? Refer here to Table S3.
 17) Line 94. Letters in Fig. 1 do not correspond with the ones in text, they are turned around. Also I was wondering, why not show average titres of each time point too?
 18) Line 104. It's not clear from the methods you did experiments looking at CPE. See major comment 4. Also add the word 'and' in 'challenge, and were pooled'. And in the next sentence change 4 into four.
 19) Line 111. This is an important result which should not be 'hidden' in the Supplementary materials in Table S5. The Table should be changed, see comment 56.
 20) Line 123. What is the idea behind EIP10? Why is this chosen and not EIP5 or EIP15? Also, 'the time in days to reach 10% saliva-positive', how is this calculated? Could not find this in the methods section.
 21) Line 127-128. Model 1 identified a linear relationship, but in what direction?
 22) Line 131-132. So what is this estimate to which it is similar?
 23) Line 145. Fig.2. The black dots represent presence of *Ae. albopictus*, but where does the data come from? There is a reference in the figure legend, but this should be mentioned in the main text. Also in the Eastern part of South America, *Ae. albopictus* is so abundant, you cannot clearly see the R0 colours. Also see major comment 5.
 24) Line 149. 'they are consistent with former published estimates'. This is part of the Discussion?
 25) Line 166-174. Should be part of the Discussion.
 26) Line 190. Change: vector presence in 'black dots' and standardized R0 in 'colour code'.
 27) Line 191. significantly 'statistically'? Or do you mean substantially?
 28) Line 219. What is not shown?

Discussion

- 29) Line 232. Add that you based your model only on *Ae. albopictus*.
 30) Line 240. Table S6 should be changed to S5.
 31) Line 240-241. Rephrase sentence for clarity.
 32) Line 246. But were similar incubation times used in this study as in the study of Heitmann et al?
 33) Line 253. Happened how, can you elaborate?
 34) Line 276. By which autochthonous species?
 35) Line 282. 'the' presence
 36) Line 298. Which 'is' important, change accordingly.
 37) Line 301. What is the distribution range of *Oc. detritus*? So what is its broader relevance? This needs to be elaborated.
 38) Line 307. Here the role of *Oc. detritus* is discussed further, but the vector distribution and chances of vector-human presence should be taken into account. Discuss this in the context of vectorial capacity.
 39) Line 320. What does that mean? Why was it underestimated? Also see major comment 2.
 40) Line 327. But *Ae. albopictus* is absent in for example Suriname, so what is the link between the maps and vector presence? See major comment 5.
 41) Line 342. I agree, but specify in what direction this difference is?

Acknowledgements

- 42) 406. Replace 'Raid' by 'Rapid'.

References

- 43) Line 480. The references should be updated as much as possible. Some were cited in 2017, is there more recent literature? Not all species names are in italics and there are misplaced capital letters which do not need to be there when using Vancouver style referencing as is recommended for by Proc Roy Soc B.

Figure legends

- 44) Line 615. See comment 51, then change condition into temperature.
- 45) Line 621-622. In this sentence *Ae. aegypti* should be changed into *Ae. albopictus*? Also this is a result which should go into the text rather than in the figure legend. 'The δs for *Ae. albopictus* (0.27) was significantly greater than that of *Oc. detritus* (0.20) (Table S2), indicating that *Ae. aegypti* is more competent for ZIKV at all temperatures.'
- Supplementary methods
- 46) Line 17. Add that plateau value = δs .
- 47) Line 28. See comment 20.
- 48) Line 33. Fill in the supplement number.
- 49) Line 37. What are models 3-7? We don't see them in the manuscript? They are mentioned in Table S1, but completely unclear what they contain.
- 50) Line 34, 38 and 40. Example of wrongly placed text in the supplementary methods. They should be placed with the Figure or Table of interest.
- 51) Line 48. Table S3. In the first column of the table, *Ae. albopictus* is not written in full. We would suggest to split the table in two, here only show numbers of the transmission experiment. Show the numbers of the survival experiment underneath Figure S1 and also refer to this new table in Line 99 of the main text.
- 52) Line 61. Figure S1. For both species, is it not strange that the survival increases with time? See for example *Oc. detritus* at 21oC at day 14.
- 53) Line 68. Figure S2 is referred to, but not discussed in the main text (line 619). Not clear what the figure adds?
- 54) Line 74. See major comment 4, move to main text.
- 55) Line 84 Table S4. In the table text, add meaning of Y, N and dpi.
- 56) Line 89. Table S5. These are actually two tables, so split them. Most importantly it is unclear in what units the relative titres are expressed. Are these TCID50, or PFU values? Compress the table by removing the 17oC row, and 0 and 5 days columns.
- 57) Line 113. Which Fig. 2 and Table 1 show 'the related analytical functions'? In the main manuscript or in the supplements?
- 58) Line 126-127. Can you describe this differently? It is not immediately clear how the standardized values correspond to the temperatures. Also this can not be derived from Fig.1 which is referred to.
- 59) Line 129. What is the source of the temperature and climate model projections?
- 60) Line 141. Fig. S4. The difference between figure S4 and S5 is not immediately clear, can this be adjusted? In panel d the solid red line is not very visible
- 61) Line 201. The model comparison of proportion of saliva-positive mosquitoes is missing?

Decision letter (RSPB-2020-0119.R1)

23-Apr-2020

Dear Dr Blagrove:

Thank you for the resubmission of your revised manuscript and the work you have put into addressing the previous referees' comments. Your revised manuscript has now been reviewed by a new reviewer and the review has been assessed by an Associate Editor. The reviewer's comments (not including confidential comments to the Editor) and the comments from the Associate Editor are included at the end of this email for your reference. As you will see, the reviewer has raised a large number of concerns with this version of the manuscript. We are therefore need to ask you to invite you to revise your manuscript to address them where possible. Many of the reviewer's suggestions should improve the clarity of the presentation. However I am fully aware that the manuscript has gone through two rounds of revision before, and I fully appreciate the work you have put into previous revisions. In normal situations, we would avoid

manuscripts being sent to new reviewers on every round of revision, but things are challenging at present. Please contact me if there is anything you would like to discuss about the revisions.

Research ethics:

Use of animals and field studies:

Please submit a copy of your revised paper within three weeks. If you are unable to meet this deadline, especially given the current global situation with the pandemic, please let us know as soon as possible, as we may be able to grant an extension.

Finally, I hope you and your co-authors are all well at this challenging time.

With best wishes,
 Professor Loeske Kruuk
 Editor, Proceedings B
 mailto: proceedingsb@royalsociety.org

Associate Editor
 Board Member: 1

Comments to Author:

Thank you for addressing the reviewer concerns. I appreciate the time and effort that has gone into this. The manuscript has now been evaluated by an additional reviewer who has also made several points, mainly to help clarify methodological aspects of the work and emphasize its broader implications. For example, they have provided suggestions for discussion points in the introduction and discussion sections, and information that could be moved from the supplementary material to the manuscript body.

Reviewer(s)' Comments to Author:

Referee: 4

Comments to the Author(s)

General comments

The manuscript by Blagrove et al. with the title Potential for Zika virus transmission by mosquitoes in temperate climates is split into two important research topics: first, vector competence studies of four temperate mosquito species, and second modelling, using outcomes of the experimental study for *Aedes albopictus* to evaluate the impact on disease risk (R0) on a global scale. To combine these two in one publication is very valuable. When reading the title this was not immediately clear, so we propose to make this clear in the title.

Overall the manuscript is well written, however, the manuscript needs to be thoroughly revised since it is hard to follow at times. Sections are often in the wrong place, e.g. many supplementary method sections should actually be placed in the main manuscript and there are discussion items in the results section. The discussion is very lengthy. We will elaborate more on these issues below.

Although the work is an interesting read for a specific niche, we think it is currently not always understandable for the broader audience of Proc Roy Soc B that may not be familiar with the subject area. We will further elaborate on our main issues below.

Major comments

- 1) From reading the manuscript it is not immediately clear which experimental data points went into the model. There is no bridge presented in the M&M or Results section that explains this. Modelling is not my personal expertise, but for this journal it has to be understandable for a broader public. This needs thorough revision.
- 2) I feel that in both Introduction and Discussion the authors should put the story in more perspective. They only mention general information on temperature and effects on pathogens or ZIKV more specifically (paragraph starting with line 67). I would like to see more information on the effect of temperature on other arboviruses such as WNV, a flavivirus just like ZIKV. In the discussion the authors do mention an example of temperature on malaria transmission (line 317-310), however arbovirus examples would be better matching and relevant to the discussion.
- 3) Line 113-120. and 225-227. I advise to remove all data on the vector competence of *Cx. pipiens pipiens* and *Cs. annulata*. Not enough tests were performed to make a solid statement. The information only makes the manuscript more confusing.
- 4) There are several items in the Supplementary materials that should be put in the main manuscript:

- a. Models of extrinsic incubation period (EIP10)
- b. Detection of infectious virus
- c. Derivation of $R_0(T)$ for ZIKV – *Ae. albopictus* model

Reading these parts are essential for understanding the manuscript. Some Figures and Tables can stay in the supplementary material such as Table S3, Figure S1, Figure S2, Table S4 and Figure S3 etc. These need to be placed in the order they appear in the main text though, which is not the case now and results in great difficulty reading and reviewing the results section of the manuscript.

- 5) The link between risk of ZIKV and vector presence is unclear. The presence of *Aedes albopictus* is plotted with black dots in Fig. 2. This needs more explanation. For example, risk of ZIKV is high in Suriname, however, *Ae. albopictus* is not present there, so it's not a realistic risk?

Minor comments

Keywords

- 1) Line 31. Too many keywords mentioned, only six allowed.

Abstract

- 2) Line 42. Specify the wide range of incubation temperatures.
- 3) Line 51. As result of what do you expect changes in the future? Temperature?

Introduction

- 4) Line 54. The Zika virus is no longer a 'rapidly emerging' disease.
- 5) Line 55 and Line 66. Include a reference at the end of these sentence to support the factual statements made here.
- 6) Line 87. This is about reported vector presence of *Ae. albopictus* specifically, adjust. Also provide the source.

Material and methods

- 7) Line 352-364. More information should be given on ambient conditions, larval diet, and year and season of sampling of the mosquitoes.
- 8) Line 320. What is an 'odorized' feeding membrane? What did it consist of?
- 9) Line 371. Give more details, which virus passage has been used, on which cells was the virus passaged, which medium was used? How much blood was used relative to virus? This should increase repeatability.
- 10) Line 378-385. More clarity is needed. For example how long were the mosquitoes salivated? The reference Blagrove et al. 2018 (48) does not give enough information. What type of mineral oil is used? How much? As it is now, there is not enough information to repeat the experiment.
- 11) Line 385. Carcass is mentioned, in the supplementary materials this is called body. Choose one of the two.
- 12) Line 390. 'the kit'.
- 13) Line 395. Need more clarity. How do you assess PFU's? Which cells are used, which medium (agar)? And how long do you wait for PFU's to show?
- 14) Line 396. 'Limit' without capital letter.
- 15) Line 398-400. See Major comment 4.

Results

- 16) Line 92. How many mosquitoes were tested at each T? Refer here to Table S3.
- 17) Line 94. Letters in Fig. 1 do not correspond with the ones in text, they are turned around. Also I was wondering, why not show average titres of each time point too?
- 18) Line 104. It's not clear from the methods you did experiments looking at CPE. See major comment 4. Also add the word 'and' in 'challenge, and were pooled'. And in the next sentence change 4 into four.
- 19) Line 111. This is an important result which should not be 'hidden' in the Supplementary materials in Table S5. The Table should be changed, see comment 56.
- 20) Line 123. What is the idea behind EIP10? Why is this chosen and not EIP5 or EIP15? Also, 'the time in days to reach 10% saliva-positive', how is this calculated? Could not find this in the methods section.
- 21) Line 127-128. Model 1 identified a linear relationship, but in what direction?
- 22) Line 131-132. So what is this estimate to which it is similar?
- 23) Line 145. Fig.2. The black dots represent presence of *Ae. albopictus*, but where does the data come from? There is a reference in the figure legend, but this should be mentioned in the main text. Also in the Eastern part of South America, *Ae. albopictus* is so abundant, you cannot clearly see the R0 colours. Also see major comment 5.
- 24) Line 149. 'they are consistent with former published estimates'. This is part of the Discussion?
- 25) Line 166-174. Should be part of the Discussion.
- 26) Line 190. Change: vector presence in 'black dots' and standardized R0 in 'colour code'.
- 27) Line 191. significantly 'statistically'? Or do you mean substantially?
- 28) Line 219. What is not shown?

Discussion

- 29) Line 232. Add that you based your model only on *Ae. albopictus*.
- 30) Line 240. Table S6 should be changed to S5.
- 31) Line 240-241. Rephrase sentence for clarity.
- 32) Line 246. But were similar incubation times used in this study as in the study of Heitmann et al?
- 33) Line 253. Happened how, can you elaborate?
- 34) Line 276. By which autochthonous species?
- 35) Line 282. 'the' presence
- 36) Line 298. Which 'is' important, change accordingly.
- 37) Line 301. What is the distribution range of *Oc. detritus*? So what is its broader relevance? This needs to be elaborated.
- 38) Line 307. Here the role of *Oc. detritus* is discussed further, but the vector distribution and chances of vector-human presence should be taken into account. Discuss this in the context of vectorial capacity.
- 39) Line 320. What does that mean? Why was it underestimated? Also see major comment 2.
- 40) Line 327. But *Ae. albopictus* is absent in for example Suriname, so what is the link between the maps and vector presence? See major comment 5.
- 41) Line 342. I agree, but specify in what direction this difference is?

Acknowledgements

- 42) 406. Replace 'Raid' by 'Rapid'.

References

- 43) Line 480. The references should be updated as much as possible. Some were cited in 2017, is there more recent literature? Not all species names are in italics and there are misplaced capital letters which do not need to be there when using Vancouver style referencing as is recommended for by Proc Roy Soc B.

Figure legends

- 44) Line 615. See comment 51, then change condition into temperature.
- 45) Line 621-622. In this sentence *Ae. aegypti* should be changed into *Ae. albopictus*? Also this is a result which should go into the text rather than in the figure legend. 'The δ s for *Ae. albopictus* (0.27) was significantly greater than that of *Oc. detritus* (0.20) (Table S2), indicating that *Ae. aegypti* is more competent for ZIKV at all temperatures.'

Supplementary methods

- 46) Line 17. Add that plateau value = δ s.
- 47) Line 28. See comment 20.
- 48) Line 33. Fill in the supplement number.
- 49) Line 37. What are models 3-7? We don't see them in the manuscript? They are mentioned in Table S1, but completely unclear what they contain.
- 50) Line 34, 38 and 40. Example of wrongly placed text in the supplementary methods. They should be placed with the Figure or Table of interest.
- 51) Line 48. Table S3. In the first column of the table, *Ae. albopictus* is not written in full. We would suggest to split the table in two, here only show numbers of the transmission experiment. Show the numbers of the survival experiment underneath Figure S1 and also refer to this new table in Line 99 of the main text.
- 52) Line 61. Figure S1. For both species, is it not strange that the survival increases with time? See for example *Oc. detritus* at 21oC at day 14.
- 53) Line 68. Figure S2 is referred to, but not discussed in the main text (line 619). Not clear what the figure adds?
- 54) Line 74. See major comment 4, move to main text.
- 55) Line 84 Table S4. In the table text, add meaning of Y, N and dpi.
- 56) Line 89. Table S5. These are actually two tables, so split them. Most importantly it is unclear in what units the relative titres are expressed. Are these TCID50, or PFU values? Compress the table by removing the 17oC row, and 0 and 5 days columns.
- 57) Line 113. Which Fig. 2 and Table 1 show 'the related analytical functions'? In the main manuscript or in the supplements?
- 58) Line 126-127. Can you describe this differently? It is not immediately clear how the standardized values correspond to the temperatures. Also this can not be derived from Fig.1 which is referred to.
- 59) Line 129. What is the source of the temperature and climate model projections?
- 60) Line 141. Fig. S4. The difference between figure S4 and S5 is not immediately clear, can this be adjusted? In panel d the solid red line is not very visible
- 61) Line 201. The model comparison of proportion of saliva-positive mosquitoes is missing?

Author's Response to Decision Letter for (RSPB-2020-0119.R1)

See Appendix C.

RSPB-2020-0119.R3 (Revision)

Review form: Reviewer 4

Recommendation

Accept with minor revision (please list in comments)

Scientific importance: Is the manuscript an original and important contribution to its field?

Good

General interest: Is the paper of sufficient general interest?

Good

Quality of the paper: Is the overall quality of the paper suitable?

Good

Is the length of the paper justified?

Yes

Should the paper be seen by a specialist statistical reviewer?

No

Do you have any concerns about statistical analyses in this paper? If so, please specify them explicitly in your report.

No

It is a condition of publication that authors make their supporting data, code and materials available - either as supplementary material or hosted in an external repository. Please rate, if applicable, the supporting data on the following criteria.

Is it accessible?

Yes

Is it clear?

Yes

Is it adequate?

Yes

Do you have any ethical concerns with this paper?

No

Comments to the Author

The manuscript by Blagrove et al. with the title Potential for Zika virus transmission by mosquitoes in temperate climates has been hugely improved. I'm very pleased with the effort that is put into this new version. All major comments have been addressed. There now is a good structure in the main manuscript and in the supplementary materials. Also, there is a better link in the introduction and discussion about other viruses and their interaction with temperature. Further, model construction and assumptions are better explained in the main text and the link between vector presence and calculated risk is clearly explained. Although I would have liked to see it removed, I understand that the authors chose to keep the vector competence data in the supplementary data of the manuscript.

Minor comments:

1. Line 89: spelling of *Ae. albopictus*.
2. Line 108: change into 30 x 30 x cm (30 cm³ is really small...) and move info on BugDorm to Line 103.
3. Line 112-123: info missing on materials used (companies, countries), for example on the Hemotek feeding system, and containers, DMEM and FCS. Is FCS Fetal Calf Serum or should FCS be FBS (Fetal Bovine Serum) instead? I would give full names besides the abbreviations.
4. Line 142-146: rephrase without the large sentence between brackets.

Supplementary materials

1. Table S2: I still don't quite understand what these values mean. If they are qPCR data, are these ct or cq values?
2. Table S3: I think the table text should be placed above the table.

Decision letter (RSPB-2020-0119.R2)

05-Jun-2020

Dear Dr Blagrove

I am pleased to inform you that your manuscript RSPB-2020-0119.R2 entitled "Potential for Zika virus transmission by mosquitoes in temperate climates" has been accepted for publication in Proceedings B.

The referee and Associate Editor have recommended publication, but the referee has also suggested some minor revisions to your manuscript. Therefore, I invite you to respond to the referee's comments and revise your manuscript. Because the schedule for publication is very tight, it is a condition of publication that you submit the revised version of your manuscript within 7 days. If you do not think you will be able to meet this date please let us know.

Finally, I hope you and your co-authors are well in these challenging times.

Yours sincerely,
 Professor Loeske Kruuk
 Editor, Proceedings B
<mailto:proceedingsb@royalsociety.org>

Associate Editor:

Board Member: 1

Comments to Author:

Thank you for all of your efforts to edit the manuscript based on the reviewer suggestions and respond to their questions and comments. I think the manuscript is much improved. The final reviewer has noticed a few additional very minor issues that should be easily fixed.

Reviewer(s)' Comments to Author:

Referee: 4

Comments to the Author(s)

The manuscript by Blagrove et al. with the title Potential for Zika virus transmission by mosquitoes in temperate climates has been hugely improved. I'm very pleased with the effort that is put into this new version. All major comments have been addressed. There now is a good structure in the main manuscript and in the supplementary materials. Also, there is a better link in the introduction and discussion about other viruses and their interaction with temperature.

Further, model construction and assumptions are better explained in the main text and the link between vector presence and calculated risk is clearly explained. Although I would have liked to see it removed, I understand that the authors chose to keep the vector competence data in the supplementary data of the manuscript.

Minor comments:

1. Line 89: spelling of *Ae. albopictus*.
2. Line 108: change into 30 x 30 x cm (30 cm³ is really small...) and move info on BugDorm to Line 103.
3. Line 112-123: info missing on materials used (companies, countries), for example on the Hemotek feeding system, and containers, DMEM and FCS. Is FCS Fetal Calf Serum or should FCS be FBS (Fetal Bovine Serum) instead? I would give full names besides the abbreviations.
4. Line 142-146: rephrase without the large sentence between brackets.

Supplementary materials

1. Table S2: I still don't quite understand what these values mean. If they are qPCR data, are these ct or cq values?
2. Table S3: I think the table text should be placed above the table.

Decision letter (RSPB-2020-0119.R3)

11-Jun-2020

Dear Dr Blagrove

I am pleased to inform you that your manuscript entitled "Potential for Zika virus transmission by mosquitoes in temperate climates" has been accepted for publication in Proceedings B.

Open Access

Paper charges

Sincerely,

Proceedings B
mailto:proceedingsb@royalsociety.org

Appendix A

Associate Editor

Board Member: 1

Comments to Author:

Thank you for submitting your manuscript “Potential for Zika virus transmission by mosquitoes in temperate climates” To Proceedings B. I have now received three reviews and evaluated the manuscript myself. While we all find the topic interesting and the manuscript generally written, several important issues have been raised. In particular, important areas seem to be oversimplified and more details and nuance are required. For example, reviewer 1 highlights several assumptions implicit to the R_0 calculations that should be teased apart or at least discussed. Similarly, reviewers 2 and 3 raised several more detailed nuances relevant to ZIKV transmission that need to be addressed, such as copy numbers, and how the study conclusions could be improved by additional experiments.

We thank the editor and the reviewers for their constructive comments. We added details related to the R_0 calculation and we added additional model analysis about the potential length of the ZIKV transmission season (LTS in months) as requested by the reviewers. Additional lab experiments have also been conducted in order to answer the reviewers’ concerns about viability of virus after replicating in the mosquito at 19°C, these experiments demonstrate that the virus is viable after replication at the lower temperatures, supporting our conclusions of transmission risk in temperate conditions.

Reviewer(s)' Comments to Author:

Referee: 1

Comments to the Author(s)

Summary: Blagrove et al. determined the vector competence of 4 species of mosquitoes from Europe for ZIKV from Brazil at one virus dose. Two species [*Cx. pipiens pipiens* and *Culiseta anulata*] did not become infected. Two *Aedes* were susceptible and the duration of the extrinsic incubation period [EIP = time from exposure to transmission] was measured at 6 temperatures and 7 time points. The primary vector of ZIKV globally [*Ae. aegypti*] was not included in this paper. No transmission was detected at 17°C supporting their calculated minimum EIP temperature threshold. Based on these data and a series of modelling assumptions, R_0 was modeled for different temperatures and used to map the distribution of risk of ZIKV transmission globally.

Thoughts: Specific comments have been added directly to the attached files and are summarized below.

Thank you for the specific comments on the manuscript, as stated the main ones are summarised below and have direct responses included here, the other more minor comments on the manuscript have been addressed accordingly (see track-changes version attached).

1. Proportion transmitting. This estimate was based on the number of females that imbibed a blood meal and survived to be tested rather than the number infected. I wonder if it is reasonable to estimate the time to transmission based, in part, on uninfected females? To accommodate these data, the EIP calculations were based on 10% with salivary secretions positive at each time point, even though <10 females were used at some time points [Suppl Table 5]? Data presented in Fig. 1 indicate that <40% of the females tested were ever positive for body infection. Examining the raw data, the authors may have biased their results by using fewer females at earlier time points and greater numbers at later time points when transmission was anticipated.

Our estimate of time to transmission was based on infected females, we used saliva-infected and saliva-uninfected-but body-infected individuals to do this with the assumption that if the body is infected it has the potential to become saliva-infected. We do not use completely uninfected mosquitoes in our estimation of time to infection.

Our use of smaller numbers for earlier time points (lower) was intentional. This was because these individuals are not as informative (0 and 5 days, which cannot have positive saliva), and given the limitation in the number of mosquitoes which can be worked with at any one time it was more useful to 'save' mosquitoes for the more informative later time points. We have added a justification for this in the supplementary materials.

Text added to supplementary materials (see Table S3): "Note that the number of mosquitoes used on each day was not equal, given experimental constraints on the numbers of mosquitoes that can be infected at one time, fewer mosquitoes were used on less informative time points (0 and 5 days) compared to later time points."

2. R_0 assumptions [Suppl Table 3].

$a(T)$ assumed it takes twice as long for albopictus to digest a blood meal and refeed as aegypti? Does this assume gonotrophic concordance?

*We are not making assumptions about egg laying and feeding intervals (gonotrophic concordance). We limited ourselves to assumptions about feeding intervals. Specifically, we halved biting rates ($a(T)$) for *Ae. albopictus* because the feeding interval takes about twice as long for *Ae. albopictus* with respect to *Ae. aegypti* (about 57% in the study by Scott et al., 2000, and we used the analytical function provided by Liu-Helmersson et al., 2014). Note that we used a similar parameter setting in our Zika R_0 global model paper that was published in PNAS in 2017 (Caminade et al., 2017).*

b was rescaled between 0 and 1 rather than use the values estimated?

In our study, we mostly focused on infection experiments in order to derive EIP estimates. Other parameters have been derived from the published literature. Importantly, we used standardized R_0 estimates (std $R_0(T)$) e.g. R_0 was rescaled to range between [0-1] by dividing R_0 values by the simulated maximum. Consequently, std $R_0(T)$ ranges between 0 and 1 by design and all constant epidemiological parameters (b , beta, m etc) should not impact the shape of the standardized $R_0(T)$ curve. We basically followed the same approach than Tesla et al., 2018 & Ryan et al. 2019. Ryan, et al. used this approach because some epidemiological parameters are highly heterogeneous and difficult to derive from field based studies. The best example is m (vector to host ratio), this parameter can vary greatly between urban and rural environments – and field based studies can yield very different results, as m can be estimated by trap catches or human landing catches. This is partly why Ryan et al. and others used a R_0 standardization method, and we followed the same approach in our study.

Beta was also rescaled rather than use vector infection rates. Here a value of 1 would exceed the values in their experiments?

Addressed in response to previous comment. Note that we use 0.5 for b and beta (median value based on Rocklov et al., 2016, this has been updated in Table S7), but as aforementioned these constant parameters should not impact the shape of the standardized $R_0(T)$ curve anyway.

Updated item in Supplementary Materials (Table S7):

Table S7: Standardized $R_0(T)$ model parameter settings. *denotes parameters which are dynamically simulated in space and time over the whole time period. T stands for temperature.

Symbol	Description	Constant/Formula	Comments	Ref
*a(T)	Biting rate (per day)	$a=(0.0043T + 0.0943)/2$	The linear dependency to temperature was based on estimates for Ae. aegypti in Thailand. Biting rates for Ae. albopictus were halved based on observed feeding interval data	(6,7)
b	Transmission probability - vector to host (0-1)	b=0.5	Median value of ref [13]. Note that constant parameter values should not impact the shape of the final standardized $R_0(T)$ estimate (std $R_0(T)$ was rescaled to range between 0-1, see Fig 2d)	(13)
β	Transmission probability - host to vector (0-1)	$\beta=0.5$	Median value of ref [13]. Note that constant parameter values should not impact the shape of the final standardized $R_0(T)$ curve (std $R_0(T)$ was rescaled to range between 0-1, see Fig 2d)	(13)
* $\mu(T)$	Mortality rate (0-1 per day)	$\mu=1/(1.1+\exp(-4.04+0.576T))+ 0.11883$ if $T < 15^\circ\text{C}$ $\mu=0.000339T^2-0.0189T+0.336$ if $15^\circ\text{C} \leq T < 26.3^\circ\text{C}$ $\mu=1/(1.065+\exp(32.2-0.92T))+ 0.073079$ if $T \geq 26.3^\circ\text{C}$	Mortality rate was derived for both mosquito vectors from published estimates based on both laboratory and field data. Due to discontinuity around the different temperature thresholds these estimates have been updated.	(8)
*u(T)	Extrinsic Incubation Rate (days)	$1/u = \text{eip_albo} = -1.0757T+43.0342$ $\text{eip_detr} = -1.07567T+46.025$	EIP(T) was estimated based on our updated laboratory data (see Figure 2). EIP(T) for Ae. detritus is also shown for comparison.	This study
m	Vector to host ratios	m= 28.2	Note that constant parameter values should not impact the shape of the final standardized $R_0(T)$ curve (std $R_0(T)$ was rescaled to range between 0-1, see Fig 2d)	(2)
r	Recovery rate (per day)	r=1/7	1 week viraemia is a common value for ZIKV.	(14)

m and to some extent b failed to consider the probability of human blood feeding which can be quite low when other hosts are plentiful? In some areas, for example, albopictus is considered a major vector of dog heartworm.

See former comments about constant parameters and vector to host ratio. We are aware that Ae. albopictus can transmit dirofilaria immitis to dogs. We used constant parameters for m, b and beta following Ryan et al., 2019 & Tesla et al., 2018. We did not use human population estimates to derive m. Ae. albopictus can also feed on several animal hosts (small mammals, birds...) as noted by the reviewer.

3. R_0 estimates. These remained <1 for most areas which would indicate the virus should not persist at the annual temperatures used? Would it be of interest to use min and max temperatures for selected geographical areas? Historically DENV and YFV caused major summer outbreaks at relatively northern latitudes, but did not persist.

*See former comments. We used standardized R_0 estimates following Ryan et al., 2019 and Tesla et al. 2018, which ranges between 0 and 1 by construction. Importantly, we have added new analysis in order to characterize the length of the ZIKV transmission season (as requested by other reviewers). We focus on standardized R_0 values exceeding 0.295 (this corresponds to the minimum temperature for ZIKV transmission by *Ae. albopictus* observed in the lab e.g. $T = 19^\circ\text{C}$) to derive months suitable for transmission (as requested by other reviewers). Originally Ryan et al. & Tesla et al. focused on standardized $R_0 > 0$ – but we think this threshold might overestimate the surface suitable for ZIKV transmission. Thus, we use the same standardized $R_0(T)$ approach but we decided to look at standardized $R_0(T)$ values for which we know that the mosquitoes become infectious in the lab (e.g. above 19°C). We did not investigate the impact of T_{\min} or T_{\max} in our model framework (but we partly addressed this issue on the mosquito vector side in Metelmann et al., 2019).*

This additional approach is now detailed in “Derivation of the length of the ZIKV transmission season (LTS)” in Supplementary Materials. Note that we moved all model details into Supplementary Materials to improve readability following other reviewer’s suggestions.

*Updated text in Supp. Materials [L141]: Derivation of the length of the ZIKV transmission season (LTS) – *Ae. albopictus* model*

To investigate seasonality in risk, we calculated the length of the transmission season (LTS) based on our standardized $R_0(T)$ estimates. If $\text{std } R_0(T) > 0.295$ (corresponding to $T=19^\circ\text{C}-33.1^\circ\text{C}$ e.g. orange and red colours on Fig. 2d) for a particular location and month, we assumed that temperature conditions were suitable for ZIKV transmission (so we assign 1 to particular location and month); conversely if $\text{std } R_0(T) \leq 0.295$, we assumed that no transmission occurred (so we assigned 0 to that particular location and month). We then sum months at risk on annual basis to derive LTS which ultimately ranges between 0-12 months.

Updated text in Abstract [L44]: “Using these data, we generated standardized basic reproduction number R_0 -based risk maps and we derived estimates for the length of the transmission season.”

Updated text in Results [L149]: “and we also discuss potential changes in the simulated length of the ZIKV transmission season (LTS thereafter) at global scale.”

*Updated text in Results [L173]: “Additionally, ZIKV could theoretically be transmitted all year long by *Ae. albopictus* in the tropics (South America, Africa and Asia, see Fig S5). In temperate regions, the length of the transmission season (LTS) is shorter and varies between 1 and 6 months (Fig. S5).”*

Updated text in Results [L178]: “ZIKV could be transmitted all year round in Central Africa (Fig. S5b). LTS then decreases as a function of the latitude over the African continent, with 1-3 months length of the transmission season simulated over the northern fringe of the Sahel and over temperate regions of Southern Africa (Fig. S5b).”

Updated text in Results [L199]: “... in particular over the southern tip of Florida and Texas where autochthonous transmission of ZIKV was reported in 2016-17 (22–24) and where ZIKV transmission could extend between 7 and 12 months (Fig. S5d).”

Updated text in Results [L209]: “...and where simulated LTS exceeds 6 months (Fig. S5f)...”

*Updated text in Discussion [L362]: “Our estimates of the potential length of the ZIKV transmission season by *Ae. albopictus* match recently published findings. In our model, seasonal ZIKV transmission could occur for 4-5 months over southern Europe; this finding is consistent with another R_0 model estimate [about 4 months transmission shown in (39)]. The pattern and magnitude of the simulated length of the ZIKV transmission season over northern and southern America is also consistent with (13,30), who also showed potential year-round ZIKV transmission over Central America, the Caribbean and the northern half of South*

America. Simulated LTS (about 7-8 months) over the southern provinces of China (Guangdong and Guangxi) also corresponds to observed seasonal transmission of dengue fever over these regions (43,44)."

New risk maps have been added (LTS estimates) in Supp. Materials:

Figure S5: Simulated length of the ZIKV transmission season (LTS in months) based on observed rainfall and temperature data (1980-2010) for a) the globe, b) Africa, c) South America, d) North America, e) Europe and f) Asia.

Figure S7: Simulated length of the ZIKV transmission season by *Ae. albopictus* over Europe. This is carried out for the 2050s (2040-59 average), left column (a, b, c, d) and the 2080s (2070-89 average), right column (e, f, g, h), from the lowest (RCP2.6, top, a, e) to the highest (RCP8.5, bottom, d, h) emission scenario.

Figure S8: Simulated length of the ZIKV transmission season by *Ae. albopictus* over North America. This is carried out for the 2050s (2040-59 average), left column (a, b, c, d) and the 2080s (2070-89 average), right column (e, f, g, h), from the lowest (RCP2.6, top, a, e) to the highest (RCP8.5, bottom, d, h) emission scenario.

Figure S9: Simulated length of the ZIKV transmission season by *Ae. albopictus* over Asia. This is carried out for the 2050s (2040-59 average), left column (a, b, c, d) and the 2080s (2070-89 average), right column (e, f, g, h), from the lowest (RCP2.6, top, a, e) to the highest (RCP8.5, bottom, d, h) emission scenario.

Figure S10: Simulated length of the ZIKV transmission season by *Ae. albopictus* over Africa. This is carried out for the 2050s (2040-59 average), left column (a, b, c, d) and the 2080s (2070-89 average), right column (e, f, g, h), from the lowest (RCP2.6, top, a, e) to the highest (RCP8.5, bottom, d, h) emission scenario.

Figure S11: Simulated length of the ZIKV transmission season by *Ae. albopictus* over South America. This is carried out for the 2050s (2040-59 average), left column (a, b, c, d) and the 2080s (2070-89 average), right column (e, f, g, h), from the lowest (RCP2.6, top, a, e) to the highest (RCP8.5, bottom, d, h) emission scenario.

References

- Caminade C., J. Turner, S. Metelmann, J.C. Hesson, M.S.C. Blagrove, T. Solomon, A.P. Morse, M. Baylis (2017). Global risk model for vector-borne transmission of Zika virus reveals the role of El Niño 2015. *Proceedings of the National Academy of Sciences*, 114(1): 119-124
- Liu-Helmersson J, Stenlund H, Wilder-Smith A, Rocklöv J (2014) Vectorial Capacity of *Aedes aegypti*: Effects of Temperature and Implications for Global Dengue Epidemic Potential. *PLoS ONE* 9(3): e89783.
- Metelmann S., Caminade C., Jones A.E., Medlock J.M., M. Baylis and A.P. Morse (2019). The UK's suitability for *Aedes albopictus* in current and future climates. *Journal of the Royal Society Interface*. early version published on line. <https://doi.org/10.1098/rsif.2018.0761>
- J Rocklöv, MB Quam, B Sudre, et al. Assessing seasonal risks for the introduction and mosquito-borne spread of Zika virus in Europe. *EBioMedicine*, 9 (2016), pp. 250-256
- Ryan SJ, Carlson CJ, Mordecai EA, Johnson LR (2019) Global expansion and redistribution of *Aedes*-borne virus transmission risk with climate change. *PLoS Negl Trop Dis* 13(3): e0007213
- Scott TW, Amerasinghe PH, Morrison AC, Lorenz LH, Clark GG, Strickman D, et al. Longitudinal studies of *Aedes aegypti* (Diptera: Culicidae) in Thailand and Puerto Rico: blood feeding frequency. *J Med Entomol* [Internet]. 2000 Jan [cited 2018 Jan 30];37(1):89–101. Available from: <http://www.ncbi.nlm.nih.gov/pubmed/15218911>
- Tesla B, Demakovskiy LR, Mordecai EA, Ryan SJ, Bonds MH, Ngonghala CN, Brindley MA, Murdock CC. 2018 Temperature drives Zika virus transmission: evidence from empirical and mathematical models. *Proc. R. Soc. B* 285: 20180795. <http://dx.doi.org/10.1098/rspb.2018.0795>

	19°C						0.109 1 NA		0.160 1 NA
	21°C					1.209 2 1.634	0.392 1 NA	0.181 1 NA	0.193 2 0.129
	24°C					0.090 1 NA	0.507 1 NA	0.299 3 0.351	0.417 2 0.463
	27°C				0.170 1 NA	0.556 2 0.575	0.313 3 0.334	0.226 2 0.109	
	31°C				0.576 1 NA	0.067 2 0.024			

Average = 0.3599, Standard deviation = 0.4832

Ae. albopictus

		Days post infection							
		0	5	7	10	14	17	21	28
Temperature	17°C								
	19°C						0.655 1 NA	1.951 1 NA	2.015 2 2.392
	21°C				0.181 1 NA		0.507 1 NA	0.863 3 0.950	1.410 2 1.808
	24°C				0.614	0.394	1.192	0.475	1.639

					1 NA	2 0.431	2 1.256	1 NA	4 1.703
	27°C				0.699 1 NA	0.460 2 0.164	2.689 1 NA	1.380 3 1.283	1.707 4 1.047
	31°C			0.090 1 NA	1.167 1 NA	0.472 2 0.147	1.432 4 1.689	1.338 2 0.695	2.842 5 3.363

Average = 1.3541, Standard deviation = 1.5382

Updated text in Results [L125]: “We also demonstrate that the titre of ZIKV in the saliva of *Ae. albopictus* is 3.8x higher than in the saliva of *Oc. detritus* ($P < 0.00001$) across all saliva-positive individuals (see Table S5).”

Updated text in Discussion [L280]: “Our data show that *Ae. albopictus* is a more competent vector than *Oc. detritus*, both in terms of overall proportion of infectious individuals, and in the titre of the virus in the saliva being significantly higher in *Ae. albopictus* (Table S5).”

I also would recommend discussion of seasonal transmission. There are large areas, where the continuous transmission is very unlikely in future decades but when suitable vector is present, the temperatures might allow seasonal transmission, which is not considered at all in the manuscript.

Very good point. We have added additional model analysis on the potential length of the ZIKV transmission season (LTS in months) as requested by other reviewers. To define suitable months for ZIKV transmission by *Ae. albopictus*, we look at months where standardized $R_0(T)$ exceeds 0.295 (this threshold corresponds to the minimum temperature for transmission e.g. 19°C in the lab).

This additional approach is now detailed in “Derivation of the length of the ZIKV transmission season (LTS)” in Supplementary Materials. Note that we moved all model details into Supplementary Materials to improve readability.

Updated text in Supp. Materials [L141]: Derivation of the length of the ZIKV transmission season (LTS) – *Ae. albopictus* model

To investigate seasonality in risk, we calculated the length of the transmission season (LTS) based on our standardized $R_0(T)$ estimates. If $\text{std } R_0(T) > 0.295$ (corresponding to $T=19^\circ\text{C}-33.1^\circ\text{C}$ e.g. orange and red colours on Fig. 2d) for a particular location and month, we assumed that temperature conditions were suitable for ZIKV transmission (so we assign 1 to particular location and month); conversely if $\text{std } R_0(T) \leq 0.295$, we assumed that no transmission occurred (so we assigned 0 to that particular location and month). We then sum months at risk on annual basis to derive LTS which ultimately ranges between 0-12 months.

Updated text in Abstract [L44]: “Using these data, we generated standardized basic reproduction number R_0 -based risk maps and we derived estimates for the length of the transmission season.”

Updated text in Results [L149]: “and we also discuss potential changes in the simulated length of the ZIKV transmission season (LTS thereafter) at global scale.”

Updated text in Results [L173]: “Additionally, ZIKV could theoretically be transmitted all year long by Ae. albopictus in the tropics (South America, Africa and Asia, see Fig S5). In temperate regions, the length of the transmission season (LTS) is shorter and varies between 1 and 6 months (Fig. S5).”

Updated text in Results [L178]: “ZIKV could be transmitted all year round in Central Africa (Fig. S5b). LTS then decreases as a function of the latitude over the African continent, with 1-3 months length of the transmission season simulated over the northern fringe of the Sahel and over temperate regions of Southern Africa (Fig. S5b).”

Updated text in Results [L199]: “... in particular over the southern tip of Florida and Texas where autochthonous transmission of ZIKV was reported in 2016-17 (22–24) and where ZIKV transmission could extend between 7 and 12 months (Fig. S5d).”

Updated text in Results [L209]: “...and where simulated LTS exceeds 6 months (Fig. S5f)...”

Updated text in Discussion [L362]: “Our estimates of the potential length of the ZIKV transmission season by Ae. albopictus match recently published findings. In our model, seasonal ZIKV transmission could occur for 4-5 months over southern Europe; this finding is consistent with another R_0 model estimate [about 4 months transmission shown in (39)]. The pattern and magnitude of the simulated length of the ZIKV transmission season over northern and southern America is also consistent with (13,30), who also showed potential year-round ZIKV transmission over Central America, the Caribbean and the northern half of South America. Simulated LTS (about 7-8 months) over the southern provinces of China (Guangdong and Guangxi) also corresponds to observed seasonal transmission of dengue fever over these regions (43,44).”

New risk maps have been added (LTS estimates) in Supp. Materials:

Figure S5: Simulated length of the ZIKV transmission season (LTS in months) based on observed rainfall and temperature data (1980-2010) for a) the globe, b) Africa, c) South America, d) North America, e) Europe and f) Asia.

Figure S7: Simulated length of the ZIKV transmission season by *Ae. albopictus* over Europe. This is carried out for the 2050s (2040-59 average), left column (a, b, c, d) and the 2080s (2070-89 average), right column (e, f, g, h), from the lowest (RCP2.6, top, a, e) to the highest (RCP8.5, bottom, d, h) emission scenario.

Figure S8: Simulated length of the ZIKV transmission season by *Ae. albopictus* over North America. This is carried out for the 2050s (2040-59 average), left column (a, b, c, d) and the 2080s (2070-89 average), right column (e, f, g, h), from the lowest (RCP2.6, top, a, e) to the highest (RCP8.5, bottom, d, h) emission scenario.

Figure S9: Simulated length of the ZIKV transmission season by *Ae. albopictus* over Asia. This is carried out for the 2050s (2040-59 average), left column (a, b, c, d) and the 2080s (2070-89 average), right column (e, f, g, h), from the lowest (RCP2.6, top, a, e) to the highest (RCP8.5, bottom, d, h) emission scenario.

Figure S10: Simulated length of the ZIKV transmission season by *Ae. albopictus* over Africa. This is carried out for the 2050s (2040-59 average), left column (a, b, c, d) and the 2080s (2070-89 average), right column (e, f, g, h), from the lowest (RCP2.6, top, a, e) to the highest (RCP8.5, bottom, d, h) emission scenario.

Figure S11: Simulated length of the ZIKV transmission season by *Ae. albopictus* over South America. This is carried out for the 2050s (2040-59 average), left column (a, b, c, d) and the 2080s (2070-89 average), right column (e, f, g, h), from the lowest (RCP2.6, top, a, e) to the highest (RCP8.5, bottom, d, h) emission scenario.

I suggest downgrading the conclusion of assessing the vector competence of four mosquito species as two of them were applicable in such a small number that comprehensive experiments could not be conducted.

Agreed, whilst we do discuss the small numbers already, we have further *downgraded various sections accordingly.*

Modified text in Abstract [L37]: "...we assessed the vector competence of two common temperate mosquito species/strains, Aedes albopictus and Ochlerotatus detritus, which were both found to be competent for ZIKV. In addition, we also assessed smaller numbers of two other species, Cx. pipiens pipiens and Cs. annulata."

Modified text in Results [L127]: "We found no evidence that field-obtained Cx. pipiens pipiens or Cs. annulata were competent for ZIKV after 17 days at 21°C (Table S6). Relatively small numbers of these two species were tested due to practical limitations (extremely low feeding rate for Cx. pipiens pipiens and difficulty in collecting large numbers of wild larvae for Cs. annulata). Because of the small numbers, the individuals were tested at 21°C only; this temperature was chosen to provide a balance between excessive mortality at high temperatures and extremely long EIPs at lower temperatures (which leads to higher mortality prior to the EIP being reached). Because of the low numbers, we could not conclude that these species are not competent for ZIKV.."

Modified text in Discussion [L266]: "We demonstrated that both wild populations of Oc. detritus (UK), and newly colonised Ae. albopictus (VERANO colony, Italy) are competent to transmit Zika virus. Whilst we found no evidence of ZIKV transmission in Cx. pipiens pipiens or Cs. annulata, our sample size was too small to conclude that they are not competent."

In the experiments it was found that in lower temperatures it took longer for the virus to be detected from the saliva. When the timeline from blood meal to positive saliva exceeds 20 days, it would be interesting to discuss the significance in practice, as what is the natural lifespan for these mosquito species in the nature.

In the field Ae. albopictus can survive for about a month (in a lab setting, it can survive several months), so when the EIP exceeds 20 days the theoretical risk of transmission decrease (if you only consider one mosquito generation). Mortality(T°) is already taken into account in our standardized $R_0(T)$ model framework – and we used the mortality scheme derived from both lab and field data by Brady et al. 2013. When the EIP reaches about 30 days in our model, this corresponds to $T=12.1^{\circ}\text{C}$ (linear model) and standardized $R_0 = 0.064$ (so this value falls well below our "suitability" thresholds).

The spelling of the manuscript needs quite a lot of revision. Both the main text and supplementary material include two different fonts. The terms with capitals, (Southern, Northern, Central etc) should be corrected throughout the text.

We have carefully checked typos and font issues in the text.

Specific comments

Abstract

Line 47 change japan to Japan

Done

Introduction

Lines 52-53 "Though infection is asymptomatic in 80% of cases (1), a small proportion of patients develop the autoimmune condition Guillain-Barré syndrome (2)". This should be rephrased as now it reads as ZIKV infection is either asymptomatic or it manifests as GBS.

Clarified, this sentence now reads as follows:

Updated text in Introduction [L54]: “Though infection is asymptomatic in 80% of cases (1), a small proportion of patients develop clinical symptoms, including the autoimmune condition Guillain-Barré syndrome (2)”.

Line 56 World Health Organisation, please change to World Health Organization

Corrected

Line 67 Please clarify what you mean by “pathogen to develop”

Clarified, this sentence now reads as follows:

Modified text in Introduction [L68]: “... at low temperatures there may be insufficient time for the pathogen to replicate, invade the salivary gland and become infectious within the lifespan of the vector, imposing a critical threshold temperature for virus transmission (11,12).”

Lines 71-73 “With large human populations in temperate climates it is also important to estimate how ZIKV transmission occur in such climates today, and with future climate change.” Do authors mean that already occurs or that could occur? Please clarify

Line 73 “future climate change” is probably not the best term as climate change itself occurs already. I suggest to change it to in future climate for example.

We meant “both currently and under predicted future climate scenarios”; Note that autochthonous transmission of ZIKV by Ae. albopictus has recently been reported in the city of Hyères, near Marseille in south-eastern France (see recent paper by Brady & Hay, 2019 and recent report by ECDC [<https://www.ecdc.europa.eu/en/publications-data/rapid-risk-assessment-zika-virus-disease-var-department-france>]). Local transmission of dengue virus by the same mosquito vector was also observed in Spain, southern France and Croatia over the past few years [https://www.ecdc.europa.eu/sites/default/files/documents/RRA-dengue-in-Spain-France_10Oct2019.pdf].

Modified text in Introduction: [L74]: “With large human populations living in temperate climates it is also important to estimate the extent of ZIKV transmission in such climates, both currently and also under predicted future climate scenarios.”

Line 84 RCPx, what does the x stand for? Should it be s?

Yes it should be RCPs (plural form), now corrected. The number following “RCP” (2.6, 4.5, 6.0 & 8.5) provides an estimate of the excess of energy (longwave radiation in W/m²) in the lower layer of the atmosphere by 2100 (2.6W/m²; 4.5W/m², 6W/m² and 8.5W/m²).

Results

Lines 88-89 “Zika virus was detected in the saliva of both Ae. albopictus (VERANO colony) and Oc. detritus at all temperatures from 19oC to 31oC, but not at 17oC” Please add copy numbers, this would be nice to see in a table. But it would clarify the effect of temperature on the transmission.

Mosquito number estimates are given in Table S3 (in Supp. Materials).

Line 152 change Tropics to tropics

Done

Line 154 change central Africa to Central Africa

Lower case is correct, corrected.

Line 157 should three dots after Malaysia be removed?

Removed

Line 163 “parts of Turkey” please specify which parts

We meant “near Istanbul and at the southern coastal border with Syria”. Interestingly Ae. albopictus has been recently reported near Istanbul and it is already established in the Latakia region <https://www.ecdc.europa.eu/en/publications-data/aedes-albopictus-current-known-distribution-august-2019>.

The text in results now reads as follows [L184]: “...with hotspots over the eastern coasts of Spain, Sardinia, Sicily, northern Italy, southern France, the coasts of Croatia and Albania, Greece and, in Turkey, near Istanbul and at the southern coastal border with Syria (Fig. 3e)”

Lines 168-171 “Examples include Ravenna in the Emilia-Romagna region in Italy where a chikungunya outbreak occurred in 2007; nearby Cadiz in southern Spain and on the eastern Catalan coasts with a few reported and suspected cases of dengue in 2018; south-eastern France where cases of dengue were reported in 2010, 2013, 2014 and 2015 (20)” Please modify this sentence.

Clarified, this sentence now reads as follows:

Updated text in Results [L190]: “Examples include the 2007 CHIKV outbreak in Ravenna in the Emilia-Romagna region in Italy; suspected cases of dengue fever in Cadiz in southern Spain in 2018; and the reported cases of dengue in south-eastern France in 2010, 2013, 2014 and 2015 (20)”

Line 184 Correct to temperature only- effect

Done

Line 187 change central Australia to Central Australia

Central Australia is not a proper noun, therefore lower case ‘c’ is correct.

Line 215 “However, there is a decrease in simulated $R_0(T)$ over Brazil in future for the 2080s (Fig. S6h).” some discussion on the reasons behind this would be good add.

This decrease is related to an increase in future temperatures (RCP8.5 – 2080s) over this region. This result is now briefly mentioned in the text as “not shown”. This result is also consistent with the generic arbovirus RCP8.5 scenario for the 2080s published by Ryan et al., 2019, but not as extreme as transmission suspiciously drops down to 0 over Brazil and parts of Africa in their study (see bottom right panel of Fig. 2 in Ryan et al., 2019).

Text modified in Results [Lxx-Lyy]: “However, there is a decrease in simulated $R_0(T)$ (Fig. 4) and a shortening of the ZIKV transmission season over Brazil in future for the 2080s (Fig. S11h), due to high temperatures (not shown).”

Discussion

Line 256 Please clarify what does “high laboratory competence” mean

Clarified

Text modified in Discussion, [L282]: “However, the observation that Oc. detritus had, nevertheless, a considerable proportion of infectious individuals suggests that there may be other unknown temperate species which are also highly competent for ZIKV and may have significant impacts on risk in temperate regions.”

Line 261 please refer to the study not only to a reference number in the text.

Corrected

Line 285 “in contrast to USA and temperate Asia” I would suggest referring to “in contrast to the maps of USA and Asia”

Corrected

Lines 292-293 “Ae. albopictus is not a major issue in Africa yet”, please retype this. This refers to the presence of Ae. albopictus’

This statement has been deleted and the text modified.

Text modified in Discussion [L331]: “ZIKV transmission by Ae. albopictus could increase over high altitude regions.”

Line 293 should altitude regions be high altitude regions?

We decided to leave this as it stands in the text. “high altitude regions” could be misleading for the reader (high altitude regions usually refers to the Himalayas e.g. Nepal & Tibetan plateau).

Line 309 “Locally they...” Does this refer to both Ae. albopictus and aegypti or other species? Please clarify

This statement refers to both Ae. albopictus and Ae. aegypti.

Text modified in Discussion [L348]: “Locally they can both achieve...”

Line 316 “Our model shows only the risk as a result of climate and vector species” please retype this. For example: as a result of the effect of climate and vector species

Updated.

This sentence now reads as follows in Discussion [L370]: “Our model only considers the impact of climate on the mosquito vector potential to transmit ZIKV”.

Lines 323-324 “The interplay between these, and other, factors will likely have a large effect on the risk to a population.” Please retype.

In general, the validity of the models used should be discussed more thoroughly as now there is very little about it.

Updated. Caveats about our model framework and how our findings compare to other studies are now discussed further in discussion. In addition, we now compare our findings with recently published studies on the same topic, for example Ryan et al. 2019 & Tesla et al., 2018.

This sentence now reads as follows in Discussion [L378]: “The interplay between climatic and other external factors will likely have a large effect on the risk to a population.”

Caveats about our model framework are discussed in a paragraph in discussion [L370]: “ Our model only considers the impact of climate on the mosquito vector potential to transmit ZIKV. Such a model cannot account for other important factors including the number of infected travellers arriving, and the immunological history of the local population (45,46). Herd immunity is an important factor in risk and is expected to be low in naïve European, North American and temperate Asian populations. Conversely, antibody-dependent enhancement (ADE) of infection, most commonly associated with DENV, may enhance ZIKV viremia and therefore potentially increase the likelihood of mosquito acquisition of the virus (47). Such ADE may have been a significant driving force in non-flavivirus-naïve populations, but is likely to be reduced in temperate populations. The interplay between climatic and other external factors will likely have a large effect on the risk to a population..”

*Our results are also compared to other modelling studies in a paragraph in Discussion [L362]: “Our estimates of the potential length of the ZIKV transmission season by *Ae. albopictus* match recently published findings. In our model, seasonal ZIKV transmission could occur for 4-5 months over southern Europe; this finding is consistent with another R_0 model estimate [about 4 months transmission shown in (39)]. The pattern and magnitude of the simulated length of the ZIKV transmission season over northern and southern America is also consistent with (13,30), who also showed potential year-round ZIKV transmission over Central America, the Caribbean and the northern half of South America. Simulated LTS (about 7-8 months) over the southern provinces of China (Guangdong and Guangxi) also corresponds to observed seasonal transmission of dengue fever over these regions (43,44).”*

In results the authors state that: “If we remove the rainfall effect in our R_0 model (temperature only effect), the surface at risk increases significantly and to some extent unrealistically, with large values simulated in desert regions such as the Sahara (Fig. S3b), parts of the Middle East (Fig. S3a), the “yet there is no discussion about the rainfall model

*We used the MARA criterion which was originally described by Craig et al., 1999 for malaria (*An. gambiae*). This rainfall threshold basically masks out desert regions (the selected rainfall threshold is consistent with the Koppen Geiger classification for arid regions). Note that we employed the same rainfall criterion in Caminade et al., 2017. We raise this important point because the published study by Ryan et al., 2019 (only based on temperature) shows a potential 4-5 months transmission of “arboviruses” by *Ae. albopictus* in the middle of the Sahara desert, desert areas in the Middle east and central Australia (see Fig 1B in their study); this result is quite puzzling. Details about the rainfall model are given in Supplementary Materials (section “Derivation of $R_0(T)$ for ZIKV – *Ae. albopictus* model”).*

The limitations of the study should include discussion of the *Cx. pipiens pipiens* and *Cs. annulata* experiments as they were not comprehensive compared to other two species. One option would be to exclude data on these as obviously their vector competence must be studied further.

*Agreed, whilst we do discuss the small numbers already, we have further **downgraded various sections accordingly.***

*Modified text in Abstract [L37]: “...we assessed the vector competence of two common temperate mosquito species/strains, *Aedes albopictus* and *Ochlerotatus detritus*, which were both found to be competent for ZIKV. In addition, we also assessed smaller numbers of two other species, *Cx. pipiens pipiens* and *Cs. annulata*.”*

*Modified text in Results [L127]: “We found no evidence that field-obtained *Cx. pipiens pipiens* or *Cs. annulata* were competent for ZIKV after 17 days at 21°C (Table S6). Relatively small numbers of these two species were tested due to practical limitations (extremely low feeding rate for *Cx. pipiens pipiens* and difficulty in collecting large numbers of wild larvae for *Cs. annulata*). Because of the small numbers, the individuals were tested at 21°C only; this temperature was chosen to provide a balance between excessive mortality at high temperatures and extremely long EIPs at lower temperatures (which leads to higher mortality prior to the EIP being reached). Because of the low numbers, we could not conclude that these species are not competent for ZIKV..”*

Modified text in Discussion [L266]: “We demonstrated that both wild populations of Oc. detritus (UK), and newly colonised Ae. albopictus (VERANO colony, Italy) are competent to transmit Zika virus. Whilst we found no evidence of ZIKV transmission in Cx. pipiens pipiens or Cs. annulata, our sample size was too small to conclude that they are not competent.”

Materials and methods

Line 359 Please specify where the human blood for feeding the mosquitoes was achieved

Details added

Text added in Materials and Methods, [L415] “Blood was obtained from NHS transfusion service in Speke, Merseyside, UK. Blood was heparinized to prevent coagulation.”

Is RNA extraction with Trizol the best choice for saliva? In my experience I might suspect losing some RNA from the weak positive samples with this method.

Our HSE/SAPO licence for the laboratory requires all viruses to be deactivated in TriZol before removal from the room for extraction. We have previously seen no significant change in detection when testing this method on other viruses compared to other methods, but acknowledge this possibility.

Figures and tables

In all the map figures that are zoomed (so not including the world map), in supplementary material and the main text, the coordinates would be helpful especially in the boundaries of zoomed maps, in order to recognize easier where is e.g. equator (grids to the boundaries of these maps), so that it would possible and easier to estimate more accurate (latitude and longitude) risk area.

Latitude/longitude bounding boxes (including grid and coordinates) have been added to all maps in the main manuscript and in Supplementary Materials.

Figure 2 There is no referring to figure 2c in the text

We now refer to Fig. 2c in the text

Modified text in Results [L141]: “this does not significantly impact the shape of the resulting $R_0(T)$ function as mortality (U-shape curve, Fig. 2c) tends to determine the upper and lower suitability boundaries for standardized $R_0(T)$ ”.

Figure 3 Dots are very small, should be bigger to be able to see them

We decided to leave them relatively small, otherwise it would be difficult to see the coloured maps. Global maps (see Fig. 3a for example) complement the regional maps because we only show raw model outputs (without presence data).

Figure 4, parts a-h are not explained in the caption. Please add this.

The caption for Fig. 4 and for all climate change projections have been updated.

*Modified caption for Fig. 4: “**Fig. 4 – Annual mean standardized $R_0(T)$ - future projections for Ae. albopictus potential to transmit ZIKV.** This is carried out for the 2050s (2040-59 average), left column (a, b, c, d) and the 2080s (2070-89 average), right column (e, f, g, h), from the lowest (RCP2.6, top, a, e) to the highest (RCP8.5, bottom, d, h) emission scenario. The beige colour highlight standardized $R_0(T)$ values for which some ZIKV transmission by Ae. albopictus might occur in the laboratory [17-19°C]; orange, red and dark red colours*

correspond to standardized $R_0(T)$ values for which ZIKV transmission by *Ae. albopictus* occurs in our infection experiments [above 19°C, see Fig. 1a].”

Figure 4 line 233 “the most extreme” quite strong expression, prefer to high or higher emission scenario etc
Figure legends should consistently be placed under the figure and table legends above the table. There is variation in this now. Please correct.

Replaced by “highest emission scenario” when appropriate. We also replaced “mildest emission scenario” by “lowest emission scenario” for consistency. We have now placed all figure legends under the figures and all table legends above the tables (in the main manuscript and in Supp. Materials).

Supplementary material

Lines 76-78 “we employed the Global Precipitation Climatology Centre (GPCC) global rainfall data available at similar spatial and time resolution for the same time period (7).” Exactly same or just similar resolution? Please specify.

The same. Both monthly GPCC rainfall and CAMS temperature data are available on the same 0.5 x 0.5 degree grid from 1948-01->present (the only difference is the modulo in longitude – the CAMS data ranges between 0-360° longitude, GPCC data ranges between -180-180° longitude).

We updated the text in Supp. Materials [L155]: “rainfall data available at the same spatial and time resolution”.

Lines 78-79 “We calculated annual average for the 1980-2010 period, as advised by the World Meteorological Office guidelines” Please add a reference for this.

A reference to the related WMO 2017 report has been added in Supp. Materials.

Updated reference in Supp. Materials: “WMO Guidelines on the calculation of Climate Normals, WMO-No. 1203 (2017) ISBN 978-92-63-11203-3 available at [\[https://library.wmo.int/index.php?lvl=notice_display&id=20130#.Xfe4e-j7SUK\]](https://library.wmo.int/index.php?lvl=notice_display&id=20130#.Xfe4e-j7SUK).”

Lines 83-84 “for the ensemble mean of 5 GCM (hadgem2-es, ipsl-cm5a-lr, miroc-esm-chem, gfdl-esm2m, noresm1-m)” Could you explain why did you choose to make ensemble of these 5 GCMs specifically?

These 5 GCMs were pre-selected by the ISI-MIP team in Potsdam (PIK) because of daily climate data availability in the CMIP5 archive at the start of the ISI-MIP project (see Hempel et al., 2013 for further details). This climate model data was bias calibrated (using the WATCH observed climate data) by the same team at PIK in Potsdam and used in a very large multi-sectorial climate change impact assessment [about 116 publications now, see <https://www.isimip.org/outcomes/publications-overview-page/>].

Supplementary figure 6 legend “The beige colour depicts standardized $R_0(T)$ values for which some ZIKV transmission by *Ae. albopictus* might occur in the laboratory [17-19°C]”. Please clarify this.

*Based on our lab experiments – we showed that *Ae. albopictus* was able to become infectious at 19°C; while it was not able to become infectious at 17°C. Consequently, given our experimental setting, we cannot discard the hypothesis that some mosquitoes might become infectious between 17°C and 19°C (+/- a small temperature increment). This is why we show standardized $R_0(T)$ values corresponding to [17-19°C] in beige (some mosquitoes might become infectious –and we will have to do more experiments at low temperature to estimate such threshold). For values above 19°C we have solid evidence that *Ae. albopictus* mosquitoes become infectious.*

*We reworded this statement in Fig 2. And Fig. S4 captions: “...the beige colour depicts standardized $R_0(T)$ values [ranging between 0.201 and 0.295] for which *Ae. albopictus* might become infected by ZIKV in the*

laboratory [17-19°C]; orange, red and dark red colours depict temperatures above 19°C (ZIKV transmission by *Ae. albopictus* occurs in the laboratory above that temperature threshold, see Fig. 1a).“

Supplementary table 1. Retype model names: 1. Equation, for example

Supplementary table 5 the caption of the table is above the table as it should be but the legend is placed under the table. Please correct.

Corrected

References

Caminade C., J. Turner, S. Metelmann , J.C. Hesson, M.S.C. Blagrove, T. Solomon, A.P. Morse , M. Baylis (2017). *Global risk model for vector-borne transmission of Zika virus reveals the role of El Nino 2015. Proceedings of the National Academy of Sciences*, 114(1): 119-124

Brady OJ, Johansson MA, Guerra CA, Bhatt S, Golding N, Pigott DM, et al. *Modelling adult Aedes aegypti and Aedes albopictus survival at different temperatures in laboratory and field settings. Parasit Vectors [Internet]. 2013 Dec 12 [cited 2018 Jan 30];6(1):351. Available from: <http://parasitesandvectors.biomedcentral.com/articles/10.1186/1756-3305-6-351>*

Brady O.J. & Hay S. (2019). *The first local cases of Zika virus in Europe Lancet* 394(10213): 1991-1992.

Craig MH, Snow RW, le Sueur D (1999) *A climate-based distribution model of malaria transmission in sub-Saharan Africa. Parasitol Today* 15(3):105–111.

Hempel, S., Frieler, K., Warszawski, L., Schewe, J., and Piontek, F. (2013): *A trend-preserving bias correction – the ISI-MIP approach, Earth Syst. Dynam.*, 4, 219–236, <https://doi.org/10.5194/esd-4-219-2013>, 2013.

Ryan SJ, Carlson CJ, Mordecai EA, Johnson LR (2019) *Global expansion and redistribution of Aedes-borne virus transmission risk with climate change. PLoS Negl Trop Dis* 13(3): e0007213

Tesla B, Demakovskiy LR, Mordecai EA, Ryan SJ, Bonds MH, Ngonghala CN, Brindley MA, Murdock CC. 2018 *Temperature drives Zika virus transmission: evidence from empirical and mathematical models. Proc. R. Soc. B* 285: 20180795. <http://dx.doi.org/10.1098/rspb.2018.0795>

Comments to the Author(s)

The authors are describing an interesting study of ZIKV transmission, suggesting firstly that *Oc. detritus* may vector ZIKV in temperate regions at temperature 19C-24C and that *Ae. albopictus* could also transmit ZIKV at lower temperatures than previously reported (19, 21, 27C).

We thank the reviewer for these positive comments.

The result of *Oc. detritus* transmitting ZIKV is important, and its implications in the regions where this mosquito is present are significant. The authors state this in their discussion and point out the difficulty of vector control to mosquitoes that are breeding in natural water sources.

The main concerns in this study arise from interpretation of the obtained results, and considerations of what these results mean. The findings are interesting, but should be considered preliminary. Further evidence and more detailed information of the transmission temperature (and other) thresholds would be needed before applying the values to modelling and risk evaluations.

*Overall we tend to disagree with this comment. This study is a large scale multi-disciplinary study involving significant resource. We assembled a team of world leading entomologists, statisticians, epidemiologists and climate scientists. This enabled us to start a novel risk assessment and developing improved models. We plan to look in more details at the minimum temperature for transmission [17-19°C], so that we can tailor our future risk estimates. However, this is far beyond the scope of the response time for this manuscript and the study as we have designed it requires six smaller more finely calibrated incubators and over a year's worth of work. Critically we identified a minimum temperature for ZIKV transmission by *Ae. albopictus* (between 17-19°C). For comparison purposes, the study by Tesla et al. identified a minimum temperature ranging between 16°C and 20°C for ZIKV transmission by *Ae. aegypti* (Tesla et al., 2019).*

The experimental evidence shows that no transmission (virus in the saliva) occurs at temperature of 17C in either of the species tested (*Ae. albopictus* or *Oc. detritus*). Therefore, the statement that the authors have identified a minimum temperature threshold between 17 and 19 C (abstract line 42) is an overestimate. The authors are speculating (without evidence) that the transmission would occur also below 19C although they do not have any data to support this assumption.

We detected some transmission at 19°C, so the minimum threshold should be below 19. We did not detect any transmission at 17°C so we believe the minimum to be above 17. Hence, our statement that the minimum is between 17-19 is supported by our data. We do agree that further testing should be done to identify the exact lowest limit, whether that is between 17.1-18.9C in the tested mosquito species (see also our former comment).

*Text added in Discussion [L355]: "4) Our work demonstrates that the minimum temperature for transmission of viable ZIKV is between 17 and 19°C. Future experiments, including EIP and virus infectivity at these temperatures, are planned to narrow this range at low temperatures and therefore produce more accurate estimations of the absolute limit of risk. A recent study, based on infection experiments conducted at low temperatures, showed that former published estimates for the rate of development of the malaria parasite, *P. falciparum*, in both *Anopheles stephensi* and *An. gambiae* was greatly underestimated in the range of 17-20°C (12)."*

Overall the manuscript structure and clarity should be improved, and the relevant contents should be shown in the manuscript (and not in the supplementary file). There are too many different map images provided, it seems that the authors cannot themselves decide what is relevant and what is not.

We disagree with this comment. We tried to find the right balance between the lab experiment and the risk modelling component of the study, and this is always a complicated issue in such ambitious multi-disciplinary studies (reviewers with a modelling background would request more details on the epidemiological risk model while entomologists will request more details on the lab or field experiments...). We have tried to maximize readability of our paper by keeping only essential figures and tables in the main text; and including background details and supporting figure & tables in Supp. Materials. Importantly, methods about models have been moved to Supp. Materials. All authors are happy with the structure of the revised manuscript.

The authors should provide evidence for their laboratory findings to have implications also to real life.

In the manuscript we discuss “real life” implications of our work extensively in discussion, namely our model outputs match recent circulation of arboviruses in southern Europe, USA and Asia. The south-eastern coasts of France also appear at risk in our model estimates; and this finding is consistent with recently reported autochthonous transmission of Zika virus in the city of Hyères in Oct-Nov 2019 (see recent paper by Brady & Hay 2019 and recent ECDC report available at <https://www.ecdc.europa.eu/en/publications-data/rapid-risk-assessment-zika-virus-disease-var-department-france>). The recent emergence of Zika virus in southern France is now mentioned in the updated version of the manuscript.

*Text updated in Discussion [L188]: “. It is noteworthy that our simulated hotspots match areas where recent minor autochthonous transmission of dengue or chikungunya virus has been observed in Europe. Examples include the 2007 CHIKV outbreak in Ravenna in the Emilia-Romagna region in Italy; suspected cases of dengue fever in Cadiz in southern Spain in 2018; and the reported cases of dengue in south-eastern France in 2010, 2013, 2014 and 2015 (20). Furthermore, recent autochthonous transmission of ZIKV was reported in Oct-Nov 2019 in Hyères, nearby Marseille in south-eastern France, where our model predicts moderate risk of transmission (21). This result is consistent with seasonal transmission of ZIKV over southern Europe (3-5 months, Fig S5e) occurring during the warmest spring and summer months (not shown). The picture is different for the USA. A large region where *Ae. albopictus* is already established could theoretically sustain ZIKV transmission (Fig. 3d). The risk is largest over the southern states, in particular over the southern tip of Florida and Texas where autochthonous transmission of ZIKV was reported in 2016-17 (22–24) and where ZIKV transmission could extend between 7 and 12 months (Fig. S5d). Largest standardized R_0 values are simulated over Cuba, the Dominican Republic, Puerto Rico and the Caribbean where active ZIKV circulation (very likely by another mosquito vector, *Ae. aegypti*) was reported in 2016-17 and where year round ZIKV transmission by *Ae. albopictus* is theoretically possible (Fig. S5d). Over temperate Asia, conditions over a large region covering southern China, southern Japan, and Taiwan could sustain ZIKV transmission by *Ae. albopictus* (Fig. 3f). In particular, the largest standardized R_0 value is shown over the southern Guangdong, Guangxi and Hainan provinces, over Taiwan and the most southerly Japanese islands where observed autochthonous transmission of dengue was recently reported (25,26) and where simulated LTS exceeds 6 months (Fig. S5f).”*

The authors do not show the data of the virus amount in the saliva of *Oc. detritus*, or compare that to those obtained from *Ae. albopictus*. The results should be discussed also in the light of the results obtained with *Ae. aegypti*, that showed the low temperatures not supporting virus transmission to saliva (Tesla B et al., 2018).

*Agreed and we added the qRT-PCR data from the saliva infections showing relative titres of virus between the two species and between the time and temperature points (and discussed see Table S5 and the text). In addition, we now compare our results with former published studies in discussion. Tesla et al. showed that no *Ae. aegypti* mosquitoes became infected at 16°C and they also show that a very small proportion of mosquitoes became infected at 20°C, 20 days post infection (Fig. 2c in their paper). In our study, a small proportion of *Ae. albopictus* became infected at 19°C, 17 days post infection (Fig. 1a), and we could not detect infectious *Ae. albopictus* mosquitoes at 17°C. Thus both estimates are quite similar and consistent. The fact that *Ae. aegypti* could not become infectious at low temperature is not that surprising because it's a tropical / semi-tropical mosquito species while we used a temperate strain of *Ae. albopictus* from Italy that can be found in temperate climes and can survive at lower temperatures.*

Item added to the supplementary materials:

Table S5: Relative titre of ZIKV in saliva of *Oc. detritus* (top) and *Ae. albopictus* (bottom). All titres are shown relative to the average of all titres for simplicity (1). Each cell shows: the average titre, the number of positive samples, and the standard deviation. Below each table is the overall average and standard deviation for all positive samples of the species. The difference between the two species is significant ($P < 0.00001$, Mann-Whitney U Test, two tailed).

Oc. detritus

		Days post infection							
		0	5	7	10	14	17	21	28
Temperature	17°C								
	19°C						0.109 1 NA		0.160 1 NA
	21°C					1.209 2 1.634	0.392 1 NA	0.181 1 NA	0.193 2 0.129
	24°C					0.090 1 NA	0.507 1 NA	0.299 3 0.351	0.417 2 0.463
	27°C				0.170 1 NA	0.556 2 0.575	0.313 3 0.334	0.226 2 0.109	
	31°C				0.576 1 NA	0.067 2 0.024			

Average = 0.3599, Standard deviation = 0.4832

Ae. albopictus

		Days post infection							
		0	5	7	10	14	17	21	28
Temperature	17°C								
	19°C						0.655 1 NA	1.951 1 NA	2.015 2 2.392
	21°C				0.181 1 NA		0.507 1 NA	0.863 3 0.950	1.410 2 1.808
	24°C				0.614 1 NA	0.394 2 0.431	1.192 2 1.256	0.475 1 NA	1.639 4 1.703
	27°C				0.699	0.460	2.689	1.380	1.707

					1 NA	2 0.164	1 NA	3 1.283	4 1.047
	31°C			0.090	1.167	0.472	1.432	1.338	2.842
				1 NA	1 NA	2 0.147	4 1.689	2 0.695	5 3.363

Average = 1.3541, Standard deviation = 1.5382

Text added to “results” in the main manuscript [L124]: “We also demonstrate that the titre of ZIKV in the saliva of Ae. albopictus is 3.8x higher than in the saliva of Oc. detritus (P = 0.00001) across all saliva-positive individuals (See Table S5).”

Text modified in discussion [L280]: “Our data show that Ae. albopictus is a more competent vector than Oc. detritus, in terms of both the overall proportion of infectious individuals, and the titre of the virus in the saliva, being significantly higher in Ae. albopictus (Table S5).”

It would also be important to demonstrate that in the case of Ae albopictus and Oc. detritus, at low temperatures, the virus PCR positive saliva would actually be infectious (in cell culture or mosquito infection).

We have added a paragraph in “Materials and Methods” in the main manuscript and items in Supp. Materials to discuss this important issue further.

Materials and Methods [443]: “Detection of infectious virus

To demonstrate that ZIKV is infectious after incubation at 19°C, an additional 30 mosquitoes were fed with a ZIKV containing blood meal using the same methods described earlier. After the blood meal, the mosquitoes were incubated at 19°C for 14 days, before their saliva was extracted. Cytopathic effect (CPE) assays were used to test saliva samples collected for infectious virus. Salivary collections were pooled (into 10 groups of three) and added to Vero (African green monkey kidney) cells maintained in Dulbecco’s modified Eagle’s medium (DMEM) with 2% FBS and 0.05mg/mL gentamycin at 37°C and 5% CO2. Cells were monitored daily for CPE for 7 days..

In Supp. Materials [see Table S4 & Fig. S3]: “Detection of infectious virus”

Table S4: Results of CPE assay. Grey cells indicate CPE was observed.

Sample	CPE (Y/N)						
	1 dpi	2 dpi	3 dpi	4 dpi	5 dpi	6 dpi	7 dpi
Positive control	N	Y	Y	Y	Y	Y	Y
Negative control	N	N	N	N	N	N	N
1	N	N	Y	Y	Y	Y	Y
2	N	N	N	N	N	N	N
3	N	N	N	N	N	N	N
4	N	Y	Y	Y	Y	Y	Y
5	N	N	N	N	N	N	N
6	N	N	N	N	N	N	N
7	N	Y	Y	Y	Y	Y	Y
8	N	N	Y	Y	Y	Y	Y
9	N	N	N	N	N	N	N
10	N	N	N	N	N	N	N

Figure S3: CPE assay photographs. Pictures of CPE assay shown in table S4 from day 4. Positive and negative controls shown, with all 10 pools of three saliva samples.

And as a control, it would have been good to add *Ae aegypti* to the tests to rule out e.g. viral strain properties or other possibly affecting factors that are different from the previously published work.

*In this study we focus on temperate mosquito species. Studying the tropical/semi-tropical *Ae. aegypti* mosquito vector was out of the scope of the current study, and we already discussed this issue in discussion. We used a commonly used ZIKV strain which was used in other published studies (see “Virus blood meal preparation” in Materials and Methods for further details).*

The manuscript should be edited to highlight the actual findings and not the modelling and estimates that are at present highly speculative. If such maps are shown, they should be clearly explained and discussed in the context of other, previously published risk maps. The authors state in the abstract “Our R0-based risk maps show significant risk of ZIKV transmission beyond the current observed range in southern USA, southern China

and southern European countries. “It may be so that the maps show risk beyond currently observed areas, but the maps look pretty modest in comparison to other predicted risk maps of *Ae. albopictus* transmission, for example: Ryan SJ et al, PLOS N trop Dis 2019. It would be important for the temperate regions to take the seasonality (e.g. summer months transmission) into account and also show these in the maps. The authors are only talking about sustained transmission (in the discussion lines 332-333).

*As aforementioned, finding the right balance between the experimental and the risk modelling component of such multi-disciplinary study is always complicated. We now discuss the results by Ryan et al. & Tesla et al. further in discussion and we moved details about modelling sections in Supplementary Materials. The published study by Ryan et al. is interesting, but they do not consider a rainfall effect in their model e.g. their model is only driven by temperature. Consequently, their model estimate leads to very odd and unrealistic estimates for the length of the transmission season (about 4-5 months) in the middle of the Sahara desert, the Middle East and central Australia (Fig. 1b in their paper). Future simulations (RCP8.5 – 2080s) also yield puzzling results for *Ae. albopictus* (with a reduction to “0 months at risk” in central Africa and South America by the 2080s under the RCP8.5 emission scenario, see Fig. 2b in their paper). As requested by other reviewers, we have now calculated the potential length of the ZIKV transmission season. We used a similar approach to Ryan et al, based on standardized R_0 outputs (ranging between 0-1) but we decided to look at standardized R_0 outputs > 0.295 (which corresponds to $T \geq 19^\circ\text{C}$ e.g. minimum temperature for ZIKV transmission found in our lab experiments) instead of standardized $R_0 > 0$ (like Tesla et al. did in their study). We now discuss simulations of seasonal transmission in Results and discussion and we compare our findings to other studies published in the USA, Europe and China in discussion.*

This additional approach is now detailed in “Derivation of the length of the ZIKV transmission season (LTS)” in Supplementary Materials. Note that we moved all model details into Supplementary Materials to improve readability.

*Updated text in Supp. Materials [L141]: Derivation of the length of the ZIKV transmission season (LTS) – *Ae. albopictus* model*

To investigate seasonality in risk, we calculated the length of the transmission season (LTS) based on our standardized $R_0(T)$ estimates. If $\text{std } R_0(T) > 0.295$ (corresponding to $T=19^\circ\text{C}-33.1^\circ\text{C}$ e.g. orange and red colours on Fig. 2d) for a particular location and month, we assumed that temperature conditions were suitable for ZIKV transmission (so we assign 1 to particular location and month); conversely if $\text{std } R_0(T) \leq 0.295$, we assumed that no transmission occurred (so we assigned 0 to that particular location and month). We then sum months at risk on annual basis to derive LTS which ultimately ranges between 0-12 months.

Updated text in Abstract [L44]: “Using these data, we generated standardized basic reproduction number R_0 -based risk maps and we derived estimates for the length of the transmission season.”

Updated text in Results [L149]: “and we also discuss potential changes in the simulated length of the ZIKV transmission season (LTS thereafter) at global scale.”

*Updated text in Results [L173]: “Additionally, ZIKV could theoretically be transmitted all year long by *Ae. albopictus* in the tropics (South America, Africa and Asia, see Fig S5). In temperate regions, the length of the transmission season (LTS) is shorter and varies between 1 and 6 months (Fig. S5).”*

Updated text in Results [L178]: “ZIKV could be transmitted all year round in Central Africa (Fig. S5b). LTS then decreases as a function of the latitude over the African continent, with 1-3 months length of the transmission season simulated over the northern fringe of the Sahel and over temperate regions of Southern Africa (Fig. S5b).”

Updated text in Results [L199]: “... in particular over the southern tip of Florida and Texas where autochthonous transmission of ZIKV was reported in 2016-17 (22–24) and where ZIKV transmission could extend between 7 and 12 months (Fig. S5d).”

Updated text in Results [L209]: "...and where simulated LTS exceeds 6 months (Fig. S5f)..."

Updated text in Discussion [L362]: "Our estimates of the potential length of the ZIKV transmission season by *Ae. albopictus* match recently published findings. In our model, seasonal ZIKV transmission could occur for 4-5 months over southern Europe; this finding is consistent with another R_0 model estimate [about 4 months transmission shown in (39)]. The pattern and magnitude of the simulated length of the ZIKV transmission season over northern and southern America is also consistent with (13,30), who also showed potential year-round ZIKV transmission over Central America, the Caribbean and the northern half of South America. Simulated LTS (about 7-8 months) over the southern provinces of China (Guangdong and Guangxi) also corresponds to observed seasonal transmission of dengue fever over these regions (43,44)."

New risk maps have been added (LTS estimates) in Supp. Materials:

Figure S5: Simulated length of the ZIKV transmission season (LTS in months) based on observed rainfall and temperature data (1980-2010) for a) the globe, b) Africa, c) South America, d) North America, e) Europe and f) Asia.

Figure S7: Simulated length of the ZIKV transmission season by *Ae. albopictus* over Europe. This is carried out for the 2050s (2040-59 average), left column (a, b, c, d) and the 2080s (2070-89 average), right column (e, f, g, h), from the lowest (RCP2.6, top, a, e) to the highest (RCP8.5, bottom, d, h) emission scenario.

Figure S8: Simulated length of the ZIKV transmission season by *Ae. albopictus* over North America. This is carried out for the 2050s (2040-59 average), left column (a, b, c, d) and the 2080s (2070-89 average), right column (e, f, g, h), from the lowest (RCP2.6, top, a, e) to the highest (RCP8.5, bottom, d, h) emission scenario.

Figure S9: Simulated length of the ZIKV transmission season by *Ae. albopictus* over Asia. This is carried out for the 2050s (2040-59 average), left column (a, b, c, d) and the 2080s (2070-89 average),

right column (e, f, g, h), from the lowest (RCP2.6, top, a, e) to the highest (RCP8.5, bottom, d, h) emission scenario.

Figure S10: Simulated length of the ZIKV transmission season by *Ae. albopictus* over Africa. This is carried out for the 2050s (2040-59 average), left column (a, b, c, d) and the 2080s (2070-89 average),

right column (e, f, g, h), from the lowest (RCP2.6, top, a, e) to the highest (RCP8.5, bottom, d, h) emission scenario.

Figure S11: Simulated length of the ZIKV transmission season by *Ae. albopictus* over South America. This is carried out for the 2050s (2040-59 average), left column (a, b, c, d) and the 2080s

(2070-89 average), right column (e, f, g h), from the lowest (RCP2.6, top, a, e) to the highest (RCP8.5, bottom, d, h) emission scenario.

The authors should report only results from *Oc. detritus* and *Ae. Albopictus*, as there is comparable data for these two. There is only very limited testing done (not all temperatures/time points) for *Cx. Pipiens* and *Cs. Annulata*. So the statement that four mosquito species were studied (abstract lines 37, 38) is misleading.

This statement has now been rephrased to lessen the prominence of the other two species. However, we do with to keep our reference to the negative results for Cx. Pipiens and Cs. Annulata, the sample size for each species is clearly stated and both are difficult to work for different reasons which are also stated in the manuscript. We believe it is a good practice for the reader to be made aware of our negative findings and reasons difficult of working with them.

Modified text in Abstract [L37]: "...we assessed the vector competence of two common temperate mosquito species/strains, Aedes albopictus and Ochlerotatus detritus, which were both found to be competent for ZIKV. In addition, we also assessed smaller numbers of two other species, Cx. pipiens pipiens and Cs. annulata."

Modified text in Results [L127]: "We found no evidence that field-obtained Cx. pipiens pipiens or Cs. annulata were competent for ZIKV after 17 days at 21°C (Table S6). Relatively small numbers of these two species were tested due to practical limitations (extremely low feeding rate for Cx. pipiens pipiens and difficulty in collecting large numbers of wild larvae for Cs. annulata). Because of the small numbers, the individuals were tested at 21°C only; this temperature was chosen to provide a balance between excessive mortality at high temperatures and extremely long EIPs at lower temperatures (which leads to higher mortality prior to the EIP being reached). Because of the low numbers, we could not conclude that these species are not competent for ZIKV.."

Modified text in Discussion [L266]: "We demonstrated that both wild populations of Oc. detritus (UK), and newly colonised Ae. albopictus (VERANO colony, Italy) are competent to transmit Zika virus. Whilst we found no evidence of ZIKV transmission in Cx. pipiens pipiens or Cs. annulata, our sample size was too small to conclude that they are not competent."

One of the concerns of this paper is the small numbers of mosquitoes used in some of the experiments. The numbers of mosquitoes in experiments are only found in the supplementary file. The n-numbers should be presented in the actual manuscript in some form.

Our sample size is broadly comparable, and generally higher than other published studies (Jansen et al. used [21-30] Ae. japonicus mosquitoes per temperature for example). Whilst we accept that we use lower sample sizes per cohort than many other studies (Waite et al. for example), we have used a much higher number of cohorts and our logistic regression model (to derive the EIP) considers the whole data (across all days and temperature data points) for a given mosquito species, therefore the total number is important rather than the number per cohort. We used the saliva from a total of 1,065 mosquitoes (combined Ae. albopictus and Oc. detritus numbers). This is far higher than most studies have used.

We disagree with placing the relatively large table in the main text just for the purpose of showing n-numbers, it is easily accessible in the supplementary materials (Table S3), and as stated, the n is very large.

However, we do accept that the sample size for the two minor species (Culex pipiens pipiens and Culiseta annulata) are very low, and have 'downgraded' our focus on them (see former comment for changes).

The authors should cite the relevant previous work on the topic, such as the work done using *Aedes japonicus*

(Jansen et al., 2018) and *Aedes vexans* (Gendernalik et al., 2017). To what extent are the study methodologies comparable?

We now discuss the work by Jansen et al., 2018 & Gendernalik et al., 2017 in the discussion, highlighting similarities and difference to our work and what the combined data indicate.

Text added to Discussion, [L304]: *“Other studies have investigated the importance of wild-caught temperate vectors. Gendernalik et al., 2017 (33) demonstrated 80% competence in a common worldwide vector *Ae. vexans* (caught in the USA) incubated at 27°C. Studies on wild-caught *Ae. japonicus* highlight the importance of assessing the EIP at temperate-realistic temperatures, as Jansen et al., 2018 (34) showed competence at 27°C but very little at 21°C. Combined, these studies show significant potential risk from temperate vectors, but highlight the need to assess transmission at lower temperatures to determine risk in these regions.”*

The authors are discussing their findings in context of *Ae. albopictus*, but very little is said about *Oc. detritus*. For example, the authors should give some overview to the readers about *Oc. detritus* distribution.

*Oc. detritus tends to breed in brackish water – so the distribution of salt marshes could be used as a good approximation for their potential spatial distribution (see for example data gathered by Mcowen et al. 2017). However, the species is not present in all areas of brackish water (e.g. they are totally absent in some potential areas and also larger pools tend to have predators and do not contain *Oc. detritus*) therefore, a map of brackish waters would be an extreme overestimation of the range of this species. Furthermore, this mosquito species has not been extensively studied in the lab and the field (compared to *Ae. aegypti* or *Ae. albopictus* for example) so we are lacking key epidemiological parameters (biting and mortality rates) to investigate their standardized R_0 dependency to temperature.*

Discussion lines 333-335: “The comprehensive analysis of EIP and minimum temperature requirements presented here offer a route to enable informed risk-management and outbreak preparedness in more specific temperate situations.” What do the authors mean by this? How is the data presented here applicable to risk management or outbreak preparedness?

Our EIP estimates can be used in other diverse risk modelling studies to inform vector control and potential seasonal risk of disease transmission (using climate observations and seasonal climate forecast to build early warning systems for example). The EIP data is also useful for comparison purposes to other published studies (Tesla et al., 2018; Jansen et al., 2018 & Gendernalik et al., 2019 for example) as this is an important blooming field of research.

Line 385: “As is standard practice with qRT-PCR experiments, a ‘cut-off’ of 40 Ct was applied to minimise false positives towards the limit of detection.” What do the authors mean by this? The used method is a probe-based assay? I would understand this if it was a SYBR green assay.

Apologies this was a copy/paste error. The incorrect information was removed.

References

Brady O. & Hay S. (2019). *The first local cases of Zika virus in Europe* 394(10213): 1991-1992.

Caminade C., J. Turner, S. Metelmann, J.C. Hesson, M.S.C. Blagrove, T. Solomon, A.P. Morse, M. Baylis (2017). *Global risk model for vector-borne transmission of Zika virus reveals the role of El Niño 2015*. *Proceedings of the National Academy of Sciences*, 114(1): 119-124

Mcowen C, Weatherdon L, Bochove J, Sullivan E, Blyth S, Zockler C, Stanwell-Smith D, Kingston N, Martin C, Spalding M, Fletcher S (2017) *A global map of saltmarshes*. *Biodiversity Data Journal* 5: e11764. <https://doi.org/10.3897/BDJ.5.e11764>

Ryan SJ, Carlson CJ, Mordecai EA, Johnson LR (2019) Global expansion and redistribution of Aedes-borne virus transmission risk with climate change. *PLoS Negl Trop Dis* 13(3): e0007213

Tesla B, Demakovskiy LR, Mordecai EA, Ryan SJ, Bonds MH, Ngonghala CN, Brindley MA, Murdock CC. 2018 Temperature drives Zika virus transmission: evidence from empirical and mathematical models. *Proc. R. Soc. B* 285: 20180795. <http://dx.doi.org/10.1098/rspb.2018.0795>

Waite JL, Suh E, Lynch PA, Thomas MB. 2019 Exploring the lower thermal limits for development of the human malaria parasite, *Plasmodium falciparum*. *Biol. Lett.* 15: 20190275.

Appendix B

Associate Editor Board Member

Comments to Author:

I appreciate the efforts that the authors have made to address the initial reviewer concerns. However, several issues remain that have been raised by the reviewers. For example, reviewer 1 particularly draws attention to the importance of including *Ae. aegypti* in ZIKV models, and the need for clarity on the parameter estimates presented. Reviewer 2 points out additional information required for understanding and contextualizing the results from saliva samples. These aside, I am ultimately left questioning how much of an impact the results here would have in progressing our understanding of ZIKV risk, and if they would interest the broad readership of Proc B.

*Answer: We agree that *Ae. aegypti* is the major mosquito vector of DENV and ZIKV in tropical and semi-tropical regions, and this is consistent with former modelling results by our group (Caminade et al. 2017). However, recently observed transmission of DENV and ZIKV in temperate regions of Europe (mostly on the Mediterranean and Adriatic coasts) by *Ae. albopictus* is worrying, and we think that our study provides interesting insights into potential risk under current and future climate conditions in temperate regions of Europe, the USA and Asia. The recent autochthonous transmission of ZIKV by *Ae. albopictus* in Hyères nearby Marseille, south-eastern France in Nov 2019 is also consistent with our model findings. Additionally, to the best of our knowledge, we are the first group to provide detailed temperature dependent EIP schemes for ZIKV by *Ae. albopictus* and investigate the potential of another UK mosquito to transmit ZIKV at a surprisingly high level of competence (as highlighted by two of the reviewers in the first round).*

We have also addressed the comment by reviewer three with significant additions to the text contextualizing the additional ZIKV viability experiment results and methodology.

This work was made possible by a multi-disciplinary collaboration of various experts and access to world class facilities at the Liverpool School of Tropical Medicine in the UK. We believe that the updated version of the manuscript addresses the reviewers' comments and hope that our work will be considered for publication in Proc. Roy. Soc. B. In the following sections, please find a detailed response to the reviewers' comments.

Reviewer(s)' Comments to Author:

Referee: 1

Comments to the Author(s).

In this paper the authors extrapolate solid laboratory data into a global model for ZIKV risk based on temperature driven parameters. As *Ae. albopictus* seems limited to areas with relatively high rainfall/humidity, it was not clear how the authors accommodated these data into their models. Specific comments have been entered directly on the attached files using tracked changes [a little cumbersome after converting the .pdf to .docx].

Summary thoughts follow.

1. Writing/English. The paper could use some additional editing to improve the sentence

structure and general flow of ideas. I have tried to help [see attached file], but I feel a good 'read' would be useful.

We thank the reviewer for their very significant 'proof-reading' of the manuscript and their more minor suggestions for improvement, for which ~90% we agreed with and changed to increase readability. This also includes typos, such as 14 days instead of 21 in the CPE assay supplementary methods, which we thank the reviewer for spotting. We have also re-ordered the results section to maintain a logical field-to-lab-to-model flow.

A few linked comments pertaining to numbers of *Cx. pipiens pipiens* or *Cs. annulata* (below) are commented on further:

Comment 16: Numbers in the supplementary materials for this single temperature/time point seemed adequate to measure susceptibility?

Comment 22: Unless I misunderstand experiment results, the numbers tested would seem suitable to indicate these are refractory to infection.

We tend to agree, given that our numbers are quite large, however, both other reviewers commented the opposite and requested that these results to be down-played (which we did in the first review round). It is therefore clear that these results are open to interpretation by the reader, consequently we worded the results as 'no evidence for competence', which keeps the statement conservative.

2. Model. My greatest problem was using *Aedes albopictus* and *Ae. detritus* to model ZIKV transmission risk based on R_0 estimated for temperature areas [some of which don't have detritus] or limited populations of *albopictus*, and not including the primary vector, *Ae. aegypti*. As the authors acknowledge, large ZIKV outbreaks have occurred mainly where there are populations of *aegypti*. Although *albopictus* is an efficient laboratory vector of many arboviruses, it does not seem to be a primary vector of any arbovirus, probably because of its general host selection patterns. Similar comments relate to other species mentioned such as *Culex* spp., *Ae. japonicus*, *Ae. vexans*, etc. These species may become naturally infected but will not sustain transmission human-to-human transmission. As the authors point out, *albopictus* is broadly distributed in the USA, but secondary ZIKV and DENV transmission from travellers have mostly occurred where there are suitable populations of *aegypti* and not within the more northern temperate areas with established *albopictus* despite the repeated introductions of viruses by travellers.

We agree that *Ae. aegypti* is the main vector of dengue and Zika virus in Tropical and semi-Tropical regions. A former publication by our group (Caminade et al. 2017, PNAS) consistently shows that *Ae. aegypti* primarily drives large R_0 values in the Tropics. However, we have been very careful about limiting discussion of model results in Tropical regions and we mostly focus our discussion on temperate regions of Europe, the USA and Asia (both in results and discussion sections). We disagree about the reviewer's comment: "secondary ZIKV and DENV transmission from travellers have mostly occurred where there are suitable populations of *aegypti* and not within the more northern temperate areas with established *albopictus* despite the repeated introductions of viruses by travellers". Whilst this statement is mostly true for North America (where autochthonous transmission of ZIKV by *Ae. aegypti* was observed in Texas and Florida) it is not correct for Europe, where DENV (in south-eastern France, Croatia, Spain mostly) and ZIKV (Hyères nearby Marseille in south-eastern France in Nov 2019) transmission by *Ae. albopictus* was observed. There was no contribution from *Ae. aegypti* - the most up to date surveillance data shows that *Ae. aegypti* is restricted to forested areas on the eastern side of the Black sea in Europe [see

<https://www.ecdc.europa.eu/en/publications-data/aedes-aegypti-current-known-distribution-july-2019>]. Additionally, a recent study by one of our co-authors (Manica et al., 2019) shows that *Ae. albopictus* could also potentially transmit Yellow Fever, in the Lazio region in Italy, not far from Rome's International Airport. In short, our paper mostly focuses on the temperate limits, as reflected in the title. But we have also made some additional changes to the manuscript to further highlight that this is our focus:

Example of modified text: "Our study focuses on the risk in temperate regions and hence does not account for transmission by the primary ZIKV vector *Ae. aegypti*, which may be important in transmission in some of the warmer areas of regions we investigated. We don't consider this a limitation per se, but rather a focus on different regions."

Many other small changes made as per reviewer's commented attachment

3. Parameters. I didn't understand some of the model parameters:

We have added references and justification to the table in the supplementary materials dealing with the model parameters (pasted below) and have addressed each individual comments here.

$b=0.5$: Does this mean that half the albopictus bites are on humans result in transmission? If this is for infectious females, I think this could be low. If based all females, then it is too high.

b refers to the probability that a bite by an infected vector results in transmission of infection to the host (also referred as to " b_h " in the scientific literature) and so yes, $b = 0.5$ does mean half of bites by infectious females lead to transmission. Given it refers to infectious (& not all) females, the reviewer thinks 0.5 might be low, but it is hard to know. This parameter is very difficult to estimate in the field. Given that it could range from 0-1, choosing the median value of 0.5 is a common way of addressing the uncertainty. We used a median value of $b = 0.5$, consistently with Rocklov et al., 2016 and Guzzetta et al., 2016. This value is also consistent with b estimate for dengue and *Ae. albopictus* based on the meta-review paper by Andraud et al., 2012. As noted by the reviewer, as we standardize R_0 values, constant parameters do not affect the shape of the standardized R_0 function, hence our main findings.

See updated table below for changes

$m = 28.2$: Does this mean that throughout the world there are, on average, 28 albopictus per human host? I agree that a constant value will not alter the shape of the temperature dependent curves, but rather should alter the magnitude of the R_0 estimates. What would R_0 be if this value was 2.8? This was even more problematic in a one host model where all females feed on humans -- not the normal feeding patterns observed for albopictus where many bites are diverted to other hosts.

Vector to host ratio (m) is always difficult to model; m can be fixed as a constant or dynamically modelled using SEIR type models, or estimated using trap field data combined with population data for a given surface area (see Romeo-Aznar et al., 2018). A recent study conducted by field entomologists in Italy, estimates that vector to host in the Lazio region surrounding Rome can range from 4 to 138 *Ae. albopictus* females per inhabitant (Guzzetta

et al., 2016), so our m value falls well within that range and is close to the geometric mean (23.5) which is commonly used for averaging insect numbers. Given the large spatial heterogeneity in m (between urban, rural areas, given local predators...), and large differences that are shown in field-based studies, we originally decided to use a constant value for m , before applying standardization to R_0 . In order to provide baseline estimates for m , b , β and R_0 , we analysed all basic reproduction number models that were recently reviewed by Kobres et al., 2019 in October 2019. Many groups used constant and very diverse values for m (see Table R1). Our vector to host ratio (m) value was originally fixed to 28.2 to yield maximum R_0 values of 4 (with a R_0 peak at 29°C), following preliminary R_0 ZIKV values for Colombia (Nishiura et al., 2016). This maximum R_0 value of 4 was also consistent with the work by Rocklov et al., 2016 and Ferguson et al. 2016 (see Table R1 at end of this document). Given our R_0 equation, $m=2.8$ would roughly leads to $R_0 = 0.4$ and Zika could not have sustained transmission.

Importantly, Ferguson et al used a similar technique in their 2016 Science paper. Instead of scaling m (as we did) their team instead scaled the parameter b (probability of an infected vector to transmit a pathogen to a host) which has the same effect of yielding a realistic seasonal range in R_0 : “(b) Assigned as 0.7 to give a mean seasonal peak R_0i (given values of Δi) of 4.1 and an annual mean R_0i of 2.3.”, see Table S6 in the related Supp. Materials available at [\[https://science.sciencemag.org/content/sci/suppl/2016/07/13/science.aag0219.DC1/FergusonSM.pdf\]](https://science.sciencemag.org/content/sci/suppl/2016/07/13/science.aag0219.DC1/FergusonSM.pdf)

Furthermore, and as noted by the reviewer, we standardize R_0 (following Ryan et al., 2019), consequently, the shape of the standardized $R_0(T)$ function is not at all affected. For consistency, we decided to retain the b , beta and m parameter values given by Guzzetta et al., 2016 for Italy, because their R_0 type model was specifically designed to assess the risk of ZIKV transmission by *Ae. albopictus* in temperate regions.

Updated text and table in sup methods:

Table S7: Standardized $R_0(T)$ model parameter settings. *denotes parameters which are dynamically simulated in space and time over the whole time period. T stands for temperature.

Symbol	Description	Constant/Formula	Comments	Ref
*a(T)	Biting rate (per day)	$a=(0.0043T + 0.0943)/2$	The linear dependency to temperature was based on estimates for Ae. aegypti in Thailand. Biting rates for Ae. albopictus were halved based on published observed feeding interval data	(6,7)
b	Transmission probability - vector to host (0-1)	b=0.5	Baseline value of ref (14). Note that constant parameter values should not impact the shape of the final standardized $R_0(T)$ estimate (std $R_0(T)$ was rescaled to range between 0-1, see Fig 2d)	(14)

β	Transmission probability - host to vector (0-1)	$\beta=0.0665$	Baseline value of ref (14) Note that constant parameter values should not impact the shape of the final standardized $R_0(T)$ curve (std $R_0(T)$ was rescaled to range between 0-1, see Fig 2d)	(14)
* $\mu(T)$	Mortality rate (0-1 per day)	$\mu=1/(1.1+\exp(-4.04+0.576T))+0.11883$ if $T < 15^\circ\text{C}$ $\mu=0.000339T^2-0.0189T+0.336$ if $15^\circ\text{C} \leq T < 26.3^\circ\text{C}$ $\mu=1/(1.065+\exp(32.2-0.92T))+0.073079$ if $T \geq 26.3^\circ\text{C}$	Mortality rate was derived for both mosquito vectors from published estimates based on both laboratory and field data. Due to discontinuity around the different temperature thresholds these estimates have been updated.	(8)
* $u(T)$	Extrinsic Incubation Rate (days)	$1/u = \text{eip_albo} = -1.0757T+43.0342$ $\text{eip_detr} = -1.07567T+46.025$	EIP(T) was estimated based on our updated laboratory data (see Figure 2). EIP(T) for Ae. detritus is also shown for comparison.	This study
m	Vector to host ratios	$m=12.9$	Calculated as maximum vector to host density ratios (508/39.4) in ref [x] Note that constant parameter values should not impact the shape of the final standardized $R_0(T)$ curve (std $R_0(T)$ was rescaled to range between 0-1, see Fig 2d)	(14)
r	Recovery rate (per day)	$r=1/7$	1 week viraemia is a common value for ZIKV.	(15)

References

- Andraud M, Hens N, Marais C, Beutels P (2012) Dynamic Epidemiological Models for Dengue Transmission: A Systematic Review of Structural Approaches. PLoS ONE 7(11): e49085. <https://doi.org/10.1371/journal.pone.0049085>
- Caminade C., J. Turner, S. Metelmann, J.C. Hesson, M.S.C. Blagrove, T. Solomon, A.P. Morse, M. Baylis (2017). Global risk model for vector-borne transmission of Zika virus reveals the role of El Niño 2015. Proceedings of the National Academy of Sciences, 114(1): 119-124. [doi:10.1073/pnas.1614303114](https://doi.org/10.1073/pnas.1614303114)
- Ferguson NM, Cucunuba ZM, Dorigatti I, Nedjati-Gilani GL, Donnelly CA, Basanez MG, et al. EPIDEMIOLOGY. Countering the Zika epidemic in Latin America. Science (New York, NY). 2016;353(6297):353–4.
- Guzzetta, Giorgio and Poletti, Piero and Montarsi, Fabrizio and Baldacchino, Frederic and Capelli, Gioia and Rizzoli, Annapaola and Rosà, Roberto and Merler, Stefano, Assessing the

potential risk of Zika virus epidemics in temperate areas with established *Aedes albopictus* populations. *Eurosurveillance*, 21, 30199 (2016), <https://doi.org/10.2807/1560-7917.ES.2016.21.15.30199>

Kobres P-Y, Chretien J-P, Johansson MA, Morgan JJ, Whung P-Y, Mukundan H, et al. (2019) A systematic review and evaluation of Zika virus forecasting and prediction research during a public health emergency of international concern. *PLoS Negl Trop Dis* 13(10): e0007451. <https://doi.org/10.1371/journal.pntd.0007451>

Manica M, Guzzetta G, Filipponi F, Solimini A, Caputo B, della Torre A, et al. (2019) Assessing the risk of autochthonous yellow fever transmission in Lazio, central Italy. *PLoS Negl Trop Dis* 13(1): e0006970. <https://doi.org/10.1371/journal.pntd.0006970>

Nishiura H, Mizumoto K, Villamil-Gómez WE, Rodríguez-Morales AJ (2016) Preliminary estimation of the basic reproduction number of Zika virus infection during Colombia epidemic, 2015-2016. *Travel Med Infect Dis* 14(3): 274–276.

Rocklöv J, Quam MB, Sudre B, German M, Kraemer MUG, Brady O, et al. Assessing Seasonal Risks for the Introduction and Mosquito-borne Spread of Zika Virus in Europe. *EBioMedicine* [Internet]. 2016 Jul [cited 2018 Jan 30];9:250–6. Available from: <http://www.ncbi.nlm.nih.gov/pubmed/27344225>

Romeo-Aznar Victoria, Paul Richard, Telle Olivier and Pascual Mercedes Mosquito-borne transmission in urban landscapes: the missing link between vector abundance and human density *285Proc. R. Soc. B* <http://doi.org/10.1098/rspb.2018.0826>

Ryan SJ, Carlson CJ, Mordecai EA, Johnson LR. Global expansion and redistribution of Aedes-borne virus transmission risk with climate change. Han BA, editor. *PLoS Negl Trop Dis* [Internet]. 2019 Mar 28 [cited 2019 Aug 16];13(3):e0007213. Available from: <http://dx.plos.org/10.1371/journal.pntd.0007213>

Additional information

Table R1:

R₀ for Zika virus based on the published literature. Most *R₀* models were derived from the most recent review by Kobres et al., 2019. Bold references denote modelling papers published for temperate regions and for *Ae. albopictus*.

R₀	Epidemiological parameters (m, b and beta)	Reference
R₀ = 2.0555 [95% CI: 0.523-6.3] – Brazil Colombia and Salvador	m = 5 [range 1-10] b = 0.4 β = 0.5	Gao et al., 2016 [https://doi.org/10.1038/srep28070] Ae. aegypti

R_0 range depending on location and time of year R_{0_max} = 6.9 – Argentina	$m = 1.5$ $b = [0-1]$ temperature dependent $\beta = 0.62$	Orellano et al. 2017 [doi: 10.26633/RPSP.2017.120] Ae. aegypti
R_0 ranges between 0-4 (temperature dependency) - Europe	$m = NA$ (only temperature based function for R_0) b and $\beta = [0-1]$ temperature dependent	Rocklov et al., 2016 [https://doi.org/10.1016/j.ebiom.2016.06.009] Ae. albopictus and Ae. aegypti
R_{0_max} ranges between 0-3.5 - Italy	m, if estimated from raw trap catches, (assuming one trap = one host) can range up to about 300 (see Fig. 4). $m_max = 12.9$ (ratio from table 1 using maximum vector density and minimum host density) $b = 0.5$ (baseline, similar to our estimate) $\beta = 0.066$	Guzzeta et al., 2016 [http://dx.doi.org/10.2807/1560-7917.ES.2016.21.15.30199] Ae. albopictus
R_{0_median} ranges from 2.6-4.8 – for various French Polynesian islands	NA – parameters modelled as distributions	Kucharski et al., 2016 [https://doi.org/10.1371/journal.pntd.0004726]
$R_0 = 1.61$ [1.53-1.69] for French Polynesia $R_0 = 1.36$ [1.3-1.42] for Martinique	NA – R_0 modelled using clinical case data	Andronico et al., 2017 [https://doi.org/10.1093/aje/kwx008]
R_0 range = [1.5-4.1] for Yap, Micronesia, Tahiti and Moorea, French Polynesia and New Caledonia	NA – modelled statistically or dynamically using mostly host factors	Champagne et al., 2016 [http://dx.doi.org/10.7554/eLife.19874.001]

“R₀ rapidly declined from 10.3 (95% CI: 8.3, 12.4) in the first disease generation to 2.2 (95% CI: 1.9, 2.8)” for Colombia.	NA – modelled statistically using clinical case data	Chowell et al., 2016 [doi: 10.1371/currents.outbreaks.f14b2217c902f453d9320a43a35b9583]
R₀ = 0.16 [0.13-0.19] for Florida, USA.	NA modelled using clinical case data	Dinh et al., 2016 [doi: https://doi.org/10.1186/s12976-016-0046-1]
R₀ = [0-4.1]	m is spatially dynamic b = 0.7 β = 0.7	Ferguson et al., 2016 [DOI: 10.1126/science.aag0219]
Reproduction number = [4.8-14] for Yap Islands	NA modelled using clinical case data	Funk et al., 2016 [https://doi.org/10.1371/journal.pntd.0005173]
R₀ = 2.56 (range 1.42-3.83) for Colombia	NA modelled using clinical case data and time intervals	Majumder et al., 2016 [https://dx.doi.org/10.2196%2Fpublichealth.5814]
R₀_median = 0.82 [0-13.1] R₀ eastern USA	m = 2 [0.5-10] b = 0.35 [0.1-0.75] β = 0.31 [0.1-0.75]	Manore et al., 2017 [https://doi.org/10.1371/journal.pntd.0005255] Ae. albopictus
R₀ = 1.88 [1.59-2.22] averaged for French West Indies and Polynesia islands	NA	Riou et al., 2017 [https://doi.org/10.1016/j.epidem.2017.01.001]
R₀ = 1.41 (95% CI): 1.15–1.74 in San Andres and 4.61 (95% CI: 4.11–5.16 in Girardot	NA modelled using clinical case data	Rojas et al., 2016 [https://doi.org/10.2807/1560-7917.ES.2016.21.28.30283]

Ro = 3.8 [2.4,5.6], and that the fraction of cases due to sexual transmissio n was 0.23 [0.01,0.47], Baranquilla , Colombia	NA	Towers et al., 2016 [https://doi.org/10.1016/j.epidem.2016.10.003]
R₀ = 2.33, 95% CI: 1.97–2.97 for Rio de Janeiro		Villela et al., 2017 [https://doi.org/10.1017/S0950268817000358]

Referee: 3

Comments to the Author(s).

I am glad to see that the authors have tested the saliva samples for viable virus, however this part requires some clarification in the text.

We thank the reviewer for his/her positive comment. We have added multiple additional references to the experiment where relevant, clarified that the virus is demonstrably viable at 19°C, and expanded the description of the methodology use for this experiment.

Examples of edits to the text are shown below:

“newly colonised *Ae. albopictus* (VERANO colony, Italy) are competent to transmit viable Zika virus at temperatures as low as 19°C”

“In order to demonstrate that ZIKV in the saliva of the mosquitoes incubated at the lowest positive temperature (19°C) is viable, we tested the ability of virus extracted from saliva to infect VERO cells via a cytopathic effect (CPE) assay. Saliva from 30 G₁₁ *Ae. albopictus* (VERANO colony) females was extracted 21 days after ZIKV challenge were pooled into 10 lots of three samples. 4 of these 10 pools were positive for viable virus, indicating at least 13% of saliva samples were positive (consistent with day 21 19°C *Ae. albopictus* results shown in Figure 1a). This confirmed that our minimum temperature for demonstrable transmission does produce infectious virus and hence does present a transmission risk.”

Appendix C

General comments

The manuscript by Blagrove et al. with the title Potential for Zika virus transmission by mosquitoes in temperate climates is split into two important research topics: first, vector competence studies of four temperate mosquito species, and second modelling, using outcomes of the experimental study for *Aedes albopictus* to evaluate the impact on disease risk (R_0) on a global scale. To combine these two in one publication is very valuable. When reading the title this was not immediately clear, so we propose to make this clear in the title.

Overall the manuscript is well written, however, the manuscript needs to be thoroughly revised since it is hard to follow at times. Sections are often in the wrong place, e.g. many supplementary method sections should actually be placed in the main manuscript and there are discussion items in the results section. The discussion is very lengthy. We will elaborate more on these issues below.

Although the work is an interesting read for a specific niche, we think it is currently not always understandable for the broader audience of Proc Roy Soc B that may not be familiar with the subject area. We will further elaborate on our main issues below.

We thank the reviewer for their comments and detailed suggestions on how to improve the readability of the manuscript. We have made the vast majority of the changes suggested and have re-ordered parts of the manuscript to aid readability, including moving more discursive sections of the results to the discussion. Furthermore, we have clarified which experimental data are used in our model and moved the essential components of the modelling methods into the main manuscript.

We have also put more context into both the introduction and discussion, comparing our results to the effect of temperature on other viruses. Overall we feel that this has greatly improved the readability and context of the manuscript and consequently its appeal to the broader audience of Proc Roy Soc B.

Major comments

1) From reading the manuscript it is not immediately clear which experimental data points went into the model. There is no bridge presented in the M&M or Results section that explains this. Modelling is not my personal expertise, but for this journal it has to be understandable for a broader public. This needs thorough revision.

We agree with the reviewer. Due to word count limitation – we have now added a short paragraph about model construction and assumptions in the main text (as there is not

space for all of our interdisciplinary methods) which contains all information needed to understand and follow the main text. We then make references to the ESM for more details for the specific modelling audience. We have now organised the Supp. Material into three large sections for clarity: A – Model Supplementary Information; B – Experimental Data Supplementary Information, and C – Additional Analyses and Maps. The only experimental data feeding our model is the newly derived EIPs for ZIKV using the *Ae. albopictus* data, the other epidemiological parameters have been estimated using other peer reviewed studies. We have clarified this in the main text in both the introduction and discussion (details below with the specific points). Note that the Material and Method section is now section 2, followed by the Results in section 3, consequently many items have been re-ordered. We hope that the new paper structure addresses the reviewer's concerns.

2) I feel that in both Introduction and Discussion the authors should put the story in more perspective. They only mention general information on temperature and effects on pathogens or ZIKV more specifically (paragraph starting with line 67). I would like to see more information on the effect of temperature on other arboviruses such as WNV, a flavivirus just like ZIKV. In the discussion the authors do mention an example of temperature on malaria transmission (line 317-310), however arbovirus examples would be better matching and relevant to the discussion.

We have added further examples of the effect of temperature on arboviruses to both the introduction and discussion.

Modified text: "Temperature has a profound effect on the ability of insect vectors to transmit pathogens, especially viruses. Decreasing environmental temperature increases the time taken for the vector to become infectious after taking an infected blood meal (termed the extrinsic incubation period, (EIP)) [12], for example *Culex tarsalis* females infected with West Nile virus (WNV) can become infectious at 5 days post infection at 30°C, but require 36 days at 14°C [13]"

Modified text): "Our finding of a minimum temperature for transmission of ZIKV in both *Ae. albopictus* and *Oc. detritus* of between 17°C and 19°C is supported by the findings of Heitmann et al. [33] showing ZIKV transmission in *Ae. albopictus* at 27°C but none at 18°C after 21 days. Interestingly, this minimum temperature is substantially higher than the minimum recorded temperature for transmission of the closely related WNV at 14°C [13], indicating that ZIKV may not present as much risk to temperate regions as WNV. ZIKV appears to develop at lower temperatures than DENV, for which most studies have shown no or excessively slow viral development at temperatures lower than 20°C [34], indicating that ZIKV presents a greater risk to temperate regions. Previous studies of ZIKV competence by *Ae. albopictus* have almost exclusively used only 'standard' tropical insectary temperatures of 27±1°C or with a single additional lower temperature [33]."

3) Line 113-120. and 225-227. I advise to remove all data on the vector competence of *Cx. pipiens pipiens* and *Cs. annulata*. Not enough tests were performed to make a solid statement. The information only makes the manuscript more confusing.

We agree that they could add confusion to the manuscript so have been removed from the main text. However, a previous reviewer suggested highlighting these results as they believed that they were important. Consequently, the results for these two species have been moved to the supplementary materials and only referred to briefly in the results.

4) There are several items in the Supplementary materials that should be put in the main manuscript:

- a. Models of extrinsic incubation period (EIP10)
- b. Detection of infectious virus
- c. Derivation of $R_0(T)$ for ZIKV – *Ae. albopictus* model

Reading these parts are essential for understanding the manuscript. Some Figures and Tables can stay in the supplementary material such as Table S3, Figure S1, Figure S2, Table S4 and Figure S3 etc. These need to be placed in the order they appear in the main text though, which is not the case now and results in great difficulty reading and reviewing the results section of the manuscript.

We agree and have moved b. 'Detection of infectious virus' into the main text in its entirety. As mentioned above, we have added a summary section on model construction and assumptions in the main manuscript (merging a. and c.), this new section is sufficient to understand and follow the work. We refer to the ESM for more specific additional details.

Text added (repaste once refs updated):

Model construction and assumptions:

First, we used a logistic regression model to estimate the EIP10, e.g. the extrinsic incubation period measured as the time until 10% of infected mosquitoes become infectious for a given temperature, based on our experimental data available at 17°C, 19°C, 24°C, 27°C and 31°C. See section "Models of extrinsic incubation period (EIP10)" in Supp. Mat. for further details.

Second, we include this newly obtained $EIP_{10}(T^\circ)$ scheme for ZIKV and *Ae. albopictus* into a basic reproduction number R_0 model for ZIKV (one vector – one host model). Given recently published evidence, the model has been slightly modified.

$R_0(T)$ is given by:

$$R_0(T) = \left(\frac{b\beta a(T)^2}{\mu(T)} \right) \left(\frac{v(T)}{v(T) + \mu(T)} \right) \left(\frac{m}{r} \right)$$

The biting (a), mortality (μ) and extrinsic incubation ($v=1/EIP_{10}(T^\circ)$) rates depend dynamically on temperature. All other epidemiological parameters are fixed to a constant value (Table S1). A rainfall criterion is applied to mask desertic regions following [22]. The final $R_0(T)$ estimate is standardized to range between 0 and 1, to be consistent with previously published models and make our results directly comparable to such studies [23]. We also calculate the potential length of the ZIKV transmission season (LTS in months) based on standardized $R_0(T)$. For further details about the standardized $R_0(T)$ model and LTS see sections "Derivation of $R_0(T)$ for ZIKV – *Ae. albopictus* model" and "Derivation of the length of the ZIKV transmission season (LTS) – *Ae. albopictus* model" in Supp. Mat.

Finally, we integrate the $R_0(T)$ model with observed gridded climate data for the recent period and with an ensemble of climate change projections for the future to investigate regions at risk of ZIKV transmission by *Ae. albopictus*. See section "Observed climate datasets and climate change scenarios input data" in Supp. Mat. for further details.

5) The link between risk of ZIKV and vector presence is unclear. The presence of *Aedes albopictus* is plotted with black dots in Fig. 2. This needs more explanation. For example, risk of ZIKV is high in Suriname, however, *Ae. albopictus* is not present there, so it's not a realistic risk?

The risk we are calculating is mostly derived from temperature and rainfall data and does not consider the introduction or movement of the vector (by road, containers etc). This is now clearly stated in the manuscript. We received similar criticisms for a study published in 2012 (Caminade et al., 2012) that correctly anticipated the spread of *Ae. albopictus* in European countries to date. Note that the highly cited study by Benedict et al., 2008 (which uses climate predictors with probable means of introduction via tyre shipments and shared borders with infested countries) also listed Suriname as a potential niche for *Ae. albopictus* in South America. *Ae. albopictus* has been found in neighbouring French Guyana and Amapa state in Brazil (see https://www.scielo.br/scielo.php?script=sci_arttext&pid=S0044-59672019000100071)

Minor comments

Keywords

1) Line 31. Too many keywords mentioned, only six allowed.

Done

Abstract

2) Line 42. Specify the wide range of incubation temperatures.

Done

3) Line 51. As result of what do you expect changes in the future? Temperature?

Done

Introduction

4) Line 54. The Zika virus is no longer a 'rapidly emerging' disease.

Done

5) Line 55 and Line 66. Include a reference at the end of these sentence to support the factual statements made here.

Done

6) Line 87. This is about reported vector presence of *Ae. albopictus* specifically, adjust. Also provide the source.

Done

Material and methods

7) Line 352-364. More information should be given on ambient conditions, larval diet, and year and season of sampling of the mosquitoes.

Done

Text added: "All UK mosquitoes were collected in April-May 2016, and reared in the water which they were found (with no additional diet supplements) at ambient UK conditions, they were allowed to emerge and mate in 30cm³ BugDorm cages in Leahurst Campus, University of Liverpool, until approximately one week post emergence, where they were transported to an insectary at 25°C, 12:12 light:dark and 70% RH."

8) Line 320. What is an 'odorized' feeding membrane? What did it consist of?

Done

9) Line 371. Give more details, which virus passage has been used, on which cells was the virus passaged, which medium was used? How much blood was used relative to virus? This should increase repeatability.

Done

Text added: "The PE243 isolate of ZIKV from Brazil was used for all infection experiments, virus was cultured in Vero E6 cells, using DMEM and 10% FCS, and prepared to a final titre of 1x10⁶ PFU/mL in 50% human blood."

10) Line 378-385. More clarity is needed. For example how long were the mosquitoes salivated? The reference Blagrove et al. 2018 (48) does not give enough information.

What type of mineral oil is used? How much? As it is now, there is not enough information to repeat the experiment.

Done

11) Line 385. Carcass is mentioned, in the supplementary materials this is called body. Choose one of the two.

Done

12) Line 390. 'the kit'.

Done

13) Line 395. Need more clarity. How do you assess PFU's? Which cells are used, which medium (agar)? And how long do you wait for PFU's to show?

Done

Text added: "Vero E6 cells were infected with serial dilutions of ZIKV and incubated under an overlay of DMEM with 2% FCS and 0.6% Avicel (FMC BioPolymer) at 37°C for 5–7 days. Cells were fixed with 4% formaldehyde then stained with Giemsa for visualisation"

14) Line 396. 'Limit' without capital letter.

Done

15) Line 398-400. See Major comment 4.

Done

Results

16) Line 92. How many mosquitoes were tested at each T? Refer here to Table S3.

Done

17) Line 94. Letters in Fig. 1 do not correspond with the ones in text, they are turned around. Also I was wondering, why not show average titres of each time point too?

Many thanks for spotting, letters in the sub-panels on Fig. 1 have been re-ordered accordingly

18) Line 104. It's not clear from the methods you did experiments looking at CPE. See major comment 4. Also add the word 'and' in 'challenge, and were pooled'. And in the next sentence change 4 into four.

Done

19) Line 111. This is an important result which should not be 'hidden' in the Supplementary materials in Table S5. The Table should be changed, see comment 56.

We agree that the result is important and it is referred to in the text, however, nothing beyond simply seeing that the virus was viable is shown by adding the figures from the ESM into the main text. As these pictures use a lot of space, and do not add any extra information we feel that they are best kept in the ESM as the manuscript is already at the length limit.

20) Line 123. What is the idea behind EIP10? Why is this chosen and not EIP5 or EIP15? Also, 'the time in days to reach 10% saliva-positive', how is this calculated? Could not find this in the methods section.

EIP10, as stated in the text, is the time in days to reach 10% saliva-positive. In other words, this is calculated as the time in days to reach the tenth percentile of the plateau value (shown on Fig S2) for a given temperature. This is now clarified in the text. Similar comment applies to EIP50 (time to reach median of the plateau value) and EIP90 (time required to reach the 90th percentile of the plateau value for a given temperature). In general, modellers tend to use EIP10, EIP50 (which is preferred most of the time) and EIP90. For example, see the work by Ohm et al., 2018 for malaria parasite incubation periods in *Anopheles* mosquitoes, and note that the theoretical framework is similar for viruses.

For the majority of our conditions, 50% competence was never reached, therefore, we used EIP10 as the majority of our positive conditions did reach this level, resulting in interpolation rather than extrapolation. In addition, the EIP10 is a more reliable estimate of "the first few mosquitoes becoming infectious". Our EIP10 estimate for ZIKV in *Ae. albopictus* is also in very good agreement with the highly cited study by Chouin-Caneiro et al., 2016. Who showed infection in ZIKV infected *Ae. albopictus* 14 days post infection at 29+-1C (fixed temperature) and our EIP10 estimate indicates about 12 days incubation period at the same temperature.

21) Line 127-128. Model 1 identified a linear relationship, but in what direction?

It's a negative linear relationship (when temperature increases the duration of the EIP shortens). We have clarified this point in the text:

Updated text: "Model 1 identified a negative linear relationship between the duration of the EIP and temperature..."

22) Line 131-132. So what is this estimate to which it is similar?

By "estimate", we mean the peak of simulated std $R_0(T)$ shown at 28-29°C for ZIKV and *Ae. albopictus* (Fig S4d). The peak in standardized $R_0(T)$ is very similar to former published results based on a R_0 model estimating ZIKV transmission risk by *Ae.*

albopictus (see Fig. S15D in Caminade et al., 2017). Note that the EIP parameter was derived for DENV and *Ae. albopictus* at time of publication (Caminade et al., 2017), hence therefore we conducted the current study in order to obtain robust EIP estimates for ZIKV and *Ae. albopictus* at different temperatures (even if our new results do not impact the shape of the standardized $R_0(T)$ function). We updated the text and the associated reference for clarity (the former version of the manuscript was referring to Brady et al. 2013 who only focused on adult mosquito survival parameters).

We also notice that our reference was incorrect and corrected it, it should have been Caminade et al., 2017

Updated text: "Our standardized $R_0(T)$ peak estimate for *Ae. albopictus* potential to transmit ZIKV (Fig S3d. solid red line) occurs at slightly higher temperature than highlighted previously for *Ae. aegypti* [24] and is very similar to a published estimate [25] (Fig. S3d solid black line)."

23) Line 145. Fig.2. The black dots represent presence of *Ae. albopictus*, but where does the data come from? There is a reference in the figure legend, but this should be mentioned in the main text. Also in the Eastern part of South America, *Ae. albopictus* is so abundant, you cannot clearly see the R_0 colours. Also see major comment 5.

Presence points for *Ae. albopictus* are based on data from Kraemer et al., 2015 (Ref 26 in the main text). You are correct, we only mentioned the data reference in the legend of Fig 2. We have now added a sentence in Method section (see below). We agree that the overlay hides the standardized R_0 model outputs over the eastern coasts of South America (Fig 1c), but the reader can still refer to Fig 1a (global maps for which we hide the presence points for that particular purpose) to see the model outputs.

Updated text (in Supp. Mat, section "Derivation of $R_0(T)$ for ZIKV – *Ae. albopictus* model"): "Observed presence points of *Ae. albopictus*, derived from Kraemer et al. 2015, are overlaid onto standardized R_0 maps for the recent period on Figure 2."

24) Line 149. 'they are consistent with former published estimates'. This is part of the Discussion?

Done

25) Line 166-174. Should be part of the Discussion.

Done

26) Line 190. Change: vector presence in 'black dots' and standardized R_0 in 'colour code'.

Done

27) Line 191. significantly 'statistically'? Or do you mean substantially?

Done

28) Line 219. What is not shown?

Done

Discussion

29) Line 232. Add that you based your model only on *Ae. albopictus*.

Done

30) Line 240. Table S6 should be changed to S5.

Done

31) Line 240-241. Rephrase sentence for clarity.

Done

32) Line 246. But were similar incubation times used in this study as in the study of Heitmann et al?

Similar, they went to 21 days, added to text

33) Line 253. Happened how, can you elaborate?

Done

34) Line 276. By which autochthonous species?

This is clearly detailed in the next sentence

35) Line 282. 'the' presence

Done

36) Line 298. Which 'is' important, change accordingly.

Done

37) Line 301. What is the distribution range of *Oc. detritus*? So what is its broader relevance? This needs to be elaborated.

There are no published maps of the distribution of this species (only that it is known to inhabit brackish waters) and we also refer to other known and unknown vectors, so the overall 'range' is necessarily vague.

38) Line 307. Here the role of *Oc. detritus* is discussed further, but the vector distribution and chances of vector-human presence should be taken into account. Discuss this in the context of vectorial capacity.

As above, much of the details needed to discuss this are unknown. We are simply showing that there is risk, we can't estimate the degree of this risk without excessive speculation on distribution and other unknown vectors.

39) Line 320. What does that mean? Why was it underestimated? Also see major comment 2.

Example was changed to a virus example in response to major comment 2

40) Line 327. But *Ae. albopictus* is absent in for example Suriname, so what is the link between the maps and vector presence? See major comment 5.

Our model estimated climate risk, with black dots on the map showing current presence of *Ae. albopictus*. See major comment 5 for more specific details

41) Line 342. I agree, but specify in what direction this difference is?

Unsure what this means, we cannot possibly specify whether different strains of ZIKV would result specifically in higher/lower competence or higher/lower EIP as we did not test them. All we can say is different viral strains may respond differently.

Acknowledgements

42) 406. Replace 'Raid' by 'Rapid'.

Done

References

43) Line 480. The references should be updated as much as possible. Some were cited in 2017, is there more recent literature? Not all species names are in italics and there are misplaced capital letters which do not need to be there when using Vancouver style referencing as is recommended for by Proc Roy Soc B.

Reference style has been changed and may references updated to more recent examples.

Figure legends

44) Line 615. See comment 51, then change condition into temperature.

Done

45) Line 621-622. In this sentence *Ae. aegypti* should be changed into *Ae. albopictus*? Also this is a result which should go into the text rather than in the figure legend. 'The δ_s for *Ae. albopictus* (0.27) was significantly greater than that of *Oc. detritus* (0.20) (Table S2), indicating that *Ae. aegypti* is more competent for ZIKV at all temperatures.'

Done, and sentence moved to main text.

Supplementary methods

46) Line 17. Add that plateau value = δ_s .

We are unsure about what the reviewer meant with that comment. The plateau value, δ_s , is a generic parameter for all considered logistic models. If we focus on Model 1 (linear assumption), then $\delta_1=0.2741$ (for *Ae. albopictus*) and $\delta_2=0.2003$ (for *Oc. detritus*) and these values are already provided in Table S2 and discussed in the text.

47) Line 28. See comment 20.

Done

48) Line 33. Fill in the supplement number.

49) Line 37. What are models 3-7? We don't see them in the manuscript? They are mentioned in Table S1, but completely unclear what they contain.

Models 3 and 4 respectively test if temperature or day effects are non-linear. They are compared to Model 1 using maximum likelihood ratios and they show non-significant effects, with low p-values. In other words, temperature (and day) effects was not significantly nonlinear when treated separately, so the linear assumption (Model 1) is valid and sufficient. We also tested interactions between all dependent variables (species, time and temperature) in Models 6, 7 and 8, and these effects are not significant when compared to the linear Model 1. The text has been updated accordingly in Supp Mat.

50) Line 34, 38 and 40. Example of wrongly placed text in the supplementary methods. They should be placed with the Figure or Table of interest.

This has been addressed as part of major comment 4

51) Line 48. Table S3. In the first column of the table, *Ae. albopictus* is not written in full. We would suggest to split the table in two, here only show numbers of the transmission experiment. Show the numbers of the survival experiment underneath Figure S1 and also refer to this new table in Line 99 of the main text.

We tried this, however, it wasn't possible without embedding two tables into a figure legend. It looked very confusing. Referring to it in the legend of S2 is much easier to read.

52) Line 61. Figure S1. For both species, is it not strange that the survival increases with time? See for example Oc. detritus at 21oC at day 14.

Survival is measured independently for each time point immediately prior to sacrifice, e.g. the 'day 21' time point does not include the individuals from the 'day 14' time point or vice versa. Hence a low mortality in the 'day 21' cohort lead to survival appearing to increase.

Text added to legend: (Note that survival is measured from an independent cohort for each time point, hence may 'increase' over time if a later cohort had higher survival).

53) Line 68. Figure S2 is referred to, but not discussed in the main text (line 619). Not clear what the figure adds?

Done

54) Line 74. See major comment 4, move to main text.

Done

55) Line 84 Table S4. In the table text, add meaning of Y, N and dpi.

Done

56) Line 89. Table S5. These are actually two tables, so split them. Most importantly it is unclear in what units the relative titres are expressed. Are these TCID₅₀, or PFU values? Compress the table by removing the 17oC row, and 0 and 5 days columns.

Demarcated them as S5A and S5B to avoid large duplication of the legend. The relative titres were measured by qRT-PCR (not added to text). Tables have been compressed as suggested.

57) Line 113. Which Fig. 2 and Table 1 show 'the related analytical functions'? In the main manuscript or in the supplements?

These should refer to Fig S4 and Table S7. This is now corrected.

58) Line 126-127. Can you describe this differently? It is not immediately clear how the standardized values correspond to the temperatures. Also this can not be derived from Fig.1 which is referred to.

The calculated standardized R_0 threshold values simply correspond to $\text{std_}R_0(T)=17^\circ\text{C}$ and $\text{std_}R_0(T)=19^\circ\text{C}$. The analytical function for $\text{std_}R_0(T)$ can be derived using the $R_0(T)$ equation provided in "Derivation of $R_0(T)$ for ZIKV – *Ae. albopictus* model" and parameter details available in Table S7. We referred to Fig S4d (and hence the related colour code) instead of Figure 1 (experimental EIP values). This is now corrected.

59) Line 129. What is the source of the temperature and climate model projections?

The sources of all climate data are described in further details in the subsequent sub-section "Observed climate datasets and climate change scenarios input data". For clarity we updated the text:

Modified text: "Further details about climate datasets are available in the following section "Observed climate datasets and climate change scenarios input data".

60) Line 141. Fig. S4. The difference between figure S4 and S5 is not immediately clear, can this be adjusted? In panel d the solid red line is not very visible

Fig S5 and Fig S4 look very similar indeed, we only changed the analytical functions for fitting the EIP10 (sub-panel b on Fig S4-S5). In Fig S4 we use a linear EIP10 function and in Fig S5 we use an exponential EIP10 function. The other parameters (biting rates and mortality rates) are the same. The resulting $R_0(T)$ function (sub-panel d on Fig S4-S5) is similar whatever the selected fit (exp or linear). The red line is most of the time overlaid on the black solid line, excepting at cold and warm temperature ranges (roughly below 20°C and above 35°C). Here we basically compare the analytical function of our former R_0 model (Caminade et al., 2017) that was based on epidemiological parameters for *Ae. albopictus* and DENV with the new lab derived EIP parameter for ZIKV. In other words, the new function (using the right pathogen e.g. ZIKV) shows smaller ZIKV transmission risk at cold and high temperature range with respect to our former study that used assumptions based on DENV and *Ae. albopictus*. For clarity we updated the caption of Fig S5:

(same as Fig. S4, but we now use an exponential instead of a linear fit for EIP10 on FigS5b) ...".

61) Line 201. The model comparison of proportion of saliva-positive mosquitoes is missing?

The title of that empty sub section should have been removed in the submitted version. All information included in the former "Proportion of saliva-positive mosquitoes" have been moved to "Models of extrinsic incubation period (EIP10)" (based on former reviewer's comments, text from "Table S1 gives the results of generalized likelihood ratio tests..." to "...are shown on Supplementary figure 3 and Supplementary Table 3").

Consequently, the title of this empty sub section has been removed in the re-submitted version of the manuscript.